# REPRESENTATION ALIGNMENT FOR GENERATION: TRAINING DIFFUSION TRANSFORMERS IS EASIER THAN YOU THINK

**Sihyun Yu**[1]   **Sangkyung Kwak**[1,3]   **Huiwon Jang**[1]
**Jongheon Jeong**[2]   **Jonathan Huang**[3]   **Jinwoo Shin**[1*]   **Saining Xie**[4*]
[1]KAIST  [2]Korea University  [3]Scaled Foundations  [4]New York University

## ABSTRACT

Recent studies have shown that the denoising process in (generative) diffusion models can induce meaningful (discriminative) representations inside the model, though the quality of these representations still lags behind those learned through recent self-supervised learning methods. We argue that one main bottleneck in training large-scale diffusion models *for generation* lies in effectively learning these representations. Moreover, training can be made easier by incorporating high-quality external visual representations, rather than relying solely on the diffusion models to learn them independently. We study this by introducing a straightforward regularization called *REPresentation Alignment (REPA)*, which aligns the projections of noisy input hidden states in denoising networks with clean image representations obtained from external, pretrained visual encoders. The results are striking: our simple strategy yields significant improvements in both training efficiency and generation quality when applied to popular diffusion and flow-based transformers, such as DiTs and SiTs. For instance, our method can speed up SiT training by over $17.5\times$, matching the performance (without classifier-free guidance) of a SiT-XL model trained for 7M steps in less than 400K steps. In terms of final generation quality, our approach achieves state-of-the-art results of FID=1.42 using classifier-free guidance with the guidance interval.

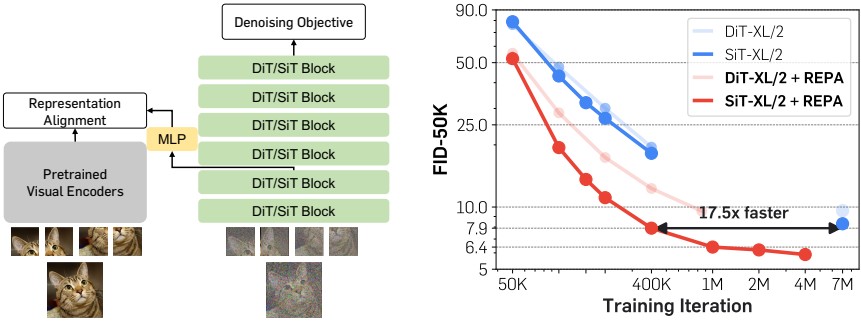

Figure 1: **Representation alignment makes diffusion transformer training significantly easier.** Our framework, REPA, explicitly aligns the diffusion model representation with powerful pretrained visual representation through a simple regularization. Notably, model training becomes significantly more efficient and effective, and achieves $>17.5\times$ faster convergence than the vanilla model.

## 1 INTRODUCTION

Generative models based on *denoising*, such as diffusion models (Ho et al., 2020; Song et al., 2021) and flow-based models (Albergo & Vanden-Eijnden, 2023; Lipman et al., 2022; Liu et al., 2023), have been a scalable approach in generating high-dimensional visual data. They achieve remarkably successful results in challenging tasks such as zero-shot text-to-image (Podell et al., 2023; Saharia et al., 2022; Esser et al., 2024) or text-to-video (Polyak et al., 2024; Brooks et al., 2024) generation.

---

[*]Equal advising.

**Project page:** `https://sihyun.me/REPA`

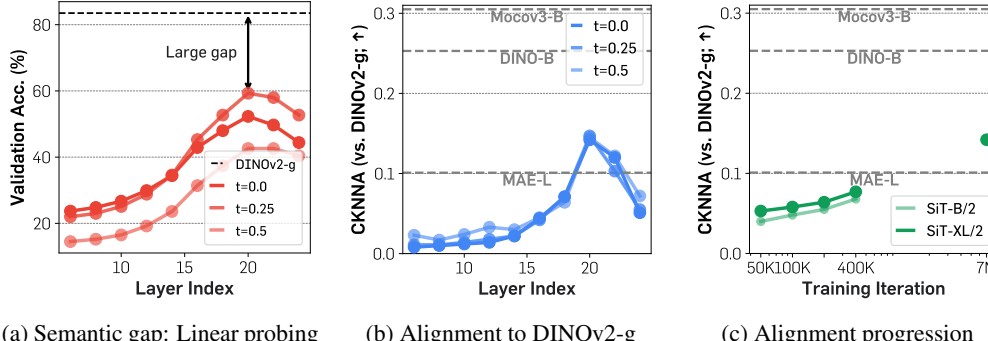

(a) Semantic gap: Linear probing     (b) Alignment to DINOv2-g     (c) Alignment progression

Figure 2: **Alignment behavior for a pretrained SiT model.** We empirically investigate the feature alignment between DINOv2-g and the original SiT-XL/2 checkpoint trained for 7M iterations. (a) While SiT learns semantically meaningful representations, a significant gap remains compared to DINOv2. (b) Using CKNNA (Huh et al., 2024), we observe that SiT already shows some alignment with DINOv2, though its absolute value is lower compared to other vision encoders. (c) Alignment improves with a larger model and longer training, but the progress remains slow and insufficient.

Recent works have explored the use of diffusion models as representation learners (Li et al., 2023a; Xiang et al., 2023; Chen et al., 2024c; Mukhopadhyay et al., 2021) and have shown that they learn discriminative features in their hidden states, and better diffusion models learn better representations (Xiang et al., 2023). In fact, this observation is closely related to earlier approaches that employ *denoising score matching* (Vincent, 2011) as a self-supervised learning method (Bengio et al., 2013), which implicitly learns a representation $\mathbf{h}$ as a hidden state of a denoising autoencoder $\mathbf{s}_\theta(\tilde{\mathbf{x}})$ through a *reconstruction* of $\mathbf{x}$ from the corrupted data $\tilde{\mathbf{x}}$ (Yang & Wang, 2023). However, the reconstruction task may not be a suitable task for learning good representations, as it is not capable of eliminating unnecessary details in $\mathbf{x}$ for representation learning (LeCun, 2022; Assran et al., 2023).

**Our approach.** In this paper, we identify that the main challenge in training diffusion models stems from the need to learn a high-quality internal representation $\mathbf{h}$. We demonstrate that the training process for generative diffusion models becomes significantly easier and more effective when supported by an external representation, $\mathbf{y}_*$. Specifically, we propose a simple regularization technique that leverages recent advances in self-supervised visual representations as $\mathbf{y}_*$, leading to substantial improvements in both training efficiency and the generation quality of diffusion transformers.

We start by performing an empirical analysis with recent diffusion transformers (Peebles & Xie, 2023; Ma et al., 2024a) and the state-of-the-art self-supervised vision model, DINOv2 (Oquab et al., 2024). Similar to prior studies (Xiang et al., 2023), we first observe that pretrained diffusion models do indeed learn meaningful discriminative representations (as shown by the linear probing results in Figure 2a). However, these representations are significantly inferior to those produced by DINOv2. Next, we find that the alignment between the representations learned by the diffusion model and those of DINOv2 (Figure 2b) is still considered weak,[1] which we study by measuring their *representation alignment* (Huh et al., 2024). Finally, we observe this alignment between diffusion models and DINOv2 improves consistently with longer training and larger models (Figure 2c).

These insights inspire us to enhance generative models by incorporating external self-supervised representations. However, this approach is not straightforward when using off-the-shelf self-supervised visual encoders (*e.g.*, by fine-tuning an encoder for generation tasks). The first challenge is an input mismatch: diffusion models work with noisy inputs $\tilde{\mathbf{x}}$, whereas most self-supervised learning encoders are trained on clean images $\mathbf{x}$. This issue is even more pronounced in modern *latent diffusion* models, which take a compressed latent image $\mathbf{z} = E(\mathbf{x})$ from a pretrained VAE encoder (Rombach et al., 2022) as input. Additionally, these off-the-shelf vision encoders are not designed for tasks like reconstruction or generation. To overcome these technical hurdles, we guide the feature learning of diffusion models using a *regularization* technique that distills pretrained self-supervised representations into diffusion representations, offering a flexible way to integrate high-quality representations.

---

[1] We describe this as "weak" because relatively, the alignments are much poorer than those seen with other self-supervised encoders (*e.g.*, MoCov3 (Chen et al., 2021)), even after extensive training.

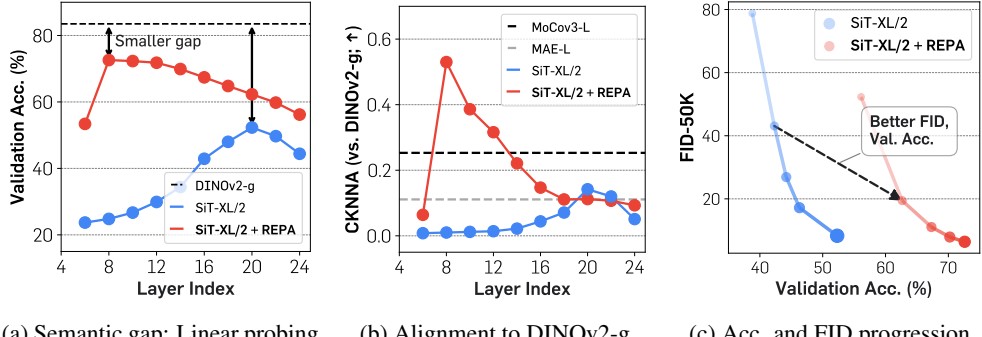

(a) Semantic gap: Linear probing    (b) Alignment to DINOv2-g    (c) Acc. and FID progression

Figure 3: **Bridging the representation gap:** (a) Our method, REPA significantly reduces the "semantic gap" between diffusion transformers and DINOv2, as demonstrated by the linear probing results on ImageNet classification. (b) With REPA, the alignment between diffusion transformers and DINOv2 improves substantially, even after just a few (*e.g.*, 8) layers. (c) Notably, with improved alignment, we can push the SiT model's generation-representation envelope: within the same number of training iterations, it delivers both better generation quality and stronger linear probing results. We use a single network trained with REPA at layer 8 and perform the evaluation at different layers.

Specifically, we introduce *REPresentation Alignment* (REPA), a simple regularization technique built on recent diffusion transformer architectures (Peebles & Xie, 2023). In essence, REPA distills the pretrained self-supervised visual representation $\mathbf{y}_*$ of a clean image $\mathbf{x}$ into the diffusion transformer representation $\mathbf{h}$ of a noisy input $\tilde{\mathbf{x}}$. This regularization reduces the semantic gap in the representation $\mathbf{h}$ (Figure 3a) and better aligns it with the target self-supervised representations $\mathbf{y}_*$ (Figure 3b). Notably, this enhanced alignment significantly boosts the *generation* performance of diffusion transformers (Figure 3c). Interestingly, with REPA, we observe that sufficient representation alignment can be achieved by aligning only the first few transformer blocks. This, in turn, allows the later layers of the diffusion transformers to focus on capturing high-frequency details based on the aligned representations, further improving generation performance.

Based on our analysis, we conduct a system-level comparison to demonstrate the effectiveness of our scheme by applying it to two recent diffusion transformers: DiTs (Peebles & Xie, 2023) and SiTs (Ma et al., 2024a). For SiT training, we show the model achieves FID=7.9 on class-conditional ImageNet (Deng et al., 2009) generation only using 400K training iteration (without classifier-free guidance; Ho & Salimans 2022) which is >17.5× faster than the vanilla SiTs. Moreover, with classifier-free guidance, our scheme shows an improved FID at the final from 2.06 to 1.80 and achieves state-of-the-art results of FID=1.42 with guidance interval (Kynkäänniemi et al., 2024).

We highlight the main contributions of this paper below:

- We hypothesize that learning high-quality representations in diffusion transformers is essential for improving their generation performance.
- We introduce REPA, a simple regularization for aligning diffusion transformer representations with strong self-supervised visual representations.
- Our framework improves the generation performance of diffusion transformers, *e.g.*, for SiTs, we achieve a 17.5× faster training for SiTs and improved FID scores on ImageNet generation.

## 2 PRELIMINARIES

We present a brief overview of *flow and diffusion-based* models through the unified perspective of *stochastic interpolants* (Albergo et al., 2023; Ma et al., 2024a); see Appendix A for more details.

We consider a continuous time-dependent process with a data $\mathbf{x}_* \sim p(\mathbf{x})$ and a Gaussian noise $\epsilon \sim \mathcal{N}(\mathbf{0}, \mathbf{I})$ on $t \in [0, T]$:

$$\mathbf{x}_t = \alpha_t \mathbf{x}_* + \sigma_t \epsilon, \quad \alpha_0 = \sigma_T = 1, \ \alpha_T = \sigma_0 = 0, \tag{1}$$

where $\alpha_t$ and $\sigma_t$ are a decreasing and increasing function of $t$, respectively. Given such a process, there exists a *probability flow ordinary differential equation* (PF ODE) with a velocity field

$$\dot{\mathbf{x}}_t = \mathbf{v}(\mathbf{x}_t, t), \tag{2}$$

where the distribution of this ODE at $t$ is equal to the marginal $p_t(\mathbf{x})$. Thus, data can be sampled by solving this PF ODE in Eq. (2) through existing ODE samplers (*e.g.*, Euler sampler) starting from a random Gaussian noise $\epsilon \sim \mathcal{N}(\mathbf{0}, \mathbf{I})$ (Lipman et al., 2022; Ma et al., 2024a).

This velocity $\mathbf{v}(\mathbf{x}, t)$ is represented as the following sum of two conditional expectations

$$\mathbf{v}(\mathbf{x}, t) = \mathbb{E}[\dot{\mathbf{x}}_t | \mathbf{x}_t = \mathbf{x}] = \dot{\alpha}_t \mathbb{E}[\mathbf{x}_* | \mathbf{x}_t = \mathbf{x}] + \dot{\sigma}_t \mathbb{E}[\epsilon | \mathbf{x}_t = \mathbf{x}], \tag{3}$$

which can be approximated with model $\mathbf{v}_\theta(\mathbf{x}_t, t)$ by minimizing the following training objective:

$$\mathcal{L}_{\text{velocity}}(\theta) := \mathbb{E}_{\mathbf{x}_*, \epsilon, t}\left[||\mathbf{v}_\theta(\mathbf{x}_t, t) - \dot{\alpha}_t \mathbf{x}_* - \dot{\sigma}_t \epsilon||^2\right]. \tag{4}$$

Moreover, there exists a reverse *stochastic differential equation* (SDE) that the marginal $p_t(\mathbf{x})$ coincides with the one of PF ODE in Eq. (2) with a diffusion coefficient $w_t$ (Ma et al., 2024a):

$$d\mathbf{x}_t = \mathbf{v}(\mathbf{x}_t, t)dt - \frac{1}{2}w_t \mathbf{s}(\mathbf{x}_t, t)dt + \sqrt{w_t}d\bar{\mathbf{w}}_t, \tag{5}$$

where the score $\mathbf{s}(\mathbf{x}_t, t)$ is the following conditional expectation

$$\mathbf{s}(\mathbf{x}_t, t) = -\sigma_t^{-1}\mathbb{E}[\epsilon | \mathbf{x}_t = \mathbf{x}]. \tag{6}$$

and it can be directly computed using the velocity $\mathbf{v}(\mathbf{x}, t)$ for $t > 0$ as

$$\mathbf{s}(\mathbf{x}, t) = \sigma_t^{-1} \cdot \frac{\alpha_t \mathbf{v}(\mathbf{x}, t) - \dot{\alpha}_t \mathbf{x}}{\dot{\alpha}_t \sigma_t - \alpha_t \dot{\sigma}_t}, \tag{7}$$

implying that data can be alternatively generated through Eq. (5) with SDE solvers.

Following Ma et al. (2024a), we mainly consider a simple linear interpolant with restricting $T = 1$: $\alpha_t = 1 - t$ and $\sigma_t = t$. However, our approach is applicable to any similar variants (*e.g.*, DDPM; Ho et al. 2020), which has a similar formulation but uses a discretized process and different $\alpha_t, \sigma_t$ that $\mathcal{N}(\mathbf{0}, \mathbf{I})$ becomes an equilibrium distribution (*i.e.*, $\mathbf{x}_t$ converges to $\mathcal{N}(\mathbf{0}, \mathbf{I})$ only if $t \to \infty$).

## 3  REPA: REGULARIZATION FOR REPRESENTATION ALIGNMENT

### 3.1  OVERVIEW

Let $p(\mathbf{x})$ be an unknown target distribution for data $\mathbf{x} \in \mathcal{X}$. Our goal is to approximate $p(\mathbf{x})$ through a model distribution using a dataset drawn from $p(\mathbf{x})$. To lower computational costs, we adopt the recent prevalent *latent diffusion* (Rombach et al., 2022). This involves learning a latent distribution $p(\mathbf{z})$, which is defined as the distribution of a compressed latent variable $\mathbf{z} = E(\mathbf{x})$, where $E$ is an encoder from a pretrained autoencoder (*e.g.*, KL-VAE; Rombach et al. 2022), with $\mathbf{x} \sim p_{\text{data}}(\mathbf{x})$.

We aim to learn this distribution by training a diffusion model $\mathbf{v}_\theta(\mathbf{z}_t, t)$ using objectives such as velocity prediction, as described in Section 2. Here, we revisit denoising score matching within the context of self-supervised representation learning (Bengio et al., 2013). From this perspective, one can think of the diffusion model $\mathbf{v}_\theta(\mathbf{z}_t, t)$ as a composition of two functions $g_\theta \circ f_\theta$ with an encoder $f_\theta : \mathcal{Z} \to \mathcal{H}$ with $f_\theta(\mathbf{z}_t) = \mathbf{h}_t$ and a decoder $g_\theta : \mathcal{H} \to \mathcal{Z}$ with $g_\theta(\mathbf{h}_t) = \mathbf{v}_t$, where the encoder $f_\theta$ implicitly learns a representation $\mathbf{h}_t$ that reconstructs the target $\mathbf{v}_t$.

However, learning a good representation through producing a prediction of the input space (*e.g.*, generating pixels) can be challenging, as the model is often not capable of eliminating unnecessary details, which is crucial for developing a strong representation (LeCun, 2022; Assran et al., 2023). We argue that a key bottleneck in the training of large-scale diffusion models *for generation* lies in representation learning, an area where current diffusion models fall short. We also hypothesize that the training process can be made easier by guiding the model with high-quality external visual representations, rather than relying solely on the diffusion model to learn them independently.

To address this challenge, we introduce a simple regularization method called *REPresentation Alignment* (REPA) using the recent diffusion transformer architectures (Peebles & Xie, 2023; Ma et al., 2024a) (see Appendix B for an illustration). In a nutshell, our regularization distills pretrained self-supervised visual representations to diffusion transformers in a simple and effective way. This allows the diffusion model to leverage these semantically rich external representations for generation, leading to a substantial boost in performance.

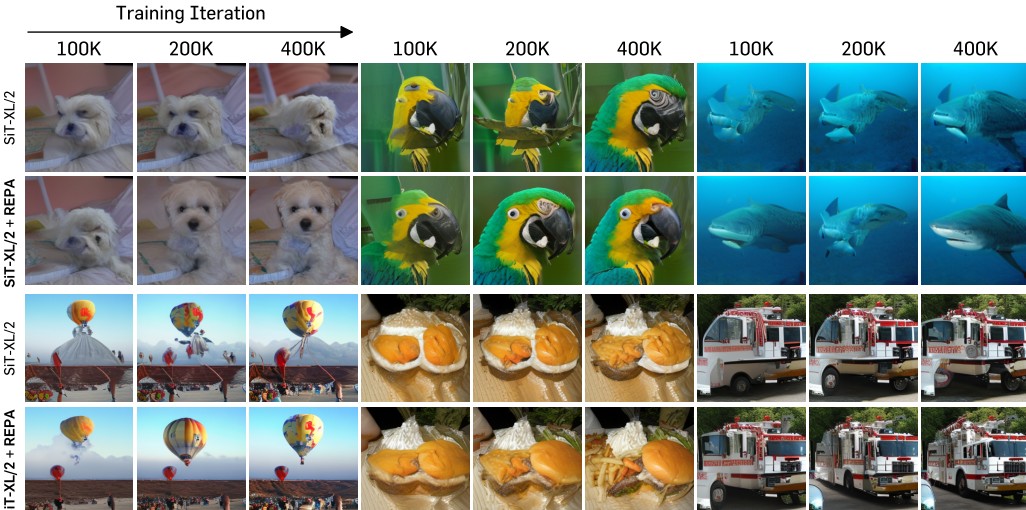

Figure 4: **REPA improves visual scaling.** We compare the images generated by two SiT-XL/2 models during the first 400K iterations, with REPA applied to one of the models. Both models share the same noise, sampler, and number of sampling steps, and neither uses classifier-free guidance.

## 3.2 OBSERVATIONS

To take a deeper dive into this, we first investigate the layer-wise behavior of the pretrained SiT model (Ma et al., 2024a) on ImageNet (Deng et al., 2009), which uses linear interpolants and velocity prediction for training. In particular, we focus on measuring the *representation gap* between the diffusion transformer and the state-of-the-art self-supervised DINOv2 model (Oquab et al., 2024). We examine this from three angles: semantic gap, feature alignment progression, and their final feature alignment. For the *semantic gap*, we compare linear probing results using DINOv2 features with those from SiT models trained for 7M iterations, following the same protocol as in Xiang et al. (2023), which involves linear probing on globally pooled hidden states of the diffusion transformer. Next, to measure *feature alignments*, we use CKNNA (Huh et al., 2024), a kernel alignment metric related to CKA (Kornblith et al., 2019), but based on mutual nearest neighbors. This allows for a quantitative assessment of alignment between different representations. We summarize the result in Figure 2 and more details (*e.g.*, definition of CKNNA) in Appendix C.1.

**Diffusion transformers exhibit a significant semantic gap from state-of-the-art visual encoders.** As shown in Figure 2a, we observe that the hidden state representation of the pretrained diffusion transformer, in line with prior works (Xiang et al., 2023; Chen et al., 2024c), achieves a reasonably high linear probing peak at layer 20. However, its performance remains well below that of DINOv2, indicating a substantial semantic gap between the two representations. Additionally, we find that after reaching this peak, linear probing performance quickly declines, suggesting that the diffusion transformer must shift away from focusing solely on learning semantically-rich representations in order to generate images with high-frequency details.

**Diffusion representations are already (weakly) aligned with other visual representations.** In Figure 2b, we report representational alignments between SiT and DINOv2 using CKNNA. In particular, the SiT model representation already shows better alignment than MAE (He et al., 2022), which is also a self-supervised learning approach based on the reconstruction of masked patches. However, the absolute alignment score remains lower than that observed between other self-supervised learning methods (*e.g.*, MoCov3 (Chen et al., 2021) *vs.* DINOv2). These results suggest that while diffusion transformer representations exhibit some alignment with self-supervised visual representations, the alignment remains weak.

**Alignment improves with larger models and extended training.** We also measure CKNNA values across different model sizes and training iterations. As depicted in Figure 2c, we observe improved alignment with larger models and extended training. However, the absolute alignment remains low and does not reach the levels observed between other self-supervised visual encoders (*e.g.*, MoCov3 and DINOv2), even after extensive training of 7M iterations.

These findings are not unique to the SiT model but are also observed in other denoising-based generative transformers. For instance, in Figure 2, we present a similar analysis using a DiT model (Peebles & Xie, 2023) pretrained on ImageNet with the DDPM objective (Ho et al., 2020; Nichol & Dhariwal, 2021). See Appendix C.2 for more details.

## 3.3 REPRESENTATION ALIGNMENT WITH SELF-SUPERVISED REPRESENTATIONS

REPA aligns patch-wise projections of the model's hidden states with pretrained self-supervised visual representations. Specifically, we use the *clean* image representation as the target and explore its impact. The goal of this regularization is for the diffusion transformer's hidden states to predict noise-invariant, clean visual representations from noisy inputs that contain useful semantic information. This provides meaningful guidance for the subsequent layers to reconstruct the target.

Formally, let $f$ be a pretrained encoder and consider a clean image $\mathbf{x}_*$. Let $\mathbf{y}_* = f(\mathbf{x}_*) \in \mathbb{R}^{N \times D}$ be an encoder output, where $N, D > 0$ are the number of patches and the embedding dimension of $f$, respectively. REPA aligns $h_\phi(\mathbf{h}_t) \in \mathbb{R}^{N \times D}$ with $\mathbf{y}_*$, where $h_\phi(\mathbf{h}_t)$ is a projection of an diffusion transformer encoder output $\mathbf{h}_t = f_\theta(\mathbf{z}_t)$ that through a trainable projection head $h_\phi$. In practice, we simply parameterize $h_\phi$ using a multilayer perceptron (MLP).

In particular, REPA achieves alignment through a maximization of patch-wise similarities between the pretrained representation $\mathbf{y}_*$ and the hidden state $\mathbf{h}_t$:

$$\mathcal{L}_{\text{REPA}}(\theta, \phi) := -\mathbb{E}_{\mathbf{x}_*, \boldsymbol{\epsilon}, t}\left[\frac{1}{N}\sum_{n=1}^{N} \text{sim}(\mathbf{y}_*^{[n]}, h_\phi(\mathbf{h}_t^{[n]}))\right], \tag{8}$$

where $n$ is a patch index and $\text{sim}(\cdot, \cdot)$ is a pre-defined similarity function.

In practice, we add this term to the original diffusion-based objectives described in Section 2 and Appendix A. For instance, for the training of a velocity model in Eq. (4), the objective becomes:

$$\mathcal{L} := \mathcal{L}_{\text{velocity}} + \lambda \mathcal{L}_{\text{REPA}} \tag{9}$$

where $\lambda > 0$ is a hyperparameter that controls the tradeoff between denoising and representation alignment. We primarily investigate the impact of this regularization on two popular objectives: Improved DDPM (Nichol & Dhariwal, 2021) used in DiT (Peebles & Xie, 2023) and linear stochastic interpolants used in SiT (Ma et al., 2024a), though other objectives can also be considered.

## 4 EXPERIMENTS

We validate the performance of REPA and the effect of the proposed components through extensive experiments. In particular, we investigate the following questions:

- Can REPA improve diffusion transformer training significantly? (Table 2, 3, 4, Figure 4, 6)
- Is REPA scalable in terms of model size and representation quality? (Table 2, Figure 5)
- Can diffusion model representations be aligned with various visual representations? (Figure 8)

## 4.1 SETUP

**Implementation details.** We strictly follow the setup in DiT (Peebles & Xie, 2023) and SiT (Ma et al., 2024a) unless otherwise specified. Specifically, we use ImageNet (Deng et al., 2009), where each image is preprocessed to the resolution of 256×256 (denoted as ImageNet 256×256), and follow ADM (Dhariwal & Nichol,

Table 1: Model configuration details.

| Config | #Layers | Hidden dim | #Heads |
|--------|---------|------------|--------|
| B/2    | 12      | 768        | 12     |
| L/2    | 24      | 1024       | 16     |
| XL/2   | 28      | 1152       | 16     |

2021) for other data preprocessing protocols. Each image is then encoded into a compressed vector $\mathbf{z} \in \mathbb{R}^{32 \times 32 \times 4}$ using the Stable Diffusion VAE (Rombach et al., 2022). For model configurations, we use the B/2, L/2, and XL/2 architectures introduced in the DiT and SiT papers, which process inputs with a patch size of 2 (see Table 1 for details). To ensure a fair comparison with DiTs and SiTs, we consistently use a batch size of 256 during training. Additional experimental details, including hyperparameter settings and computing resources, are provided in Appendix D.

Table 2: **Component-wise analysis** on ImageNet 256×256. All models are SiT-L/2 trained for 400K iterations. All metrics except accuracy (Acc.) are measured with the SDE Euler-Maruyama sampler with NFE=250 and without classifier-free guidance. For Acc., we report linear probing results on the ImageNet validation set using the latent features aligned with the target representation. We fix $\lambda = 0.5$ here. ↓ and ↑ indicate whether lower or higher values are better, respectively.

| Iter. | Target Repr. | Depth | Objective | FID↓ | sFID↓ | IS↑ | Pre.↑ | Rec.↑ | Acc.↑ |
|---|---|---|---|---|---|---|---|---|---|
| 400K | Vanilla SiT-L/2 (Ma et al., 2024a) | | | 18.8 | 5.29 | 72.0 | 0.64 | 0.64 | N/A |
| 400K | MAE-L | 8 | NT-Xent | 12.5 | 4.89 | 90.7 | 0.68 | 0.63 | 57.3 |
| 400K | DINO-B | 8 | NT-Xent | 11.9 | 5.00 | 92.9 | 0.68 | 0.63 | 59.3 |
| 400K | MoCov3-L | 8 | NT-Xent | 11.9 | 5.06 | 92.2 | 0.68 | 0.64 | 63.0 |
| 400K | I-JEPA-H | 8 | NT-Xent | 11.6 | 5.21 | 98.0 | 0.68 | 0.64 | 62.1 |
| 400K | CLIP-L | 8 | NT-Xent | 11.0 | 5.25 | 100.4 | 0.67 | 0.66 | 67.2 |
| 400K | SigLIP-L | 8 | NT-Xent | 10.2 | 5.15 | 107.0 | 0.69 | 0.64 | 68.8 |
| 400K | DINOv2-L | 8 | NT-Xent | 10.0 | 5.09 | 106.6 | 0.68 | 0.65 | 68.1 |
| 400K | DINOv2-B | 8 | NT-Xent | 9.7 | 5.13 | 107.5 | 0.69 | 0.64 | 65.7 |
| 400K | DINOv2-L | 8 | NT-Xent | 10.0 | 5.09 | 106.6 | 0.68 | 0.65 | 68.1 |
| 400K | DINOv2-g | 8 | NT-Xent | 9.8 | 5.22 | 108.9 | 0.69 | 0.64 | 65.7 |
| 400K | DINOv2-L | 6 | NT-Xent | 10.3 | 5.23 | 106.5 | 0.69 | 0.65 | 66.2 |
| 400K | DINOv2-L | 8 | NT-Xent | 10.0 | 5.09 | 106.6 | 0.68 | 0.65 | 68.1 |
| 400K | DINOv2-L | 10 | NT-Xent | 10.5 | 5.50 | 105.0 | 0.68 | 0.65 | 68.6 |
| 400K | DINOv2-L | 12 | NT-Xent | 11.2 | 5.14 | 100.2 | 0.68 | 0.64 | 69.4 |
| 400K | DINOv2-L | 14 | NT-Xent | 11.6 | 5.61 | 99.5 | 0.67 | 0.65 | 70.0 |
| 400K | DINOv2-L | 16 | NT-Xent | 12.1 | 5.34 | 96.1 | 0.67 | 0.64 | 71.1 |
| 400K | DINOv2-L | 8 | NT-Xent | 10.0 | 5.09 | 106.6 | 0.68 | 0.65 | 68.1 |
| 400K | DINOv2-L | 8 | Cos. sim. | 9.9 | 5.34 | 111.9 | 0.68 | 0.65 | 68.2 |

**Evaluation.** We report Fréchet inception distance (FID; Heusel et al. 2017), sFID (Nash et al., 2021), inception score (IS; Salimans et al. 2016), precision (Pre.) and recall (Rec.) (Kynkäänniemi et al., 2019) using 50,000 samples. We also include linear probing results (Acc.) and CKNNA (Huh et al., 2024) as discussed in Section 3.2. We provide more details of each metric in Appendix E.

**Sampler.** Following SiT (Ma et al., 2024a), we always use the SDE Euler-Maruyama sampler (for SDE with $w_t = \sigma_t$) and set the number of function evaluations (NFE) as 250 by default.

**Baselines.** We use several recent diffusion-based generation methods as baselines, each employing different inputs and network architectures. Specifically, we consider the following four types of approaches: (a) *Pixel diffusion*: ADM (Dhariwal & Nichol, 2021), VDM++ (Kingma & Gao, 2024), Simple diffusion (Hoogeboom et al., 2023), CDM (Ho et al., 2022), (b) *Latent diffusion with U-Net*: LDM (Rombach et al., 2022), (c) *Latent diffusion with transformer+U-Net hybrid models*: U-ViT-H/2 (Bao et al., 2023), DiffiT (Hatamizadeh et al., 2024), and MDTv2-XL/2 (Gao et al., 2023), and (d) *Latent diffusion with transformers*: MaskDiT (Zheng et al., 2024), SD-DiT (Zhu et al., 2024), DiT (Peebles & Xie, 2023), and SiT (Ma et al., 2024a). Here, we refer to Transformer+U-Net hybrid models that contain skip connections, which are not originally used in pure transformer architecture. Detailed descriptions of each baseline method are provided in Appendix F.

## 4.2 COMPONENT-WISE ANALYSIS

We answer the question of whether REPA leads to improved diffusion transformer training. As shown in Table 2, we discover that REPA consistently provides a substantially improved generation performance across various design choices, achieving a much better FID score than the vanilla model. Below, we provide a detailed analysis of the impact of each component.

**Target representation.** We begin by analyzing the effect of using different pretrained self-supervised encoders as the target representation. Notably, there is a strong correlation between the quality of these encoders and the performance of the corresponding aligned diffusion transformers. When a diffusion transformer is aligned with a pretrained encoder that offers more semantically meaningful representations (*i.e.*, better linear probing results), the model not only captures better semantics but also exhibits enhanced generation performance, as reflected by improved validation accuracy with linear probing and lower FID scores.

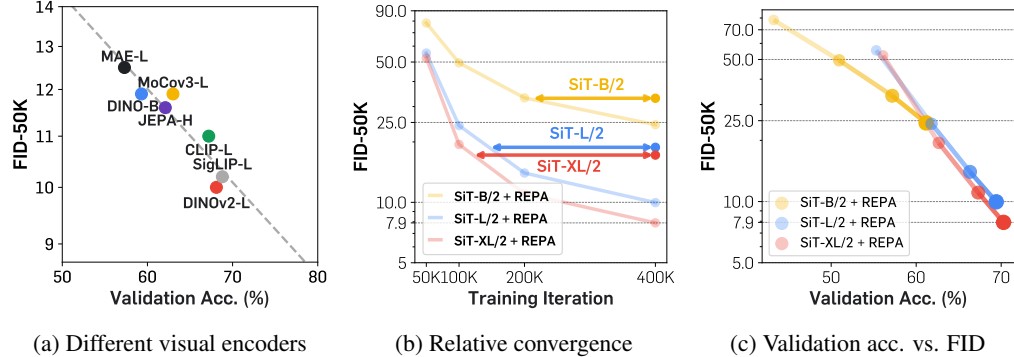

(a) Different visual encoders      (b) Relative convergence      (c) Validation acc. vs. FID

Figure 5: **Scalability of REPA.** (a) Linear probing vs. FID plot of REPA with different target encoders (400K iterations). A stronger encoder improves both discrimination and generation performance. (b) The relative improvement of REPA over the vanilla model becomes increasingly significant as the model size grows. (c) With a fixed target encoder, larger models reach better performance more quickly. In the line plot, results are marked at 50K, 100K, 200K, and 400K iters.

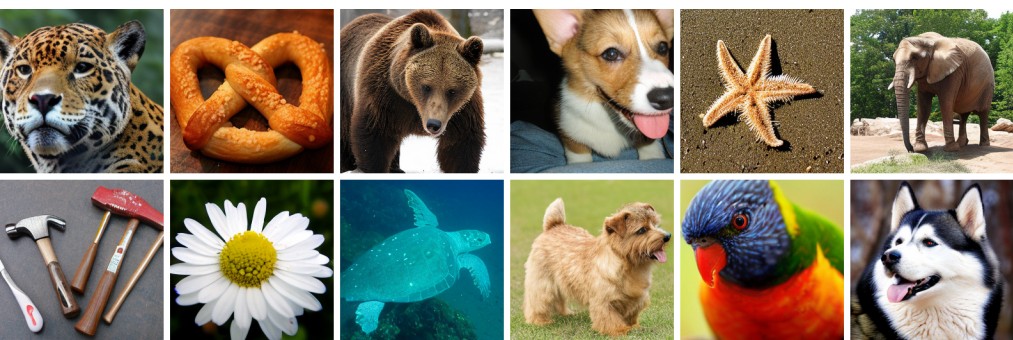

Figure 6: **Selected samples on ImageNet 256×256** from the SiT-XL/2 + REPA model. We use classifier-free guidance with $w = 4.0$.

**Target encoder size.** Next, we investigate the impact of different target representation encoder sizes by evaluating various DINOv2 models (*i.e.*, DINOv2-B, -L, -g). We observe that the performance differences are marginal, which we hypothesize is due to all DINOv2 models being distilled from the DINOv2-g model and thus sharing similar representations.

**Alignment depth.** We also examine the effect of attaching the REPA loss to different layers. We find that regularizing only the first few layers (*e.g.*, 8) in training is sufficient, as indicated by the linear probing results in Table 2. Interestingly, limiting regularization to the first few layers further enhances generation performance (*e.g.*, adding REPA to layer 6 or 8 yields best results). We hypothesize that this enables the remaining layers to concentrate on capturing high-frequency details, building on a strong representation. In future experiments, we apply REPA to the first 8 layers.

**Alignment objective.** We compare two simple training objectives for alignment: Normalized Temperature-scaled Cross Entropy (NT-Xent; Chen et al. 2020a) or negative cosine similarity (cos. sim.). Empirically, we find that NT-Xent offers advantages in the early stages (*e.g.*, 50-100K iterations), but the gap diminishes over time. Thus, we opt for cos. sim. in future experiments.

**Scalability.** Lastly, we investigate the scalability of REPA by varying the model sizes of both the target representation encoders and the diffusion transformers. In general, as summarized in Figure 5a, aligning with stronger representations improves both the generation results and the linear probing performance. Moreover, the convergence speed-up from REPA becomes more significant as the diffusion transformer model increases in size. We demonstrate this by plotting FID-50K of different SiT models with and without REPA in Figure 5b: REPA achieves the same FID level more quickly with larger models. Lastly, Figure 5c highlights the relationship between linear probing results and FID scores as model size varies, while keeping the target representation encoder fixed as DINOv2-B. Larger models exhibit a steeper performance improvement (*i.e.*, faster gains in both generation and linear evaluation) with longer training.

Table 3: **FID comparisons with vanilla DiTs and SiTs** on ImageNet 256×256. We do not use classifier-free guidance (CFG). ↓ denotes lower values are better. Iter. indicates the training iteration.

| Model | #Params | Iter. | FID↓ |
|---|---|---|---|
| DiT-L/2 | 458M | 400K | 23.3 |
| + REPA (ours) | 458M | **400K** | **15.6** |
| DiT-XL/2 | 675M | 400K | 19.5 |
| + REPA (ours) | 675M | **400K** | **12.3** |
| DiT-XL/2 | 675M | **7M** | 9.6 |
| + REPA (ours) | 675M | **850K** | 9.6 |
| SiT-B/2 | 130M | 400K | 33.0 |
| + REPA (ours) | 130M | **400K** | **24.4** |
| SiT-L/2 | 458M | 400K | 18.8 |
| + REPA (ours) | 458M | **400K** | **9.7** |
| + REPA (ours) | 458M | **700K** | **8.4** |
| SiT-XL/2 | 675M | 400K | 17.2 |
| + REPA (ours) | 675M | **150K** | **13.6** |
| SiT-XL/2 | 675M | 7M | 8.3 |
| + REPA (ours) | 675M | **400K** | **7.9** |
| + REPA (ours) | 675M | **1M** | **6.4** |
| + REPA (ours) | 675M | **4M** | **5.9** |

Table 4: **System-level comparison** on ImageNet 256×256 with CFG. ↓ and ↑ indicate whether lower or higher values are better, respectively. Results that include additional CFG scheduling are marked with an asterisk (*), where the guidance interval from (Kynkäänniemi et al., 2024) is applied for REPA.

| Model | Epochs | FID↓ | sFID↓ | IS↑ | Pre.↑ | Rec.↑ |
|---|---|---|---|---|---|---|
| *Pixel diffusion* | | | | | | |
| ADM-U | 400 | 3.94 | 6.14 | 186.7 | 0.82 | 0.52 |
| VDM++ | 560 | 2.40 | - | 225.3 | - | - |
| Simple diffusion | 800 | 2.77 | - | 211.8 | - | - |
| CDM | 2160 | 4.88 | - | 158.7 | - | - |
| *Latent diffusion, U-Net* | | | | | | |
| LDM-4 | 200 | 3.60 | - | 247.7 | 0.87 | 0.48 |
| *Latent diffusion, Transformer + U-Net hybrid* | | | | | | |
| U-ViT-H/2 | 240 | 2.29 | 5.68 | 263.9 | 0.82 | 0.57 |
| DiffiT* | - | 1.73 | - | 276.5 | 0.80 | 0.62 |
| MDTv2-XL/2* | 1080 | 1.58 | 4.52 | 314.7 | 0.79 | 0.65 |
| *Latent diffusion, Transformer* | | | | | | |
| MaskDiT | 1600 | 2.28 | 5.67 | 276.6 | 0.80 | 0.61 |
| SD-DiT | 480 | 3.23 | - | - | - | - |
| DiT-XL/2 | 1400 | 2.27 | 4.60 | 278.2 | **0.83** | 0.57 |
| SiT-XL/2 | 1400 | 2.06 | 4.50 | 270.3 | 0.82 | 0.59 |
| + REPA (ours) | 200 | 1.96 | **4.49** | 264.0 | 0.82 | 0.60 |
| + REPA (ours) | 800 | 1.80 | 4.50 | 284.0 | 0.81 | 0.61 |
| + REPA (ours)* | **800** | **1.42** | 4.70 | **305.7** | 0.80 | **0.65** |

## 4.3 SYSTEM-LEVEL COMPARISON

Based on the analysis, we perform a system-level comparison between recent state-of-the-art diffusion model approaches and diffusion transformers with REPA. First, we compare the FID values between vanilla DiT or SiT models and the same models trained with REPA. As shown in Table 3, REPA shows consistent and significant improvement across all model variants. In particular, on SiT-XL/2, aligning representation leads to FID=7.9 at 400K iteration, which already exceeds the FID of the vanilla SiT-XL at 7M iteration. Note that the performance continues to improve with longer training; for instance, with SiT-XL/2, FID becomes 6.4 at 1M iteration and 5.9 at 4M iteration. We also qualitatively compare the progression of generation results in Figure 4, where we use the same initial noise across different models. The model trained with REPA exhibits better progression.

Finally, we provide a quantitative comparison between SiT-XL/2 with REPA and other recent diffusion model methods using classifier-free guidance (Ho & Salimans, 2022). Our method already outperforms the original SiT-XL/2 with 7× fewer epochs and it is further improved with longer training. At 800 epochs, SiT-XL/2 with REPA achieves FID of 1.80 with a classifier-free guidance scale of $w = 1.35$, and achieves state-of-the-art FID of 1.42 with a extra classifier-free guidance scheduling with guidance interval (Kynkäänniemi et al., 2024). We provide selected qualitative results of SiT-XL/2 with REPA in Figure 6 and more examples in Appendix H. Moreover, we provide experimental results on ImageNet 512×512 and text-to-image generation in Appendix J and K; we show that REPA also provides significant improvements in such setups.

## 4.4 ABLATION STUDIES

**Representation gap across different timesteps.** We begin by comparing the semantic gap (measured through linear probing results) using outputs of the SiT models with different noise scale (*i.e.*, different timesteps), and maximum CKNNA values using clean DINOv2-g representations. As shown in Figure 7, REPA consistently reduces the representation gap across different noise levels, as indicated by better linear probing results and higher CKNNA values across all noise scales.

**Alignment to different visual encoders.** In addition, we extend the analysis from Section 2 to other visual encoders, not limited to the DINOv2 models. Specifically, we train SiT-L/2 models using REPA with MAE or MoCov3. As depicted in Figure 8, these models demonstrate higher CKNNA values across the corresponding target representations than the vanilla model. This indicates that REPA is effective in aligning various visual representations, not limited to DINOv2.

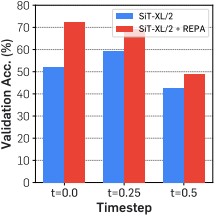 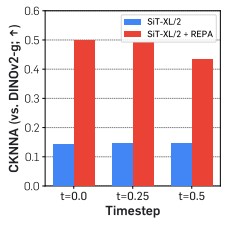 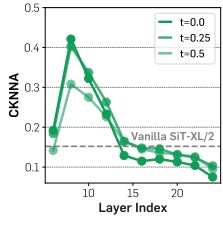 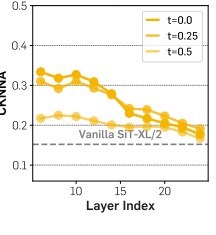

(a) Linear probing    (b) Alignment    (a) MoCov3-L    (b) MAE-L

Figure 7: **Representation gap across different timesteps.** We plot the linear probing results and maximum CKNNA values (using DINOv2-g) at different timesteps, comparing the vanilla SiT-XL/2 model and the same model trained using REPA. REPA consistently reduces the representation gap across different noise levels.

Figure 8: **Alignment to different target representations.** CKNNA values of SiT-L/2 models using REPA, with (a) MoCov3-L and (b) MAE-L as target representations. After that, we measure CKNNA using these encoders. REPA consistently improves the alignment regardless of the target representations.

**Effect of** $\lambda$. We also examine the effect of the regularization coefficient $\lambda$ by training SiT-XL/2 models for 400K with different coefficients 0.25 to 1.0 and comparing the performance. As shown in Table 5, the performance is robust to the values and it is quite saturated after $\lambda = 0.5$.

Table 5: Ablation study for $\lambda$.

| $\lambda$ | 0.25 | 0.5 | 0.75 | 1.0 |
|---|---|---|---|---|
| FID↓ | 8.6 | 7.9 | 7.8 | 7.8 |
| IS↑ | 118.6 | 122.6 | 124.4 | 124.8 |

## 5 RELATED WORK

We discuss with the most relevant literature here and provide a more discussion in Appendix I.

**Bridging diffusion models and representation learning.** Many recent works have attempted to exploit or improve representations learned from diffusion models (Fuest et al., 2024). First, there are hybrid model approaches: Yang et al. (2022) and Deja et al. (2023) train a single model capable of both classification and diffusion-based generation. Also, Tian et al. (2024) introduces a hybrid model capable of segmentation and generation. Next, several works have analyzed and exploited representations in diffusion models: Xiang et al. (2023) and Mukhopadhyay et al. (2021) observe that the intermediate representations of diffusion models have discriminative properties. Moreover, Repfusion (Yang & Wang, 2023) and DreamTeacher (Li et al., 2023b) propose knowledge distillation schemes using diffusion models to perform various downstream tasks. Our work also shares some similarities, where we focus on designing a regularization for *alignments* between recent self-supervised and diffusion representations and how they affect generation.

**Diffusion models with external representations.** Several recent studies have explored leveraging pretrained visual encoders to enhance efficiency and performance of diffusion models (Pernias et al., 2024; Li et al., 2024). Würstchen (Pernias et al., 2024) introduces a two-stage text-to-image diffusion model framework: a text-conditioned model that first generates a semantic map from a text prompt, followed by another diffusion model that synthesizes images based on the semantic map. RCG (Li et al., 2024) focuses on unconditional generation, where a compact 1D latent vector is produced by a diffusion model and subsequently used as a label for image generation by a second diffusion model. We also exploit pretrained representations for improving the diffusion model but without the need of training an additional model that learns the representation distribution.

## 6 CONCLUSION

In this paper, we have presented REPA, a simple regularization for improving diffusion transformers. In particular, we investigated whether diffusion transformer representations can be aligned with recent self-supervised representations, and if it can improve the generation performance of diffusion transformers. We showed REPA can significantly improves generation performance of diffusion transformers with faster convergence speed. We hope our work would facilitate many possible future research directions, including unifying discriminative and generative models and their representations or theoretical analysis; We provide more discussion in Appendix M.

## REPRODUCIBILITY STATEMENT

We provide hyperparameter details in Section 4 and Appendix D. We also release the implementation and model checkpoints to reproduce the results in the paper.

## ACKNOWLEDGMENT

This work was supported by Institute for Information & communications Technology Promotion(IITP) grant funded by the Korea government(MSIT) (No. RS-2019-II190075 Artificial Intelligence Graduate School Program (KAIST); No. RS-2024-00509279, Global AI Frontier Lab).

SY thanks the anonymous reviewers, Hankook Lee, Jihoon Tack, Sukmin Yun, and Yisol Choi for their insightful discussions; Myungkyu Koo and Daewon Choi for proofreading; and Kyungmin Lee for providing an excellent diffusion model implementation. SY also thanks Yewon Kim for the assistance in improving the visualizations in the paper.

JJ acknowledges support from the Institute of Information & communications Technology Planning & Evaluation (IITP) grant funded by the Korea government (MSIT) (No. RS-2019-II190079, Artificial Intelligence Graduate School Program (Korea University)), and the IITP-ITRC (Information Technology Research Center) grant funded by the Korea government (MSIT) (No. IITP-2025-RS-2024-00436857, 10%).

SX thanks discussions with Boyang Zheng and Willis Ma. SX also acknowledges support from the IITP grant funded by the Korean Government (MSIT) (No. RS-2024-00457882, National AI Research Lab Project), the Open Path AI Foundation, Amazon Research Award, Google TRC program, and NSF Award IIS-2443404.

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

# A    DESCRIPTIONS FOR DIFFUSION-BASED MODELS

We provide an overview of two types of generative models that we use in this paper, which learn the target distribution by training variants of a denoising autoencoder. We first explain denoising diffusion probabilistic models (DDPM) in Section A.1 and stochastic interpolants in Section A.2. For detailed explanations and rigorous proofs, please refer to the original papers (Albergo et al., 2023; Ma et al., 2024a) that provide excellent formulations and description.

## A.1    DENOISING DIFFUSION PROBABILISTIC MODELS

*Diffusion models* (Sohl-Dickstein et al., 2015; Ho et al., 2020) model the target distribution $p(\mathbf{x})$ via learning a gradual denoising process from Gaussian distribution $\mathcal{N}(\mathbf{0}, \mathbf{I})$ to $p(\mathbf{x})$. Formally, diffusion models learn a *reverse* process $p(\mathbf{x}_{t-1}|\mathbf{x}_t)$ of the pre-defined *forward* process $q(\mathbf{x}_t|\mathbf{x}_0)$ that gradually adds the Gaussian noise starting from $p(\mathbf{x})$ for $1 \leq t \leq T$ with a fixed $T > 0$.

For a given $\mathbf{x}_0 \sim p(\mathbf{x})$, $q(\mathbf{x}_t|\mathbf{x}_{t-1})$ can be formalized as $q(\mathbf{x}_t|\mathbf{x}_{t-1}) := \mathcal{N}(\mathbf{x}_t; \sqrt{1-\beta_t}\mathbf{x}_0, \beta_t^2 \mathbf{I})$, where $\beta_t \in (0, 1)$ are pre-defined hyperparameters set to be small. In particular, DDPM (Ho et al., 2020) shows if one formalizes the reverse process $p(\mathbf{x}_{t-1}|\mathbf{x}_t)$ (with $\alpha_t = 1 - \beta_t$. $\bar{\alpha}_t := \prod_{i=1}^{t} \alpha_i$ for $1 \leq t \leq T$) as

$$p(\mathbf{x}_{t-1}|\mathbf{x}_t) := \mathcal{N}\Big(\mathbf{x}_{t-1}; \frac{1}{\sqrt{\alpha_t}}\big(\mathbf{x}_t - \frac{\sigma_t^2}{\sqrt{1-\bar{\alpha}_t}}\boldsymbol{\epsilon_\theta}(\mathbf{x}_t, t)\big), \boldsymbol{\Sigma_\theta}(\mathbf{x}_t, t)\Big), \tag{10}$$

then $\boldsymbol{\epsilon_\theta}(\mathbf{x}_t, t)$ can be trained with a simple denoising autoencoder objective parameterized by $\boldsymbol{\theta}$:

$$\mathcal{L}_{\text{simple}} := \mathbb{E}_{\mathbf{x}_*, \boldsymbol{\epsilon}, t}\Big[||\boldsymbol{\epsilon} - \boldsymbol{\epsilon_\theta}(\mathbf{x}_t, t)||_2^2\Big]. \tag{11}$$

For $\boldsymbol{\Sigma_\theta}(\mathbf{x}_t, t)$, (Ho et al., 2020) shows it is enough to simply define it as $\sigma_t^2 \mathbf{I}$ with $\beta_t = \sigma_t^2$. After that, Nichol & Dhariwal (2021) exhibits the performance can be improved if the model jointly learns $\boldsymbol{\Sigma_\theta}(\mathbf{x}_t, t)$ with $\boldsymbol{\epsilon_\theta}(\mathbf{x}_t, t)$ in dimension-wise manner through the following objective:

$$\mathcal{L}_{\text{vlb}} := \exp(v \log \beta_t + (1-v) \log \tilde{\beta}_t), \tag{12}$$

where $v$ denotes each component per dimension from the model output and $\tilde{\beta}_t = \frac{1-\bar{\alpha}_{t-1}}{1-\bar{\alpha}_t}\beta_t$.

With a sufficiently large $T$ and an appropriate scheduling of $\beta_t$, the distribution $p(\mathbf{x}_T)$ becomes almost an isotropic Gaussian distribution. Hence, one can generate a sample starting from a random noise and perform iterative reverse process $p(\mathbf{x}_{t-1}|\mathbf{x}_t)$ to reach the data sample $\mathbf{x}_0$ (Ho et al., 2020).

## A.2   STOCHASTIC INTERPOLANTS

Different from DDPM, *flow-based models* (Esser et al., 2024; Lipman et al., 2022; Liu et al., 2023) deal with the continuous time-dependent process with a data $\mathbf{x}_* \sim p(\mathbf{x})$ and a Gaussian noise $\epsilon \sim \mathcal{N}(\mathbf{0}, \mathbf{I})$ on $t \in [0, 1]$:

$$\mathbf{x}_t = \alpha_t \mathbf{x}_0 + \sigma_t \epsilon, \quad \alpha_0 = \sigma_1 = 1, \; \alpha_1 = \sigma_0 = 0, \tag{13}$$

where $\alpha_t$ and $\sigma_t$ are a decreasing and increasing function of $t$ (respectively). There exists a *probability flow ordinary differential equation* (PF ODE) with a velocity field

$$\dot{\mathbf{x}}_t = \mathbf{v}(\mathbf{x}_t, t), \tag{14}$$

where distribution of this ODE at $t$ is equal to the marginal $p_t(\mathbf{x})$.

The velocity $\mathbf{v}(\mathbf{x}, t)$ is represented as the following sum of two conditional expectations

$$\mathbf{v}(\mathbf{x}, t) = \mathbb{E}[\dot{\mathbf{x}}_t | \mathbf{x}_t = \mathbf{x}] = \dot{\alpha}_t \mathbb{E}[\mathbf{x}_* | \mathbf{x}_t = \mathbf{x}] + \dot{\sigma}_t \mathbb{E}[\epsilon | \mathbf{x}_t = \mathbf{x}], \tag{15}$$

which can be approximated with model $\mathbf{v}_\theta(\mathbf{x}_t, t)$ by minimizing the following training objective:

$$\mathcal{L}_{\text{velocity}}(\theta) := \mathbb{E}_{\mathbf{x}_*, \epsilon, t} \left[ ||\mathbf{v}_\theta(\mathbf{x}_t, t) - \dot{\alpha}_t \mathbf{x}_* - \dot{\sigma}_t \epsilon||^2 \right]. \tag{16}$$

Note that this also corresponds to the following reverse *stochastic differential equation* (SDE):

$$d\mathbf{x}_t = \mathbf{v}(\mathbf{x}_t, t)dt - \frac{1}{2} w_t \mathbf{s}(\mathbf{x}_t, t)dt + \sqrt{w_t} d\bar{\mathbf{w}}_t, \tag{17}$$

where the score $\mathbf{s}(\mathbf{x}_t, t)$ similarly becomes the conditional expectation

$$\mathbf{s}(\mathbf{x}_t, t) = -\frac{1}{\sigma_t} \mathbb{E}[\epsilon | \mathbf{x}_t = \mathbf{x}]. \tag{18}$$

Similar to $\mathbf{v}$, $\mathbf{s}$ can be approximated with a model $\mathbf{s}_\theta(\mathbf{x}, t)$ with the following objective:

$$\mathcal{L}_{\text{score}}(\theta) := \mathbb{E}_{\mathbf{x}_*, \epsilon, t} \left[ ||\sigma_t \mathbf{s}_\theta(\mathbf{x}_t, t) + \epsilon||^2 \right]. \tag{19}$$

Here, since the score $\mathbf{s}(\mathbf{x}, t)$ can be directly computed using the velocity $\mathbf{v}(\mathbf{x}, t)$ for $t > 0$ as

$$\mathbf{s}(\mathbf{x}, t) = \frac{1}{\sigma_t} \cdot \frac{\alpha_t \mathbf{v}(\mathbf{x}, t) - \dot{\alpha}_t \mathbf{x}}{\dot{\alpha}_t \sigma_t - \alpha_t \dot{\sigma}_t}, \tag{20}$$

so it is enough to estimate only one of the two vectors.

*Stochastic interpolants* (Albergo et al., 2023) shows any $\alpha_t$ and $\sigma_t$ satisfy the three conditions

1. $\alpha_t^2 + \sigma_t^2 > 0, \; \forall t \in [0, 1]$
2. $\alpha_t$ and $\sigma_t$ are differentiable, $\forall t \in [0, 1]$
3. $\alpha_1 = \sigma_0 = 0, \; \alpha_0 = \sigma_1 = 1,$

leads to a process that interpolates between $\mathbf{x}_0$ and $\epsilon$ without bias. Thus, one can use a simple interpolant by defining them as a simple function during training and inference, such as linear interpolants with $\alpha_t = 1 - t$ and $\sigma_t = t$ or variance-preserving (VP) interpolants with $\alpha_t = \cos(\frac{\pi}{2} t)$ and $\sigma_t = \cos(\frac{\pi}{2} t)$ (Ma et al., 2024a). One another advantage of stochastic interpolants is that the diffusion coefficient $w_t$ is independent in training any of a score or a velocity model. Thus, $w_t$ can be also explicitly chosen *after training* when sampling with the reverse SDE.

Note that existing score-based diffusion models, including DDPM (Ho et al., 2020), can be similarly interpreted as an SDE formulation. In particular, their forward diffusion process can be interpreted as a pre-defined (discretized) forward SDEs that have an equilibrium distribution as $\mathcal{N}(\mathbf{0}, \mathbf{I})$ at $t \to \infty$, where the training is done on $[0, T]$ with sufficiently large $T$ (*e.g.*, $T = 1000$) that $p(\mathbf{x}_T)$ becomes almost isotropic Gaussian. Generation is done by solving the corresponding reverse SDE starting from a random Gaussian noise by assuming $\mathbf{x}_T \sim \mathcal{N}(\mathbf{0}, \mathbf{I})$, where $\alpha_t, \sigma_t$ and the diffusion coefficient $w_t$ is *implicitly* chosen from the forward diffusion process, which might lead to over-complicated design space of score-based diffusion models (Karras et al., 2022).

## B  DIFFUSION TRANSFORMER ARCHITECTURE

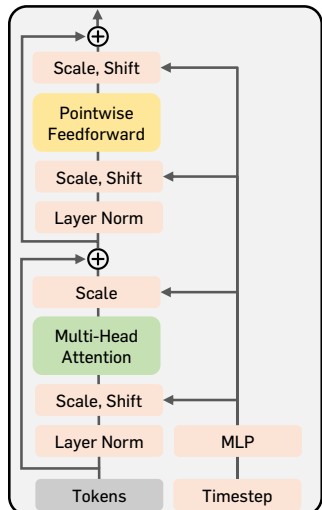

Figure 9: DiT block illustration.

We strictly follow the architecture used in DiT (Peebles & Xie, 2023) and SiT (Ma et al., 2024a). The architecture is very similar to a vision transformer (ViTs; Dosovitskiy et al. 2021): an input is patchified, reshaped to a 1D sequence of patches with a length $N$, and then fed to the model. Similar to DiT and SiT, our architecture also uses a downsampled latent image $\mathbf{z} = E(\mathbf{x})$ as an input, where $\mathbf{x}$ is a RGB image and $E$ is an encoder of the stable diffusion variational autoencoder (VAE) (Rombach et al., 2022). Different from the original ViT, our architecture also includes additional modulation layers at each attention block called AdaIN-zero layers. These layers scale and shift each hidden state with respect to the given timestep and additional conditions. We also consider a single multilayer perceptron (MLP) that projects a hidden state to the target representation space, which is only used in training. We provide an illustration of the DiT block in Figure 9.

## C  ANALYSIS DETAILS

### C.1  EVALUATION DETAILS

**CKNNA** (Centered Kernel Nearest-Neighbor Alignment) is a *relaxed version* of the popular Centered Kernel Alignment (CKA; Kornblith et al. 2019) that mitigates the strict definition of alignment. We generally follow the notations in the original paper for an explanation (Huh et al., 2024).

First, CKA have measured *global* similarities of the models by considering all possible data pairs:

$$\text{CKA}(\mathbf{K}, \mathbf{L}) = \frac{\text{HSIC}(\mathbf{K}, \mathbf{L})}{\sqrt{\text{HSIC}(\mathbf{K}, \mathbf{K})\text{HSIC}(\mathbf{L}, \mathbf{L})}}, \qquad (21)$$

where $\mathbf{K}$ and $\mathbf{L}$ are two kernel matrices computed from the dataset using two different networks. Specifically, it is defined as $\mathbf{K}_{ij} = \kappa(\phi_i, \phi_j)$ and $\mathbf{L}_{ij} = \kappa(\psi_i, \psi_j)$ where $\phi_i, \phi_j$ and $\psi_i, \psi_j$ are representations computed from each network at the corresponding data $\mathbf{x}_i, \mathbf{x}_j$ (respectively). By letting $\kappa$ as a inner product kernel, HSIC is defined as

$$\text{HSIC}(\mathbf{K}, \mathbf{L}) = \frac{1}{(n-1)^2} \Big( \sum_i \sum_j \big( \langle \phi_i, \phi_j \rangle - \mathbb{E}_l[\langle \phi_i, \phi_l \rangle] \big) \big( \langle \psi_i, \psi_j \rangle - \mathbb{E}_l[\langle \psi_i, \psi_l \rangle] \big) \Big). \quad (22)$$

CKNNA considers a relaxed version of Eq. (21) by replacing $\text{HSIC}(\mathbf{K}, \mathbf{L})$ into $\text{Align}(\mathbf{K}, \mathbf{L})$, where $\text{Align}(\mathbf{K}, \mathbf{L})$ computes Eq. (22) only using a $k$-nearest neighborhood embedding in the datasets:

$$\text{Align}(\mathbf{K}, \mathbf{L}) = \frac{1}{(n-1)^2} \Big( \sum_i \sum_j \alpha(i, j) \big( \langle \phi_i, \phi_j \rangle - \mathbb{E}_l[\langle \phi_i, \phi_l \rangle] \big) \big( \langle \psi_i, \psi_j \rangle - \mathbb{E}_l[\langle \psi_i, \psi_l \rangle] \big) \Big),$$

$$(23)$$

where $\alpha(i, j)$ is defined as

$$\alpha(i, j; k) = \mathbb{1}[i \neq j \text{ and } \phi_j \in \text{knn}(\phi_i; k) \text{ and } \psi_j \in \text{knn}(\psi_i; k)], \quad (24)$$

so this term only considers $k$-nearest neighbors at each $i$. In this paper, we randomly sample 10,000 images in the validation set in ImageNet (Deng et al., 2009) and report CKNNA with $k = 10$ based on observation in Huh et al. (2024) that smaller $k$ shows better a better alignment.

**Linear probing.** We follow the setup used in DAE (Chen et al., 2024c). Specifically, we use parameter-free batch normalization layer and train a linear layer for 90 epochs with a batch size of 16,384. We use the Adam optimizer (Kingma, 2015) with cosine decay learning rate scheduler, where the initial learning rate is set to 0.001.

## C.2 DiT Analysis

We also perform a similar analysis have done in Figure 2a (linear probing) and 2b (CKA), and illustrate the result in Figure 10. Overall is shows a similar trend; the model includes discriminative representation but the gap is still large compared with DINOv2, as shown in the linear probing results, and also weakly aligned with DINOv2 representations.

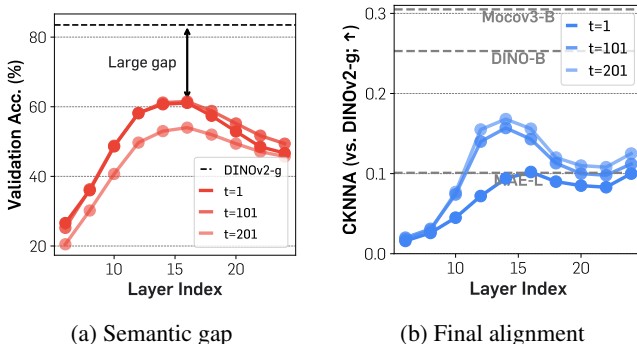

(a) Semantic gap                    (b) Final alignment

Figure 10: **Empirical study with the pretrained DiT model.** Similar to Figure 2, we compare the semantic gap and measure the feature alignment between DINOv2-g and the DiT-XL/2 model trained with 7M iterations. (a) DiT learns meaningful (discriminative) representation but it still have a large gap between DINOv2. (b) Measured with CKNNA (Huh et al., 2024), DiT already shows a weak alignment with DINOv2, but its absolute value is still small.

## C.3 Description of pretrained visual encoders

- **MAE** (He et al., 2022) proposes a self-supervised representation learning objective for vision transformers, based on the reconstruction task of masked patches of input images.

- **DINO** (Caron et al., 2021) is a self-supervised learning method based on self-distillation through the mean of momentum teacher network.

- **MoCov3** (Chen et al., 2021) studies empirical study to train MoCo (He et al., 2020; Chen et al., 2020b) on vision transformer and how they can be scaled up.

- **CLIP** (Radford et al., 2021) proposes a contrastive learning scheme on large image-text pairs.

- **DINOv2** (Oquab et al., 2024) proposes a self-supervised learning method that combines pixel-level and patch-level discriminative objectives by leveraging advanced self-supervised techniques and a large pre-training dataset.

- **I-JEPA** (Assran et al., 2023) predicts missing parts of an image by learning representations through joint-embedding, focusing on the context of the entire image without relying on pixel-level reconstruction.

- **SigLIP** (Zhai et al., 2023) replaces the traditional softmax loss with a pairwise sigmoid loss, enhancing performance and efficiency of image-text representation learning.

Moreover, in Table 6, we also provide the datasets used for training of each of pretrained visual encoder. As shown in this table, better visual representations learned from massive amounts of image data provide more improvement, regardless of whether the dataset does not include ImageNet. Note that we do not fine-tune encoders (*e.g.*, SigLIP and CLIP) with the ImageNet dataset, particularly if they were trained with another dataset, thereby separating the dataset leakage effect if we use these encoders. REPA also achieves significant improvements with these encoders, which validates that the improvement does not simply come from data leakages.

Table 6: Dataset analysis used for pretrained visual encoders.

| Method | Dataset | w/ ImageNet-1K | Text sup. | FID↓ |
|--------|---------|----------------|-----------|------|
| MAE | ImageNet-1K | O | X | 12.5 |
| DINO | ImageNet-1K | O | X | 11.9 |
| MoCov3 | ImageNet-1K | O | X | 11.9 |
| I-JEPA | ImageNet-1K | O | X | 11.6 |
| CLIP | WIT-400M | X | O | 11.0 |
| SigLIP | WebLi-4B | X | O | 10.2 |
| DINOv2 | LVD-142M | O | X | 10.0 |

# D  HYPERPARAMETER AND MORE IMPLEMENTATION DETAILS

Table 7: Hyperparameter setup.

|  | Figure 3 | Table 3 (SiT-B) | Table 3 (SiT-L) | Table 3 (SiT-XL) | Table 4 |
|---|---|---|---|---|---|
| **Architecture** | | | | | |
| Input dim. | 32×32×4 | 32×32×4 | 32×32×4 | 32×32×4 | 32×32×4 |
| Num. layers | 28 | 12 | 24 | 28 | 28 |
| Hidden dim. | 1,152 | 768 | 1,024 | 1,152 | 1,152 |
| Num. heads | 16 | 12 | 16 | 16 | 16 |
| **REPA** | | | | | |
| $\lambda$ | 0.5 | 0.5 | 0.5 | 0.5 | 0.5 |
| Alignment depth | 8 | 4 | 8 | 8 | 8 |
| $\text{sim}(\cdot, \cdot)$ | cos. sim. | cos. sim. | NT-Xent | cos. sim. | cos. sim. |
| Encoder $f(\mathbf{x})$ | DINOv2-B | DINOv2-B | DINOv2-L | DINOv2-B | DINOv2-B |
| **Optimization** | | | | | |
| Training iteration | 1M | 400K | 700K | 4M | 4M |
| Batch size | 256 | 256 | 256 | 256 | 256 |
| Optimizer | AdamW | AdamW | AdamW | AdamW | AdamW |
| lr | 0.0001 | 0.0001 | 0.0001 | 0.0001 | 0.0001 |
| $(\beta_1, \beta_2)$ | (0.9, 0.999) | (0.9, 0.999) | (0.9, 0.999) | (0.9, 0.999) | (0.9, 0.999) |
| **Interpolants** | | | | | |
| $\alpha_t$ | $1-t$ | $1-t$ | $1-t$ | $1-t$ | $1-t$ |
| $\sigma_t$ | $t$ | $t$ | $t$ | $t$ | $t$ |
| $w_t$ | $\sigma_t$ | $\sigma_t$ | $\sigma_t$ | $\sigma_t$ | $\sigma_t$ |
| Training objective | v-prediction | v-prediction | v-prediction | v-prediction | v-prediction |
| Sampler | Euler-Maruyama | Euler-Maruyama | Euler-Maruyama | Euler-Maruyama | Euler-Maruyama |
| Sampling steps | 250 | 250 | 250 | 250 | 250 |
| Guidance | - | - | - | - | 1.35 |

**Further implementation details.** We implement our model based on the original SiT implementation (Ma et al., 2024a). Throughout the experiments, we use the exact same structure as DiT (Peebles & Xie, 2023) and SiT (Ma et al., 2024a). We use AdamW (Kingma, 2015; Loshchilov, 2017) with constant learning rate of 1e-4, $(\beta_1, \beta_2) = (0.9, 0.999)$ without weight decay. To speed up training, we use mixed-precision (fp16) with a gradient clipping. We also pre-compute compressed latent vectors from raw pixels via stable diffusion VAE (Rombach et al., 2022) and use these latent vectors. Because of this, we do not apply any data augmentation, but we find this does not lead to a big difference, as similarly observed in EDM2 (Karras et al., 2024). We also use `stabilityai/sd-vae-ft-ema` decoder for decoding latent vectors to images. For MLP used for a projection, we use three-layer MLP with SiLU activations (Elfwing et al., 2018). We provide a detailed hyperparameter setup in Table 7.

**Pretrained encoders.** For MoCov3-B and -L models, we use the checkpoint in the implementation of RCG (Li et al., 2024);[2] for other checkpoints, we use their official checkpoints released in their official implementations. To adjust a different number of patches between the diffusion transformer and the pretrained encoder, we interpolate positional embeddings of pretrained encoders.

**Sampler.** For sampling, we use the Euler-Maruyama sampler with the SDE in Eq. (5) with a diffusion coefficient $w_t = \sigma_t$. We use the last step of the SDE sampler as 0.04, and it gives a significant improvement, similar to the original SiT paper (Ma et al., 2024a).

**Computing resources.** We use 8 NVIDIA H100 80GB GPUs for experiments; our training speed is about 5.4 step/s with a batch size of 256. Note that this can be further boosted with additional engineering (*e.g.*, pre-computation of pretrained encoder features).

---

[2] https://github.com/LTH14/rcg

# E  EVALUATION DETAILS

We strictly follow the setup and use the same reference batches of ADM (Dhariwal & Nichol, 2021) for evaluation, following their official implementation.[3] We use NVIDIA H100 80GB GPUs or 4090Ti GPUs for evaluation and enable tf32 precision for faster generation, and we find the performance difference is negligible to the original fp32 precision.

In what follows, we explain the main concept of metrics that we used for the evaluation.

- **FID** (Heusel et al., 2017) measures the feature distance between the distributions of real and generated images. It uses the Inception-v3 network (Szegedy et al., 2016) and computes distance based on an assumption that both feature distributions are multivariate gaussian distributions.
- **sFID** (Nash et al., 2021) proposes to compute FID with intermediate spatial features of the Inception-v3 network to capture the generated images' spatial distribution.
- **IS** (Salimans et al., 2016) also uses the Inception-v3 network but use logit for evaluation of the metric. Specifically, it measures a KL-divergence between the original label distribution and the distribution of logits after the softmax normalization.
- **Precision and recall** (Kynkäänniemi et al., 2019) are based on their classic definitions: the fraction of realistic images and the fraction of training data manifold covered by generated data.

# F  BASELINES

In what follows, we explain the main idea of baseline methods that we used for the evaluation.

- **ADM** (Dhariwal & Nichol, 2021) improves U-Net-based architectures for diffusion models and proposes classifier-guided sampling to balance the quality and diversity tradeoff.
- **VDM++** (Kingma & Gao, 2024) proposes a simple adaptive noise schedule for diffusion models to improve training efficiency.
- **Simple diffusion** (Hoogeboom et al., 2023) proposes a diffusion model for high-resolution image generation by exploring various techniques to simplify a noise schedule and architectures.
- **CDM** (Ho et al., 2022) introduces cascaded diffusion models: similar to progressiveGAN (Karras et al., 2018), it trains multiple diffusion models starting from the lowest resolution and applying one or more super-resolution diffusion models for generating high-fidelity images.
- **LDM** (Rombach et al., 2022) proposes latent diffusion models by modeling image distribution in a compressed latent space to improve the training efficiency without sacrificing the generation performance.
- **U-ViT** (Bao et al., 2023) proposes a ViT-based latent diffusion model that incorporates U-Net-like long skip connections.
- **DiffiT** (Hatamizadeh et al., 2024) proposes a time-dependent multi-head self-attention mechanism for enhancing the efficiency of transformer-based image diffusion models.
- **MDTv2** (Gao et al., 2023) proposes an asymmetric encoder-decoder scheme for efficient training of a diffusion-based transformer. They also apply U-Net-like long-shortcuts in the encoder and dense input-shortcuts in the decoder.
- **MaskDiT** (Zheng et al., 2024) proposes an asymmetric encoder-decoder scheme for efficient training of diffusion transformers, where they train the model with an auxiliary mask reconstruction task similar to MAE (He et al., 2022).
- **SD-DiT** (Zhu et al., 2024) extends MaskdiT architecture but incorporates self-supervised discrimination objective using a momentum encoder.
- **DiT** (Peebles & Xie, 2023) proposes a pure transformer backbone for training diffusion models based on proposing AdaIN-zero modules.
- **SiT** (Ma et al., 2024a) extensively analyzes how DiT training can be efficient by moving from discrete diffusion to continuous flow-based modeling.

---

[3]https://github.com/openai/guided-diffusion/tree/main/evaluations

# G    DETAILED QUANTITATIVE RESULTS

We provide evaluation results of different SiT models trained with REPA. All models are aligned with DINOv2-B representations with $\lambda = 0.5$ and negative cosine similarity. We use the 4th layer hidden states for the base model and use the 8th layer hidden states for the large and xlarge model.

Table 8: Detailed evaluation results with different model sizes. All results are reported without classifier-free guidance.

| Model | #Params | Iter. | FID↓ | sFID↓ | IS↑ | Prec.↑ | Rec.↑ | Acc.↑ |
|---|---|---|---|---|---|---|---|---|
| SiT-B/2 (Ma et al., 2024a) | 130M | 400K | 33.0 | 6.46 | 43.7 | 0.53 | 0.63 | N/A |
| + REPA (ours) | 130M | 50K | 78.2 | 11.71 | 17.1 | 0.33 | 0.48 | 43.2 |
| + REPA (ours) | 130M | 100K | 49.5 | 7.00 | 27.5 | 0.46 | 0.59 | 50.9 |
| + REPA (ours) | 130M | 200K | 33.2 | 6.68 | 43.7 | 0.54 | 0.63 | 50.9 |
| + REPA (ours) | 130M | 400K | 24.4 | 6.40 | 59.9 | 0.59 | 0.65 | 61.2 |
| SiT-L/2 (Ma et al., 2024a) | 458M | 400K | 18.8 | 5.29 | 72.0 | 0.64 | 0.64 | N/A |
| + REPA (ours) | 458M | 50K | 55.4 | 24.0 | 23.0 | 0.43 | 0.53 | 55.3 |
| + REPA (ours) | 458M | 100K | 24.1 | 6.25 | 55.7 | 0.62 | 0.60 | 61.8 |
| + REPA (ours) | 458M | 200K | 14.0 | 5.18 | 86.5 | 0.67 | 0.64 | 66.3 |
| + REPA (ours) | 458M | 400K | 10.0 | 5.20 | 109.2 | 0.69 | 0.65 | 69.4 |
| SiT-XL/2 (Ma et al., 2024a) | 675M | 7M | 8.3 | 6.32 | 131.7 | 0.68 | 0.67 | N/A |
| + REPA (ours) | 675M | 50K | 52.3 | 31.24 | 24.3 | 0.45 | 0.53 | 56.1 |
| + REPA (ours) | 675M | 100K | 19.4 | 6.06 | 67.4 | 0.64 | 0.61 | 62.9 |
| + REPA (ours) | 675M | 200K | 11.1 | 5.05 | 100.4 | 0.69 | 0.64 | 67.3 |
| + REPA (ours) | 675M | 400K | 7.9 | 5.06 | 122.6 | 0.70 | 0.65 | 70.3 |
| + REPA (ours) | 675M | 4M | 5.9 | 5.73 | 157.8 | 0.70 | 0.69 | 74.6 |

We also provide SiT-XL/2+REPA at 4M iteration with classifier-free guidance with different classifier-free guidance scales.

Table 9: Detailed evaluation results of SiT-XL+REPA at 4M iteration with different classifier-free guidance scale $w$.

| Model | #Params | Iter. | $w$ | FID↓ | sFID↓ | IS↑ | Prec.↑ | Rec.↑ |
|---|---|---|---|---|---|---|---|---|
| SiT-XL/2 (Ma et al., 2024a) | 675M | 7M | 1.500 | 2.06 | 4.50 | 270.3 | 0.82 | 0.59 |
| + REPA (ours) | 675M | 4M | 1.300 | 1.80 | 4.55 | 268.6 | 0.80 | 0.63 |
| + REPA (ours) | 675M | 4M | 1.325 | 1.79 | 4.51 | 276.8 | 0.81 | 0.62 |
| + REPA (ours) | 675M | 4M | 1.350 | 1.80 | 4.50 | 284.0 | 0.81 | 0.61 |
| + REPA (ours) | 675M | 4M | 1.375 | 1.84 | 4.48 | 291.7 | 0.82 | 0.61 |
| + REPA (ours) | 675M | 4M | 1.400 | 1.90 | 4.48 | 297.5 | 0.82 | 0.60 |

Moreover, we provide the results with the guidance interval (Kynkäänniemi et al., 2024).

| Model | #Params | Iter. | Interval | $w$ | FID↓ | sFID↓ | IS↑ | Prec.↑ | Rec.↑ |
|---|---|---|---|---|---|---|---|---|---|
| SiT-XL/2 (Ma et al., 2024a) | 675M | 7M | [0, 1] | 1.50 | 2.06 | 4.50 | 270.3 | 0.82 | 0.59 |
| + REPA (ours) | 675M | 4M | [0, 0.8] | 2.00 | 2.23 | 4.40 | 360.9 | 0.84 | 0.6 |
| + REPA (ours) | 675M | 4M | [0, 0.75] | 2.00 | 1.78 | 4.50 | 346.2 | 0.82 | 0.62 |
| + REPA (ours) | 675M | 4M | [0, 0.7] | 2.00 | 1.48 | 4.67 | 324.0 | 0.82 | 0.62 |
| + REPA (ours) | 675M | 4M | [0, 0.65] | 2.00 | 1.44 | 4.88 | 308.8 | 0.79 | 0.65 |
| + REPA (ours) | 675M | 4M | [0, 0.6] | 2.00 | 1.56 | 5.11 | 290.7 | 0.78 | 0.66 |
| + REPA (ours) | 675M | 4M | [0, 0.7] | 1.90 | 1.45 | 4.68 | 317.6 | 0.80 | 0.64 |
| + REPA (ours) | 675M | 4M | [0, 0.7] | 1.80 | 1.42 | 4.70 | 305.7 | 0.80 | 0.64 |

Table 10: Detailed evaluation results of SiT-XL+REPA at 4M iteration with different classifier-free guidance scale $w$. We apply the guidance interval (Kynkäänniemi et al., 2024).

Finally, in Figure 11, we provide a plot similar to Figure 1 using SiT-XL/2 but FIDs are obtained with a classifier-guidance scale $w = 1.35$. Similar to Figure 1, REPA provides great speedup and performance improvement compared with the vanilla model.

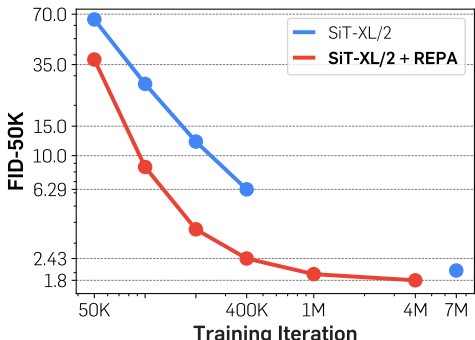

Figure 11: **Training iteration vs. FID plot.** All values are measured using a classifier-free guidance. REPA demonstrates a notable speedup and enhanced performance.

## H    MORE QUALITATIVE RESULTS

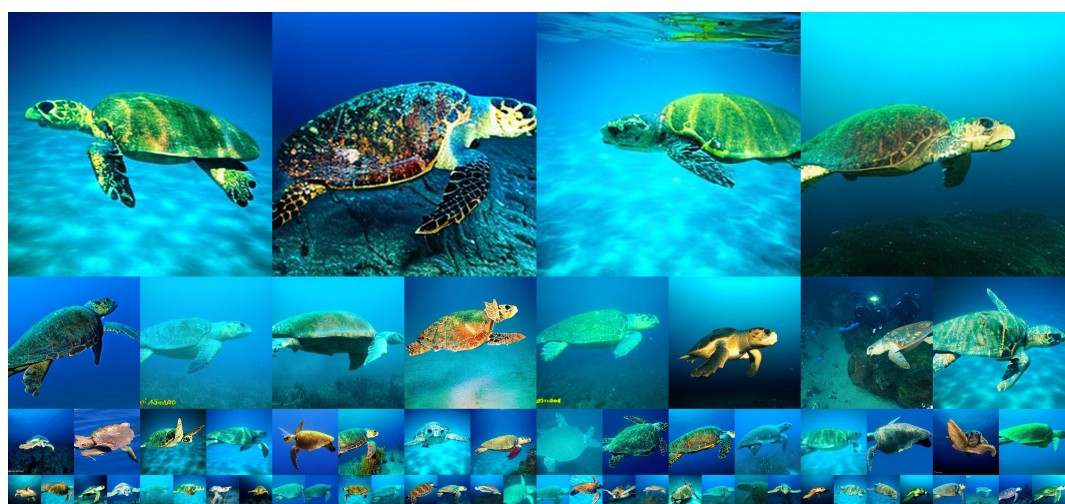

Figure 12: **Uncurated generation results of SiT-XL/2 + REPA.** We use classifier-free guidance with $w = 4.0$. Class label = "loggerhead sea turtle" (33).

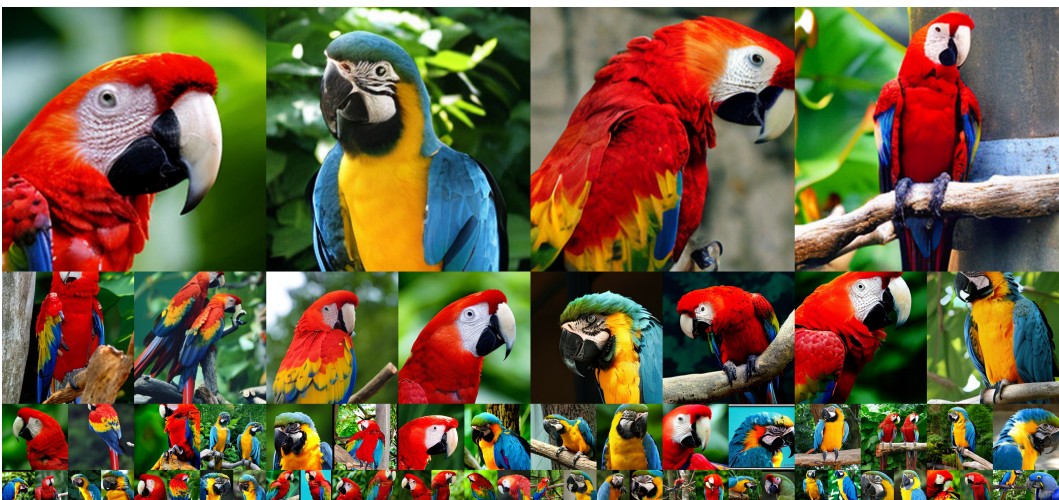

Figure 13: **Uncurated generation results of SiT-XL/2 + REPA.** We use classifier-free guidance with $w = 4.0$. Class label = "macaw" (88).

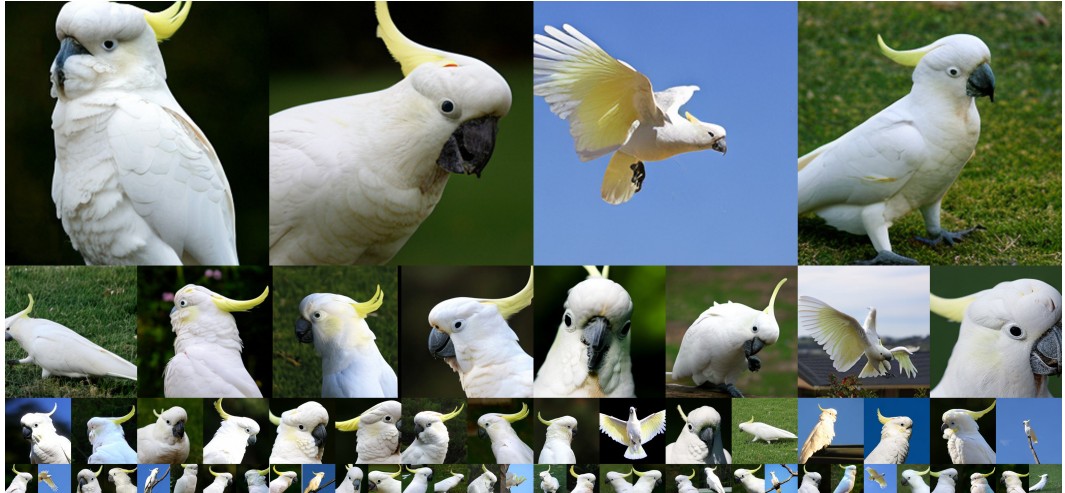

Figure 14: **Uncurated generation results of SiT-XL/2 + REPA.** We use classifier-free guidance with $w = 4.0$. Class label = "sulphur-crested cockatoo" (89).

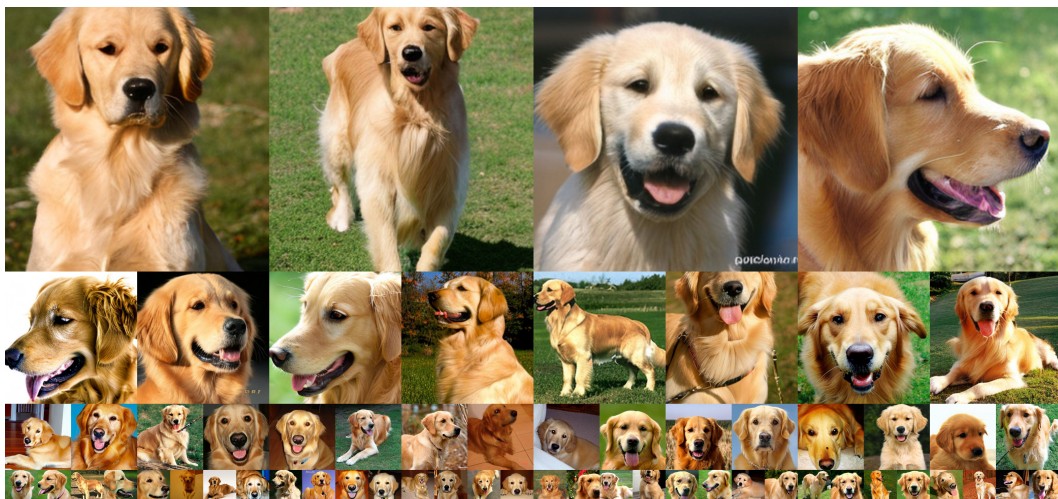

Figure 15: **Uncurated generation results of SiT-XL/2 + REPA.** We use classifier-free guidance with $w = 4.0$. Class label = "golden retriever" (207).

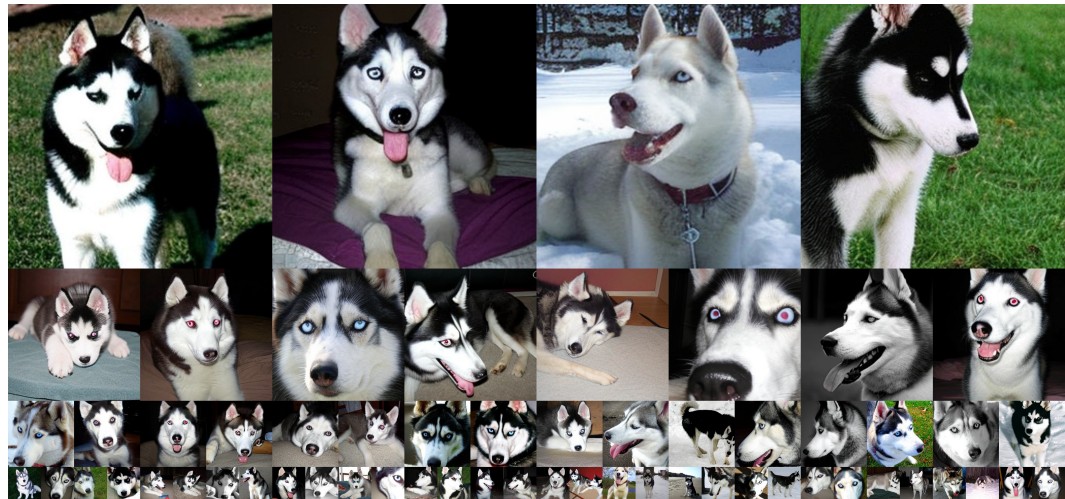

Figure 16: **Uncurated generation results of SiT-XL/2 + REPA.** We use classifier-free guidance with $w = 4.0$. Class label = "husky" (250).

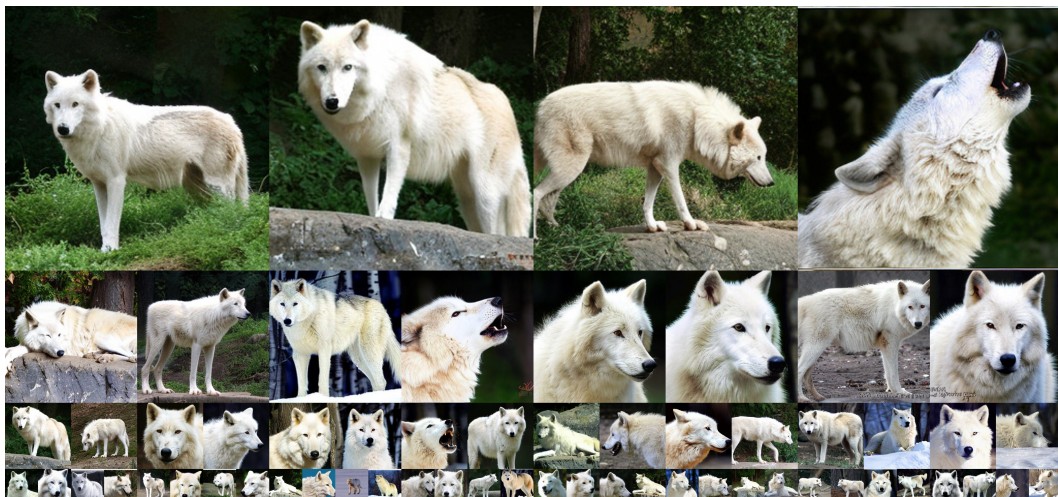

Figure 17: **Uncurated generation results of SiT-XL/2 + REPA.** We use classifier-free guidance with $w = 4.0$. Class label = "arctic wolf" (270).

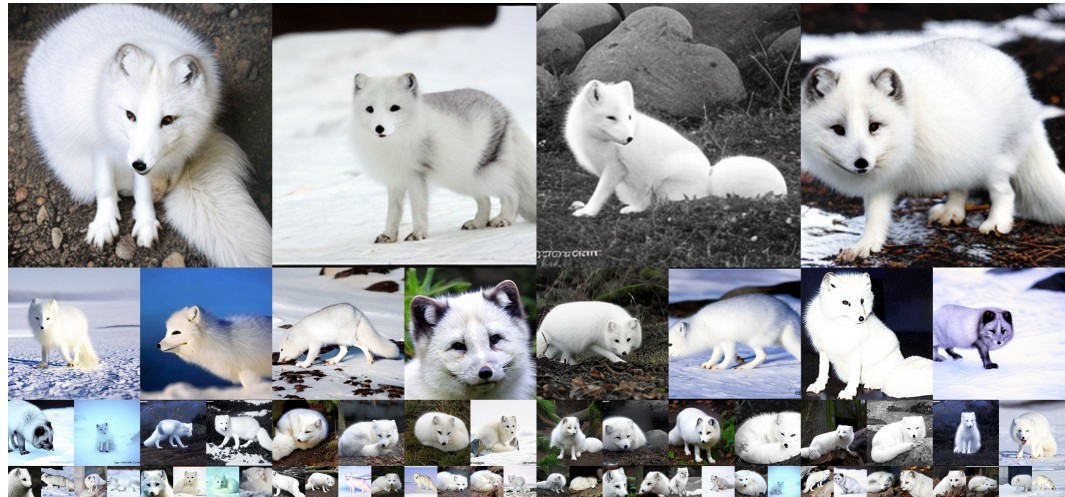

Figure 18: **Uncurated generation results of SiT-XL/2 + REPA.** We use classifier-free guidance with $w = 4.0$. Class label = "arctic fox" (279).

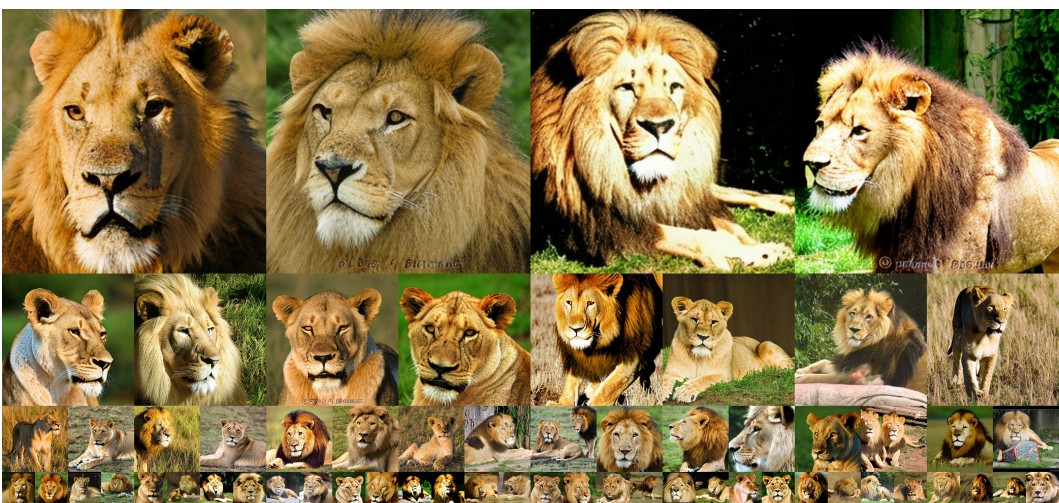

Figure 19: **Uncurated generation results of SiT-XL/2 + REPA.** We use classifier-free guidance with $w = 4.0$. Class label = "lion" (291).

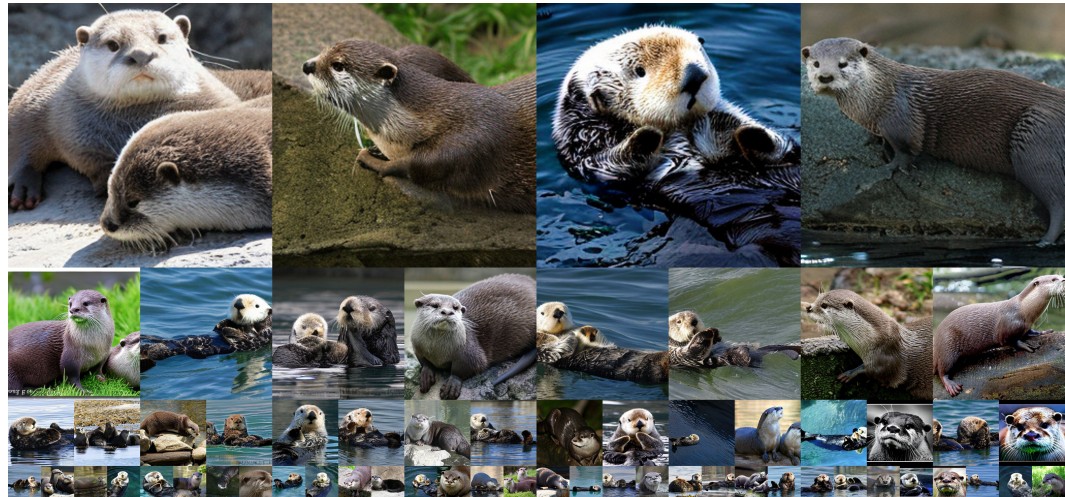

Figure 20: **Uncurated generation results of SiT-XL/2 + REPA.** We use classifier-free guidance with $w = 4.0$. Class label = "otter" (360).

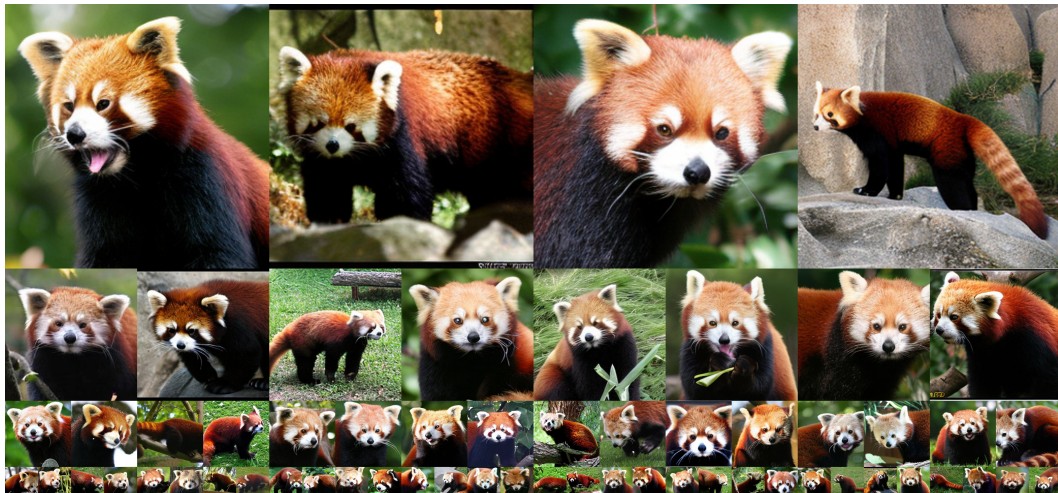

Figure 21: **Uncurated generation results of SiT-XL/2 + REPA.** We use classifier-free guidance with $w = 4.0$. Class label = "red panda" (387).

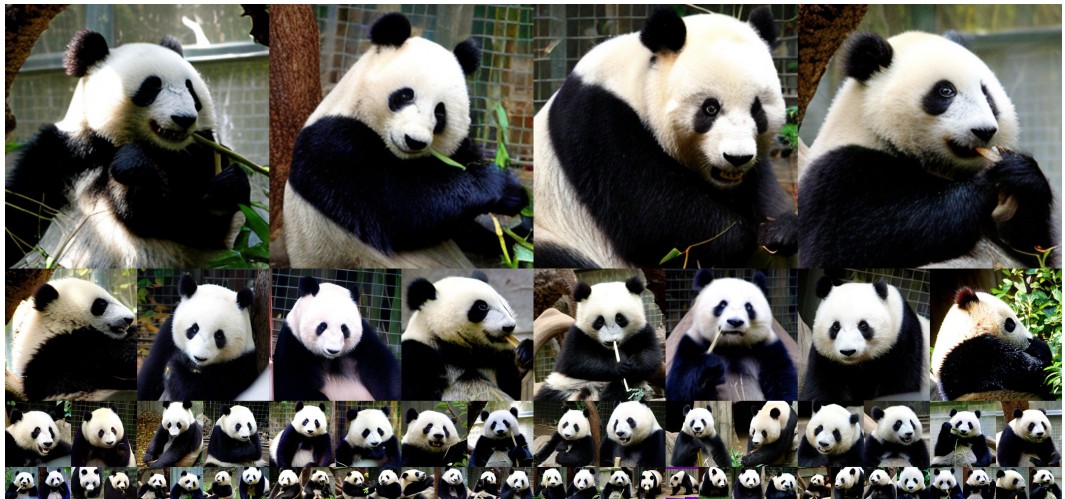

Figure 22: **Uncurated generation results of SiT-XL/2 + REPA.** We use classifier-free guidance with $w = 4.0$. Class label = "panda" (388).

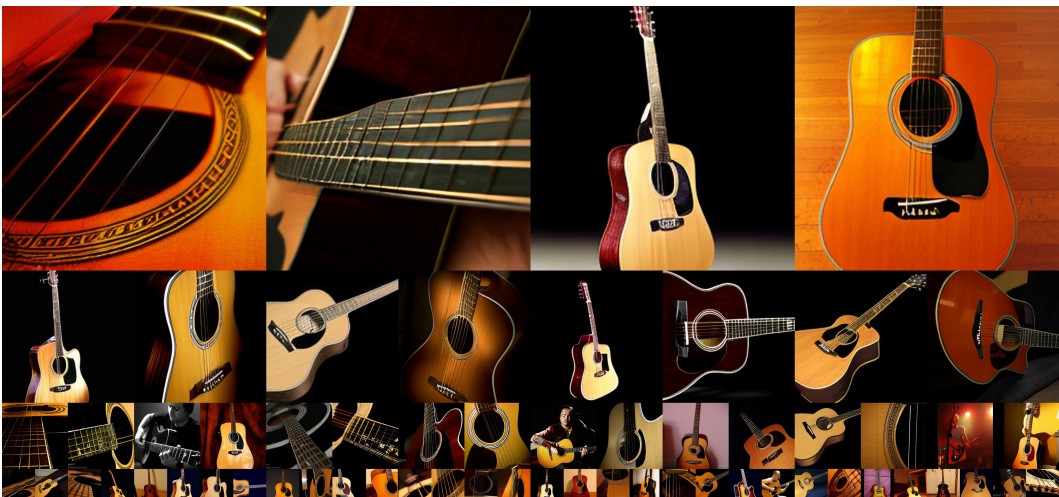

Figure 23: **Uncurated generation results of SiT-XL/2 + REPA.** We use classifier-free guidance with $w = 4.0$. Class label = "acoustic guitar" (402).

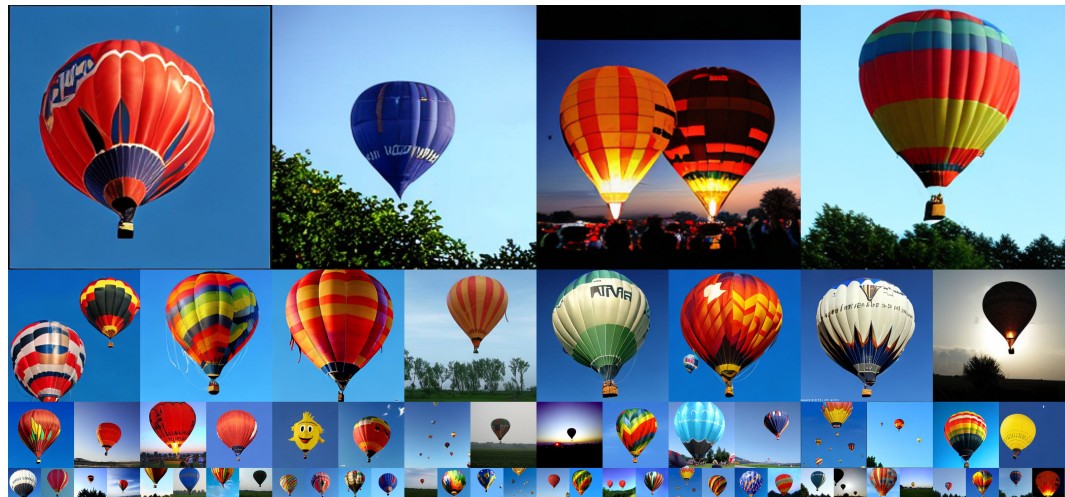

Figure 24: **Uncurated generation results of SiT-XL/2 + REPA.** We use classifier-free guidance with $w = 4.0$. Class label = "balloon" (417).

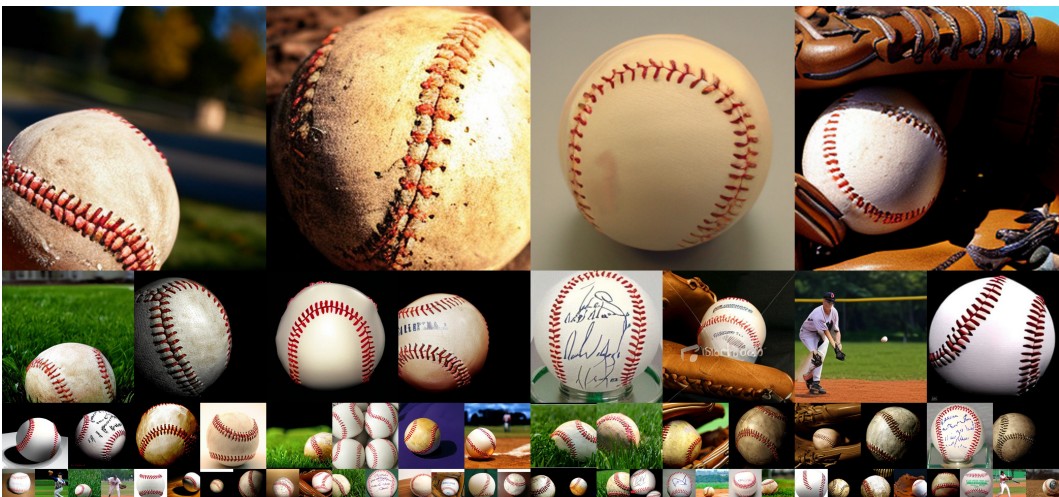

Figure 25: **Uncurated generation results of SiT-XL/2 + REPA.** We use classifier-free guidance with $w = 4.0$. Class label = "baseball" (429).

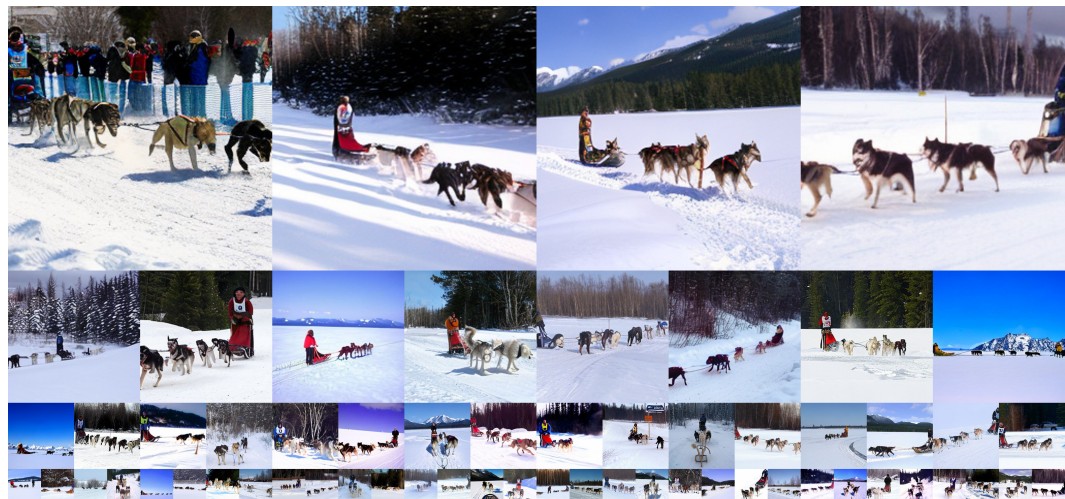

Figure 26: **Uncurated generation results of SiT-XL/2 + REPA.** We use classifier-free guidance with $w = 4.0$. Class label = "dog sled" (537).

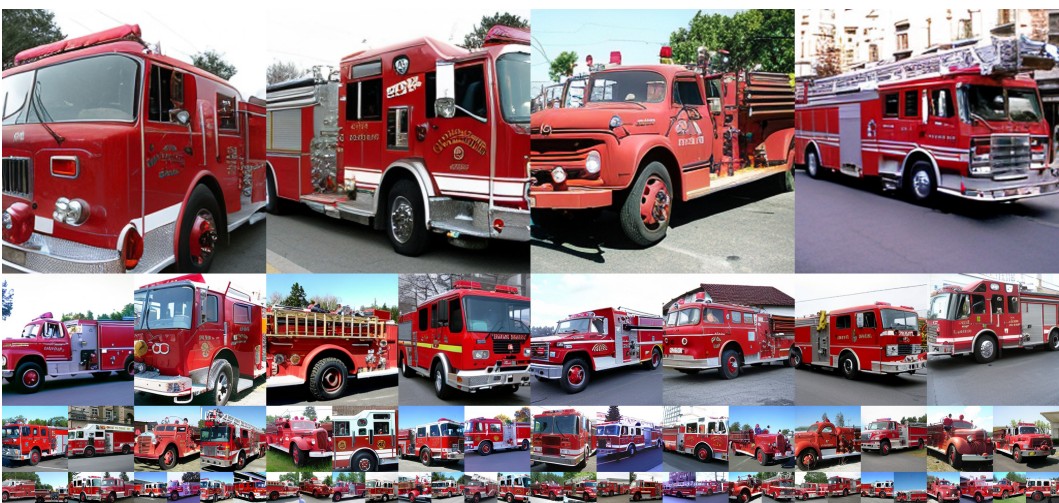

Figure 27: **Uncurated generation results of SiT-XL/2 + REPA.** We use classifier-free guidance with $w = 4.0$. Class label = "fire truck" (555).

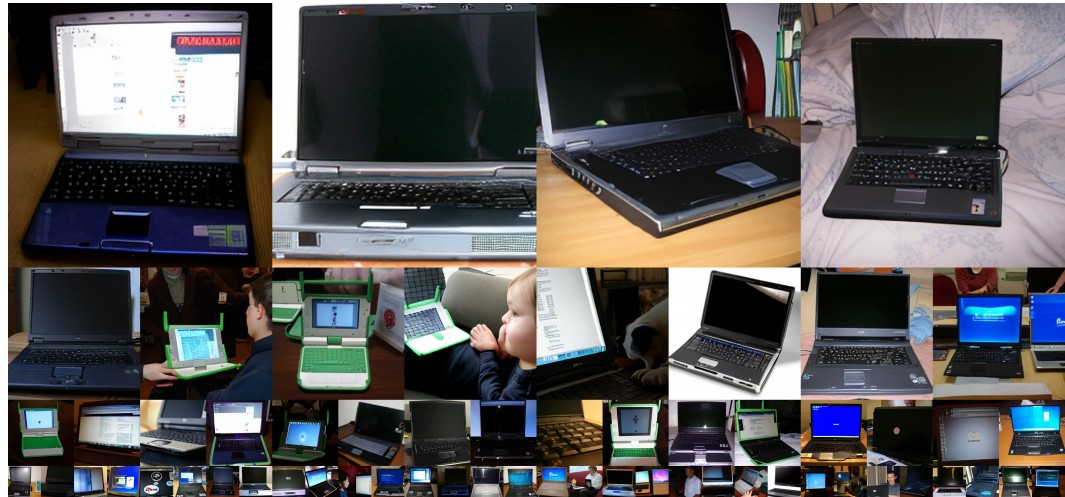

Figure 28: **Uncurated generation results of SiT-XL/2 + REPA.** We use classifier-free guidance with $w = 4.0$. Class label = "laptop" (620).

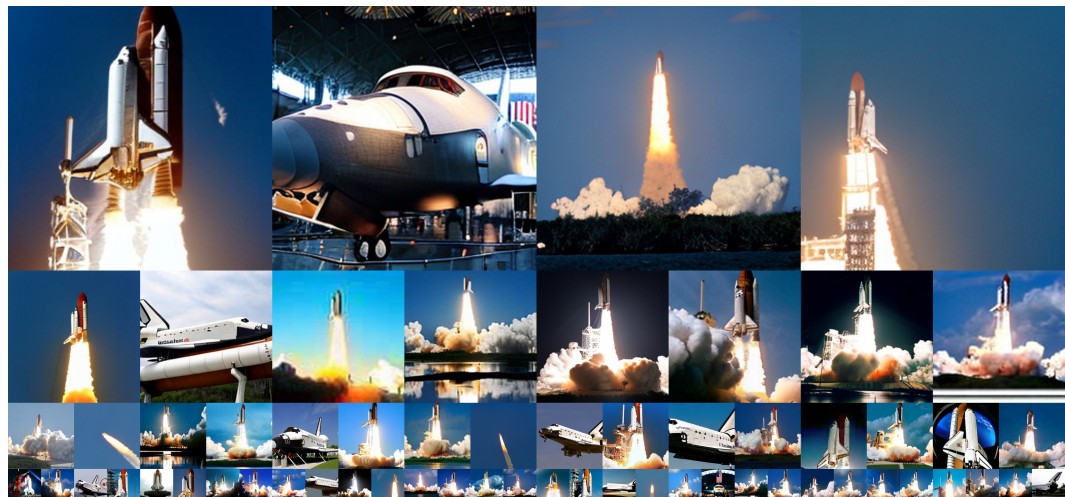

Figure 29: **Uncurated generation results of SiT-XL/2 + REPA.** We use classifier-free guidance with $w = 4.0$. Class label = "space shuttle" (812).

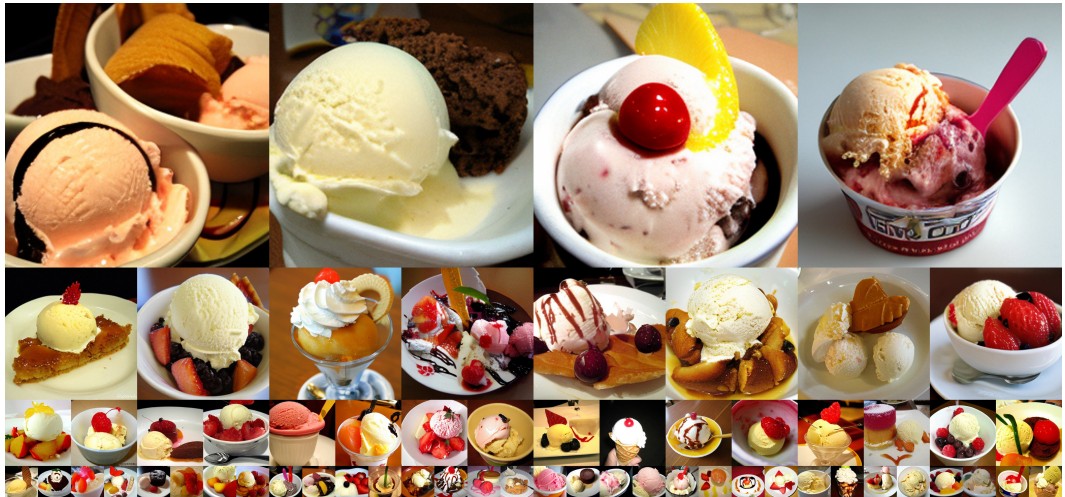

Figure 30: **Uncurated generation results of SiT-XL/2 + REPA.** We use classifier-free guidance with $w = 4.0$. Class label = "ice cream" (928).

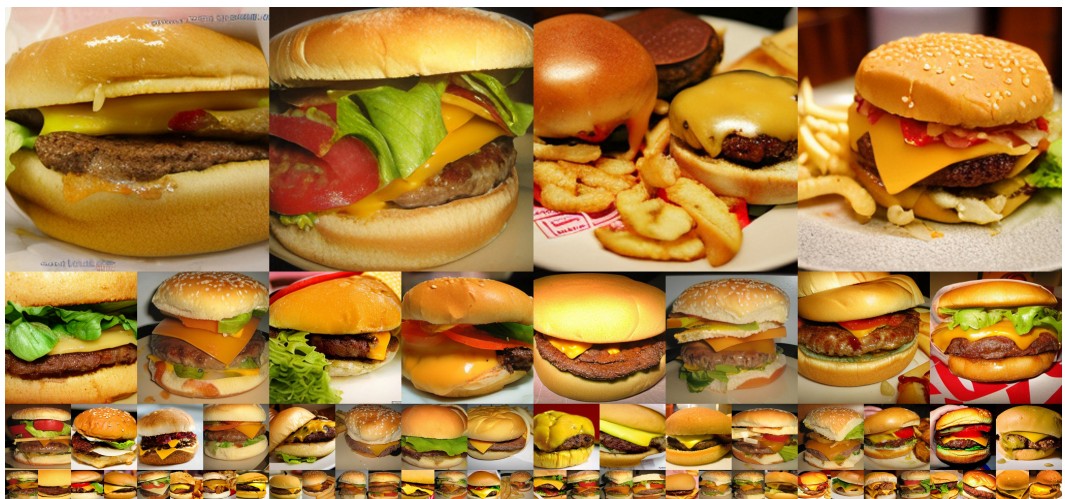

Figure 31: **Uncurated generation results of SiT-XL/2 + REPA.** We use classifier-free guidance with $w = 4.0$. Class label = "cheeseburger" (933).

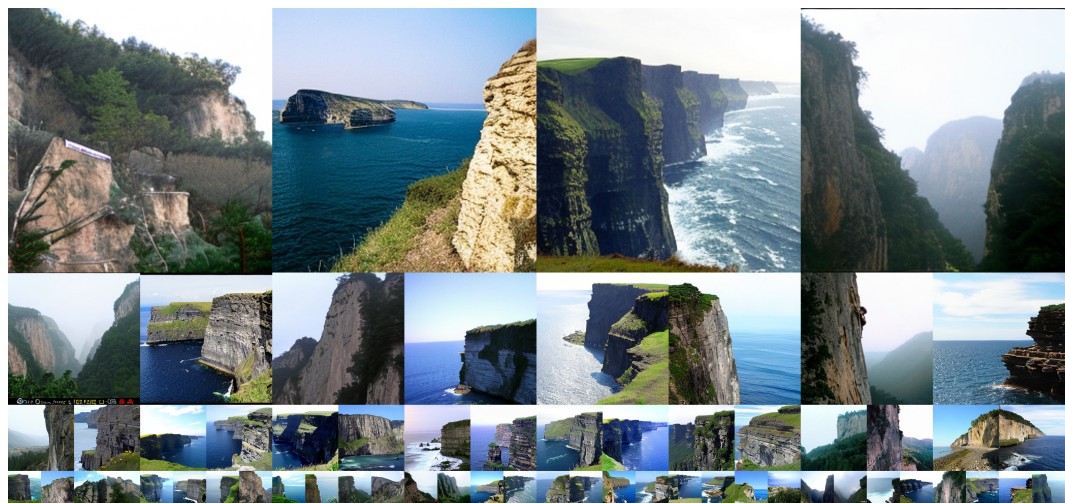

Figure 32: **Uncurated generation results of SiT-XL/2 + REPA.** We use classifier-free guidance with $w = 4.0$. Class label = "cliff drop-off" (972).

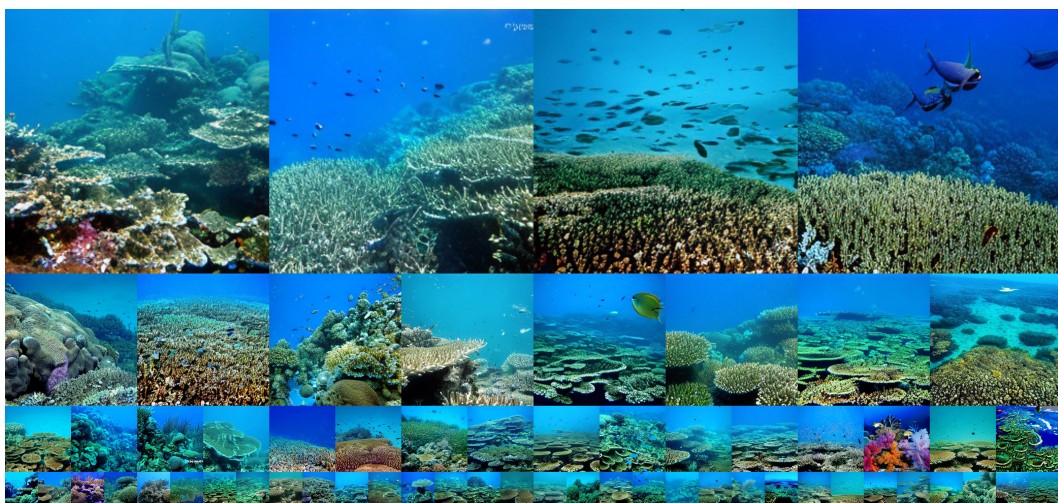

Figure 33: **Uncurated generation results of SiT-XL/2 + REPA.** We use classifier-free guidance with $w = 4.0$. Class label = "coral reef" (973).

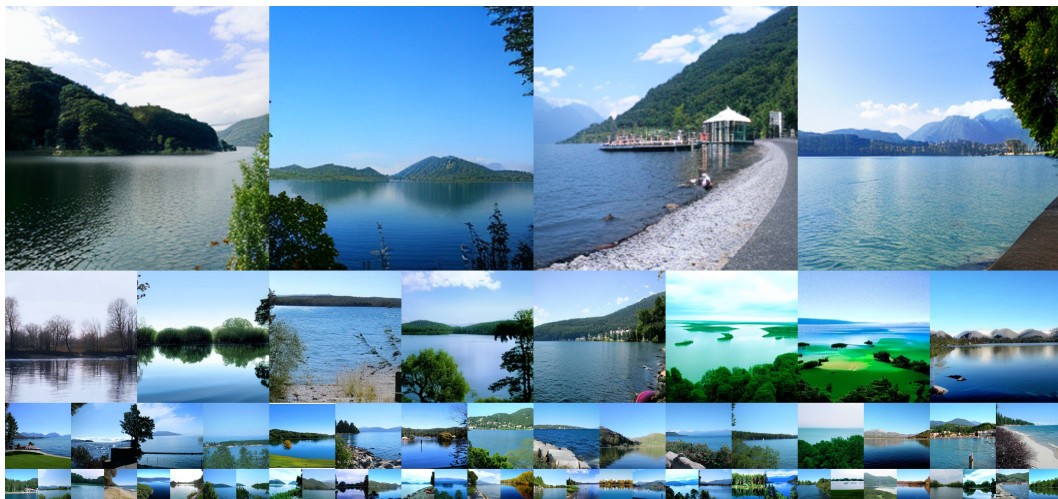

Figure 34: **Uncurated generation results of SiT-XL/2 + REPA.** We use classifier-free guidance with $w = 4.0$. Class label = "lake shore" (975).

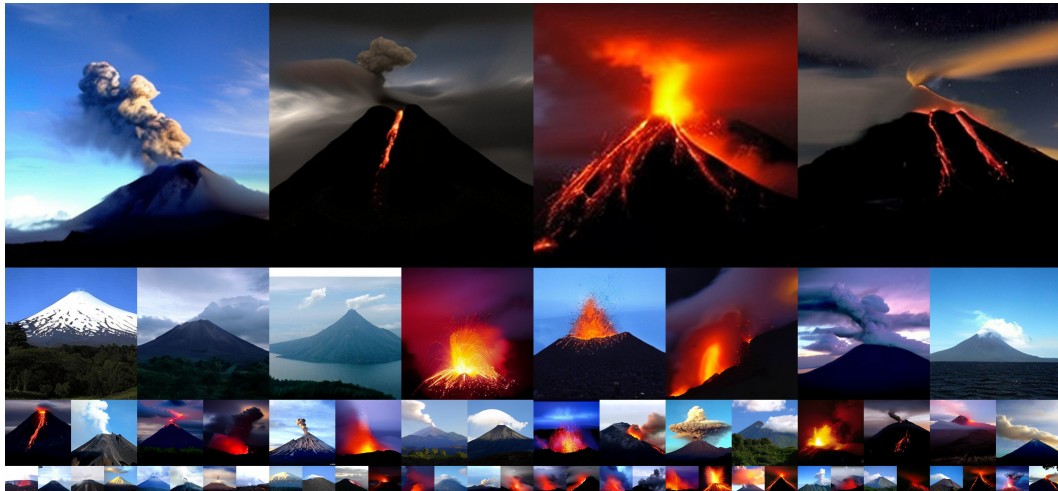

Figure 35: **Uncurated generation results of SiT-XL/2 + REPA.** We use classifier-free guidance with $w = 4.0$. Class label = "volcano" (980).

# I   MORE DISCUSSION ON RELATED WORK

**Pretrained visual encoders for generative models.** First, there have been several approaches in generative adversarial network (GAN; Goodfellow et al. 2014) that try to accelerate training with better convergence using pretrained visual encoders (Sauer et al., 2021; Kumari et al., 2022; Sauer et al., 2022; 2023a; Kang et al., 2023). They usually use pretrained visual encoders as a discriminator by leveraging their intermediate features. This approach has also been applied to the distillation of diffusion models with adversarial objectives (Sauer et al., 2023b; 2024; Kang et al., 2024). Another line of work tries to exploit the pretrained visual encoders for improving diffusion model training from scratch (Pernias et al., 2024; Li et al., 2024), usually by training two diffusion models where one model generates the pretrained representations and the other model generates the target data conditioned on the generated representation. Our method also tries to improve diffusion model training through pretrained visual encoders, but our motivation is in the alignment between the diffusion model representation and recent self-supervised visual representations.

**Denoising transformers.** Many recent works have tried to use transformer backbones for diffusion or flow-based model training. First, several works like U-ViT (Bao et al., 2023), MDT (Gao et al., 2023), and DiffiT (Hatamizadeh et al., 2024) show transformer-based backbones with *skip connections* can be an effective backbone for training diffusion models. Intriguingly, DiT (Peebles & Xie, 2023) show skip connections are not even necessary components, and a pure transformer architecture can be a scalable architecture for training diffusion-based models. Based on DiT, SiT (Ma et al., 2024a) shows the model can be further improved with continuous stochastic interpolants (Albergo et al., 2023). Moreover, VDT (Lu et al., 2024) and Latte (Ma et al., 2024b) show DiTs can be extended for video generation through a space-time attention (Arnab et al., 2021). Based on these improvements, Pixart-$\alpha$ (Chen et al., 2024b), Pixart-$\Sigma$ (Chen et al., 2024a), Stable diffusion 3 (Esser et al., 2024) show pure transformers can be scaled up for challenging text-to-image generation, and CMD (Yu et al., 2024), WALT (Gupta et al., 2024), and Sora (Brooks et al., 2024) demonstrates their success in text-to-video generation. Our work analyzes and improves the training of DiT (and SiT) architecture based on a simple feature matching regularization to the early layers.

**Generative models with auxiliary self-supervised tasks.** MaskDiT (Zheng et al., 2024) combines mask reconstruction in MAE (He et al., 2022) to diffusion model training for faster diffusion model training. Similarly, SD-DiT (Zhu et al., 2024) shows diffusion model training can be improved with an auxiliary discriminative self-supervised loss. MAGE (Li et al., 2023c) bridge MAE training and masked image modeling (Chang et al., 2022) by adjusting the masking ratio in training, which leads to a single model both capable of discrimination and generation tasks. Our method also has a similarity to these works, where our training scheme has an additional distillation loss to projection of diffusion transformer hidden states.

**Denoising as self-supervised learning task.** There have been some attempts to improve self-supervised learning with denoising: Abstreiter et al. (2021) extends the diffusion objective for a better representation learning scheme, and Chen et al. (2024c) deconstructs diffusion models to improve denoising-based representation learning. Hudson et al. (2024) introduces an encoder that learns a representation by guiding a diffusion with its output as a compact latent vector. Zaidi et al. (2023) focuses on the molecular domain and proposes a pretraining scheme based on denoising. Our work also analyzes representational gap between popular self-supervised networks and those learned from pretrained diffusion models.

## J IMAGENET 512×512 EXPERIMENT

To validate the scalability of REPA with respect to the input image resolution, we conduct an additional experiment on ImageNet 512× 512. We strictly follow the setup used in our ImageNet 256×256 experiment except an input dimension. Specifically, the input dimension for SiT becomes a $64 \times 64 \times 4$ compressed latent image from $512 \times 512 \times 3$ image pixels using stable diffusion VAE (Rombach et al., 2022). Moreover, we use an image resized to $448 \times 448$ resolution for an input to the DINOv2 (Oquab et al., 2024) encoder with an interpolation of the positional embedding.

We provide quantitative results in Table 11 and qualitative result in Figure 36. Notably, as shown in this table, the model (with REPA) already outperforms the vanilla SiT-XL/2 in terms of four metrics (FID, sFID, IS, and Prec) using >3× fewer training iterations.

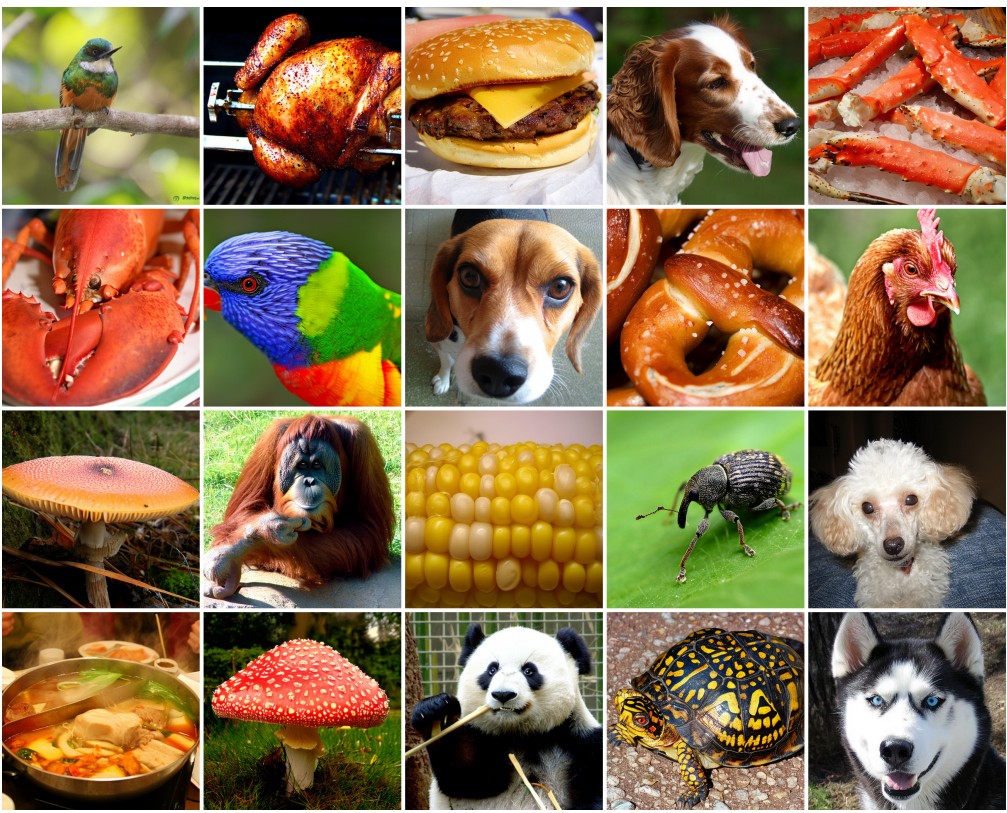

Figure 36: **Samples on ImageNet 512×512** from SiT-XL/2+REPA using CFG with $w = 4.0$.

Table 11: **System-level comparison** on ImageNet 512×512. We use CFG with $w = 1.35$.

| Model | Epochs | FID↓ | sFID↓ | IS↑ | Pre.↑ | Rec.↑ |
|---|---|---|---|---|---|---|
| *Pixel diffusion* | | | | | | |
| VDM++ | - | 2.65 | - | 278.1 | - | - |
| ADM-G, ADM-U | 400 | 2.85 | 5.86 | 221.7 | 0.84 | 0.53 |
| Simple diffusion (U-Net) | 800 | 4.28 | - | 171.0 | - | - |
| Simple diffusion (U-ViT, L) | 800 | 4.53 | - | 205.3 | - | - |
| *Latent diffusion, Transformer* | | | | | | |
| MaskDiT | 800 | 2.50 | 5.10 | 256.3 | 0.83 | 0.56 |
| DiT-XL/2 | 600 | 3.04 | 5.02 | 240.8 | 0.84 | 0.54 |
| SiT-XL/2 | 600 | 2.62 | 4.18 | 252.2 | 0.84 | 0.57 |
| + REPA (ours) | 80 | 2.44 | 4.21 | 247.3 | 0.84 | 0.56 |
| + REPA (ours) | 100 | 2.32 | **4.16** | 255.7 | **0.84** | 0.56 |
| + REPA (ours) | 200 | **2.08** | 4.19 | **274.6** | 0.83 | **0.58** |

## K    TEXT-TO-IMAGE GENERATION EXPERIMENT

We also validate REPA in text-to-image generation. We mostly follow the experimental setup used in U-ViT (Bao et al., 2023) unless otherwise specified: we train the model from scratch on a train split of the MS-COCO dataset (Lin et al., 2014) and use a validation split for evaluation. We use MMDiT (Esser et al., 2024), a simple variant of DiT that design attention layers to be jointly computed with image patches and text embeddings. We train MMDiT models for 150K iterations with a batch size of 256. We set a hidden dimension as 768 and a model depth as 24, and we use the CLIP (Radford et al., 2021) text encoder to compute text prompts from captions.

We report the results in Table 12 and Figure 37. First, as shown in the qualitative comparison in Figure 37, REPA shows consistently better results than the vanilla model. Moreover, as shown in Table 12, REPA also shows considerable improvements in T2I generation, highlighting the importance of alignment of visual representations even under the presence of text representations.

In this respect, we strongly believe that training large-scale T2I models with large-scale data using REPA will be a promising direction in the future.

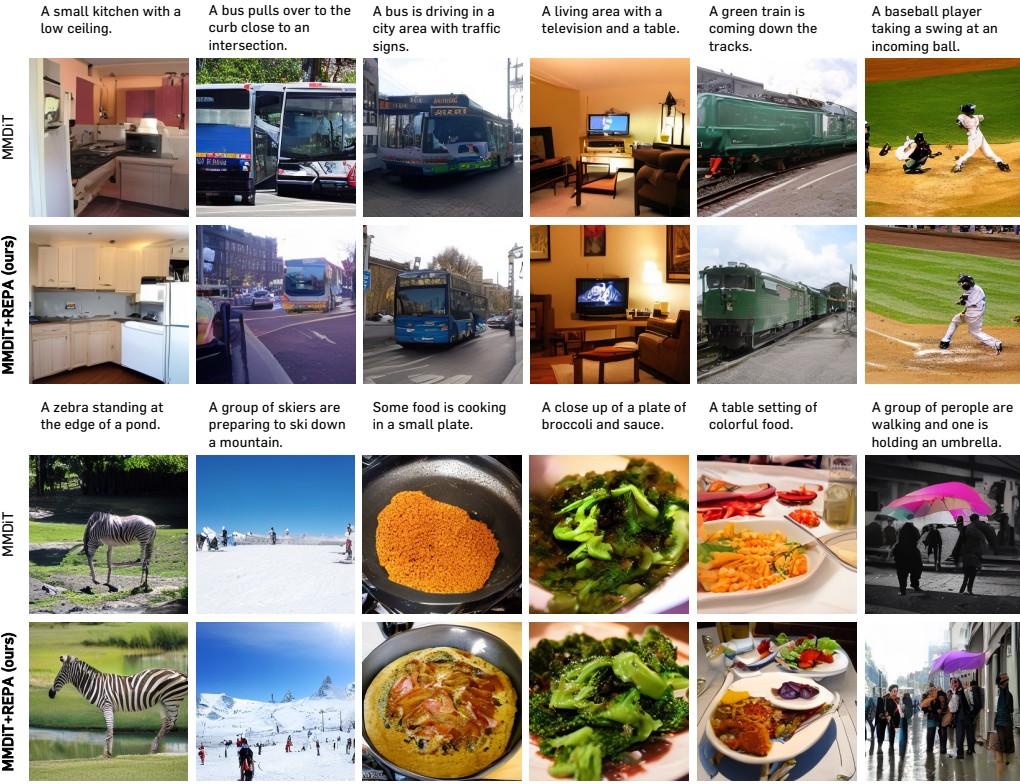

Figure 37: **Qualitative comparison on text-to-image generation (MS-COCO)**. We use classifier-free guidance with $w = 4.0$.

| Method | Type | FID |
|---|---|---|
| AttnGAN (Xu et al., 2018) | GAN | 35.49 |
| DM-GAN (Zhu et al., 2019) | GAN | 32.64 |
| VQ-Diffusion (Gu et al., 2022) | Discrete Diffusion | 19.75 |
| DF-GAN (Tao et al., 2022) | GAN | 19.32 |
| XMC-GAN (Zhang et al., 2021) | GAN | 9.33 |
| Frido (Fan et al., 2023) | Diffusion | 8.97 |
| LAFITE (Zhou et al., 2021) | GAN | 8.12 |
| U-Net (Bao et al., 2023) | Diffusion | 7.32 |
| U-ViT-S/2 (Bao et al., 2023) | Diffusion | 5.95 |
| U-ViT-S/2 (Deep) (Bao et al., 2023) | Diffusion | 5.48 |
| MMDiT (ODE; NFE=50) | Diffusion | 6.05 |
| **MMDiT+REPA (ODE; NFE=50)** | Diffusion | **4.73** |
| MMDiT (SDE; NFE=250) | Diffusion | 5.30 |
| **MMDiT+REPA (SDE; NFE=250)** | Diffusion | **4.14** |

Table 12: Quantitative comparison on text-to-image generation (MS-COCO). We use classifier-free guidance with $w = 2.0$ following the setup in (Bao et al., 2023).

## L  FEATURE MAP VISUALIZATION

We provide PCA visualizations of feature map, similar to those in DINOv2 (Oquab et al., 2024). As shown in Figure 38, REPA shows coarse-to-fine feature maps, while the vanilla model tends to show noisy feature map particular for large $t$.

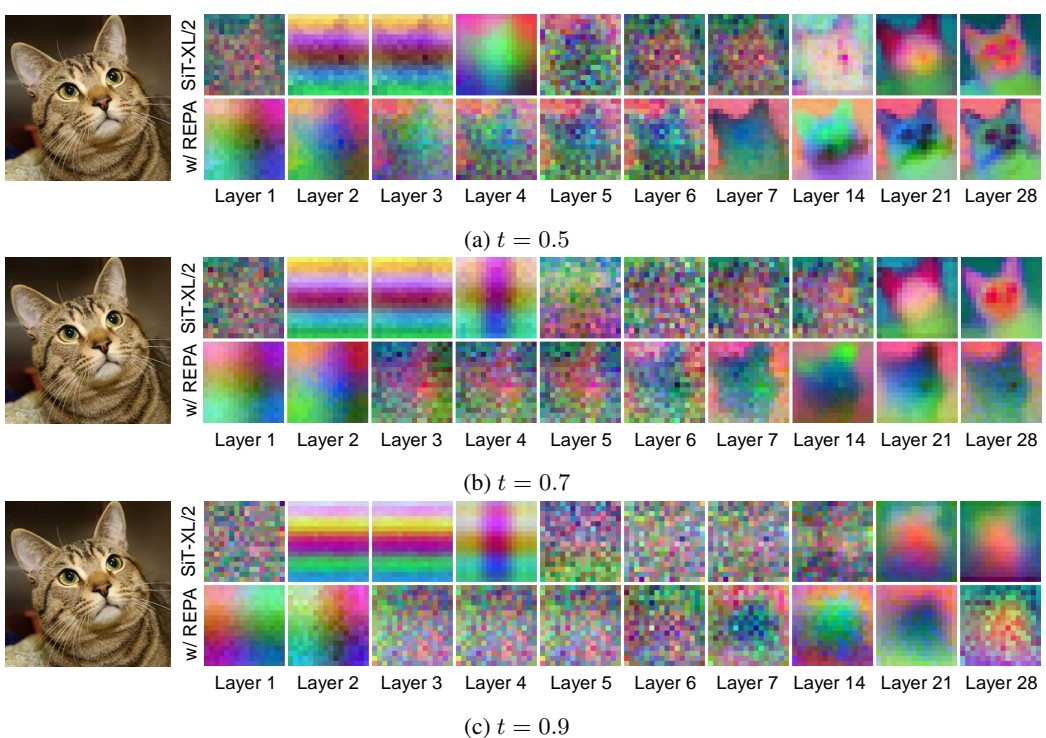

Figure 38: **PCA visualization** of layer-wise features of SiT-XL/2 and SiT-XL/2+REPA.

## M  LIMITATIONS AND FUTURE WORK

In this section, we enumerate several possible future research directions in what follows.

**Alignment depth.** Recall that we empirically showed that applying REPA to layer 8 is more beneficial than to later transformer layer embeddings (see Table 2); conducting extensive analysis of this result will be an interesting direction to further improve REPA.

**Different input data types.** We mainly focused on latent diffusion in the image domain. Exploring REPA with pixel-level diffusion or on other data domains like videos would be an interesting future work. Moreover, based on our text-to-image generation results on MS-COCO, training large-scale text-to-image diffusion models with REPA will also be an interesting direction.

**Theoretical analysis.** Exploring theoretical insights into why REPA works well will also be an exciting future direction. For instance, it will be interesting to explore the relationship between representations learned with an instance discrimination objective and a denoising objective.

**Time-varying REPA.** We think one of the interesting possible directions can be designing a weight function based on a noise schedule used in the diffusion process. We have not explored this in this work as our main focus is more on performing extensive analysis on other perspectives, such as target representations used for alignment, alignment depth, scalability of the method, *etc.* We leave this as an exciting direction for future work.

