# OpenReview forum: "Representation Alignment for Generation: Training Diffusion Transformers Is Easier Than You Think"
_ICLR.cc/2025/Conference — ICLR 2025 Oral_

### Official Review · Reviewer_MZYg · 2024-10-22

**Soundness:** 4
**Presentation:** 3
**Contribution:** 3
**Rating:** 10
**Confidence:** 4

**Summary:**

This paper finds that learning a good representation in diffusion transformers will speed up the training process and achieve better performance. By simply aligning diffusion transformer representations with recent self-supervised representations (DINOv2), the DDPM-based models (DiT) and flow-based models (SiT) show significant FID metric drops in early training.

**Strengths:**

This paper astutely captures the feature representation gap between generative and discriminative models, providing a feasible approach to enhance model representation and accelerate the training process of DiT models. The extensive analysis and observations regarding representation clarify the underlying motivation. Additionally, the experiments are detailed and validate the effectiveness of the proposed method across different settings. The presentation of Table 2 and the experiment section is very clear.

**Weaknesses:**

1. All the experiments in this paper are conducted under the setting of ImageNet 256x256. Although this work shows good results in this setting, it is still far from being applied to the currently popular high-resolution transformer-based text-to-image generation models, such as Flux. I would be more excited to see results and observations at higher resolutions or in text-to-image generation. If the generation is limited to the DiT/SiT class condition, its applications will be severely restricted.
2. Metrics for ImageNet generation often confuse me; the FID, sFID, IS, and other metrics frequently show differing trends. And FID does not effectively reflect actual quality differences at low values. How do you consider these metrics comprehensively to make a reasonable judgment about performance? For instance, in the last two rows of Table 2, NT-Xent outperforms in sFID, while Cos. sim. outperforms in FID and IS. And it would be beneficial to provide some experimental results to explain why you chose NT-Xent, rather than simply stating "Empirically, ..." in line 421. I would be glad to discuss this further with you in the Questions section.

**Questions:**

1. The experimental results indicate that the speedup for SiT training is quite significant. However, can it achieve state-of-the-art performance in the final results? MDT (ICCV'23) [1] seems to have achieved an FID of 1.79.
2. Let's discuss the gap between FID metrics and actual image performance, in Figure 6, your visualization are under classifier-free guidance with $w = 4.0$, but SiT-XL/2 with REPA achieves best FID of 1.80 under classifier-free guidance scale of $w = 1.35$. What do you think of the difference w between best visualization and best metrics, and whether this measure is unreasonable?
3. I agree with "the diffusion transformer model needs some capacity for capturing high-frequency details after learning a meaningful representation that contains good semantics.", but minimum semantic gap are at layer 20 before alignment with REPA, which is different between layer 8 after alignment with REPA  (I note this because layer 8 yields the best metrics). Could you further explain this difference?
4. To more clearly represent the process of alignment in the representation, I recommend including the definition of layers in formula (8). I suggest using $h^{[n,m]}$ to represent diffusion transformer encoder output, where $n$ is a patch index and $m$ is layer index. There is a small parenthesis error in formula (8).

[1] Gao, Shanghua, et al. "Masked diffusion transformer is a strong image synthesizer." Proceedings of the IEEE/CVF International Conference on Computer Vision. 2023.

---

> ### Author Response · Authors · 2024-11-22
> **Response to Reviewer MZYg (1/2)**
>
> We deeply appreciate your insightful comments and efforts in reviewing our manuscript. We respond to each of your comments one-by-one in what follows. In the revised draft, we mark our major revisions as “blue”.
>
> ---
>
> **[W1-1] Higher resolution experiments (e.g., 512x512)**
>
> Thanks for the suggestion! We agree that high-resolution results will make the paper more convincing. To address this concern, we additionally train SiT-XL/2 on ImageNet-512x512 with REPA and report the quantitative evaluation in the table below. Notably, as shown in this table, the model (with REPA) already outperforms the vanilla SiT-XL/2 in terms of four metrics (FID, sFID, IS, and Rec) using >3x fewer training iterations. We added this result in Appendix J in the revised manuscript with qualitative results. We will also add the final results in the final draft with longer training (please understand that it takes more than 2 weeks, so it will not be possible to update them until the end of the discussion period).
>
> \begin{array}{lcccccc}
> \hline
> \phantom{0}\phantom{0} \text{Model} & \text{Epochs} & \text{FID $\downarrow$} & \text{sFID $\downarrow$} & \text{IS $\uparrow$} & \text{Pre. $\uparrow$} & \text{Rec. $\uparrow$} \newline
> \hline
> \text{\emph{Pixel diffusion}} \newline
> \phantom{0}\phantom{0} \text{VDM++} & - & 2.65 & - & 278.1 & - & - \newline
> \phantom{0}\phantom{0} \text{ADM-G, ADM-U} & 400 & 2.85 & 5.86 & 221.7 & 0.84 & 0.53 \newline
> \phantom{0}\phantom{0} \text{Simple diffusion (U-Net)} & 800 & 4.28 & - & 171.0 & - & - \newline
> \phantom{0}\phantom{0} \text{Simple diffusion (U-ViT, L)} & 800 & 4.53 & - & 205.3 & - & - \newline
> \hline
> \text{\emph{Latent diffusion, Transformer}} \newline
> \phantom{0}\phantom{0} \text{MaskDiT} & 800 & 2.50 & 5.10 & 256.3 & 0.83 & 0.56 \newline
> \phantom{0}\phantom{0} \text{DiT-XL/2} & 600 & 3.04 & 5.02 & 240.8 & 0.84 & 0.54 \newline
> \phantom{0}\phantom{0} \text{SiT-XL/2} & 600 & 2.62 & 4.18 & 252.2 & 0.84 & 0.57 \newline
> \phantom{0}\phantom{0} \text{+REPA (ours)} & \phantom{0}80 & {2.44} & 4.21 & 247.3 & 0.84 & 0.56 \newline
> \phantom{0}\phantom{0} \text{+REPA (ours)} & 100 & 2.32 & \bf{4.16} &  255.7 & 0.84 & 0.56 \newline
> \phantom{0}\phantom{0} \text{+REPA (ours)} & 200 & \bf{2.08} & 4.19 & \bf{274.6} & 0.83 & \bf{0.58} \newline
> \hline
> \end{array}
>
> **[W1-2] Text-to-image experiments.**
>
> To address your concern, we have additionally conducted a text-to-image (T2I) generation experiment on MS-COCO [1], following the setup in previous literature [2]. We tested REPA on MMDiT [3], a variant of DiT designed for T2I generation, given that the standard DiT/SiT architectures are not suited for this task. As shown in the table below, REPA also shows clear improvements in T2I generation, highlighting the importance of alignment of visual representations even under the presence of text representations. In this respect, we believe that scaling up REPA training to large-scale T2I models will be a promising direction for future research. We added these results with qualitative comparison in Appendix K in the revised manuscript.
>
> \begin{array}{lcc}
> \hline
> \text{Method} & \text{Type} & \text{FID} \newline
> \hline
> \text{AttnGAN} & \text{GAN} & 35.49 \newline
> \text{DM-GAN}  & \text{GAN} & 32.64 \newline
> \text{VQ-Diffusion} & \text{Discrete Diffusion} & 19.75 \newline
> \text{DF-GAN}  & \text{GAN} & 19.32 \newline
> \text{XMC-GAN} & \text{GAN} & \phantom{0}9.33  \newline
> \text{Frido}   & \text{Diffusion} & \phantom{0}8.97 \newline
> \text{LAFITE}  & \text{GAN} & \phantom{0}8.12 \newline
> \hline
> \text{U-Net}  & \text{Diffusion} & \phantom{0}7.32 \newline
> \text{U-ViT-S/2} & \text{Diffusion} & \phantom{0}5.95 \newline
> \text{U-ViT-S/2 (Deep)} & \text{Diffusion} & \phantom{0}5.48 \newline
> \hline
> \text{MMDiT (ODE; NFE=50, 150K iter)} & \text{Diffusion} & \phantom{0}6.05 \newline
> \textbf{MMDiT+REPA (ODE; NFE=50, 150K iter)} & \text{Diffusion} & \phantom{0}\bf{4.73} \newline
> \hline
> \text{MMDiT (SDE; NFE=250, 150K iter)} & \text{Diffusion} & \phantom{0}5.30 \newline
> \textbf{MMDiT+REPA (SDE; NFE=250, 150K iter)} & \text{Diffusion} & \phantom{0}\bf{4.14} \newline
> \hline
> \end{array}
>
>
> [1] Microsoft COCO: Common Objects in Context, ECCV 2014
> [2] All are Worth Words: A ViT Backbone for Diffusion Models, CVPR 2023
> [3] Scaling Rectified Flow Transformers for High-Resolution Image Synthesis, ICML 2024

---

> ### Author Response · Authors · 2024-11-22
> **Response to Reviewer MZYg (2/2)**
>
> **[W2] Comprehensive analysis: different metric trends for ImageNet generation.**
>
> We mainly use FID (and IS) metrics as criteria for determining design choices, as these two metrics are the most popular, widely-used metrics, and they also showed quite clear trends across different design choices. Moreover, following your suggestion, we provide additional quantitative evaluation results for NT-Xent and cos. sim at the early training stage in the table below (due to the limited time we have, we used smaller NFE=50 here, different from NFE=250 in the manuscript). The results show that NT-Xent is more beneficial at early iterations, but the gap diminishes with longer training.
>
> \begin{array}{lcccc}
> \hline
> \text{Method} & \text{50K} & \text{100K} & \text{200K} & \text{400K} \newline
> \hline
> \text{NT-Xent} & 26.29 & 18.11 & 15.61 & 14.38 \newline
> \text{Cos. sim.} & 29.21 & 18.42 & 15.43 & 14.17 \newline
> \hline
> \end{array}
>
> ---
>
> **[Q1] Can REPA achieve state-of-the-art results (outperforming FID=1.79 of MDT)?**
>
> First, in Appendix G, we already have shown that REPA can achieve the same FID of MDT (1.79) by using a slightly smaller CFG scale $w=1.325$. Moreover, note that MDT uses CFG coefficient scheduling that can dramatically improve the performance [4, 5], where we used a constant coefficient and did not apply this technique to achieve the same FID.
>
> We also conducted an additional evaluation by applying CFG scheduling similar to MDT. If we apply a simple scheduling technique proposed in recent work [5], REPA achieves significantly improved FID=1.42. We added this result in Table 4 and Appendix G in the revised manuscript.
>
> [4] Masked Diffusion Transformer is a Strong Image Synthesizer, ICCV 2023.
> [5] Applying Guidance in a Limited Interval Improves Sample and Distribution Quality in Diffusion Models, to appear at NeurIPS 2024
>
> ---
>
> **[Q2] Why different CFG scale is used for quantitative and qualitative comparison?**
>
> This is a quite common practice in most diffusion model literature [6, 7, 8], and we simply followed their setup. Specifically, generating samples with a large CFG coefficient can provide high-quality individual samples, but they lack diversity [9], potentially leading to quantitative evaluation degradation.
>
> [6] Scalable Diffusion Models with Transformers, ICCV 2023
> [7] SiT: Exploring Flow and Diffusion-based Generative Models with Scalable Interpolant Transformers, ECCV 2024
> [8] High-Resolution Image Synthesis with Latent Diffusion Models, CVPR 2022
> [9] Classifier-free Diffusion Guidance, arXiv 2022
>
> ---
>
> **[Q3] Why do minimum semantic differences appear at later layers without REPA?**
>
> We hypothesize that the denoising task used for diffusion model training, which is a sort of reconstruction, may not be a suitable task for learning good representations. Specifically, as mentioned in the Introduction of our original manuscript,  it is not capable of eliminating unnecessary details in the image for representation learning [10, 11]. Thus, representation might be learned inefficiently and appear after unnecessarily many layers, even though the first few layers of the model have enough capacity to learn meaningful representations (note that we showed it through applying REPA at the 8th layer). Exploring such different behaviors will be an exciting future direction, and we leave it for future work.
>
> [10] A Path towards Autonomous Machine Intelligence Version, OpenReview 2022
> [11] Self-supervised learning from images with a joint-embedding predictive architecture, CVPR 2023
>
> ---
>
> **[Q4] Notation**.
>
> Thanks for the suggestion! We reflected them (notation and fixing typos) in the revised manuscript.

---

> > ### Comment · Reviewer_MZYg · 2024-11-24
> >
> > Thanks to the author's reply and experiment, which addressed most of my concerns, I think this is a good paper, and I keep my rating 8.

---

> > > ### Author Response · Authors · 2024-11-24
> > > **Response to Reviewer MZYg**
> > >
> > > Dear Reviewer MZYg,
> > >
> > > We are happy to hear that our rebuttal addressed your concerns well! Also, we appreciate your support for our work. If you have any further questions or suggestions, please do not hesitate to let us know.
> > >
> > > Best regards,
> > > Authors

---

> > > > ### Comment · Reviewer_MZYg · 2024-11-26
> > > >
> > > > Dear Author,
> > > >
> > > > Since the discussion period has been delayed, I still have some questions about the manuscript that I would like to discuss with you further.
> > > >
> > > > **1.Additional Metrics for Text-to-Image Evaluation:**
> > > > Several reviewers, including myself, have mentioned that the application of REPA in text-to-image generation can bring novel insights. While I acknowledge the good performance of MMDiT on MSCOCO, the FID metric alone is not always indicative of the actual quality in text-to-image generation practices. Therefore, I suggest that the authors include additional metrics, such as the CLIP score or aesthetics score, or conduct user studies to further evaluate REPA's performance in text-to-image generation.
> > > >
> > > > **2.Application to Pretrained Models:**
> > > > Furthermore, we are all aware that pre-training for text-to-image generation requires substantial computational resources. If REPA can be quickly fine-tuned and applied to existing pretrained models through methods like LoRA or other techniques to achieve feature alignment and demonstrate stronger performance, it would significantly enhance its application value.
> > > >
> > > > I will consider raising my score if the author can consider the above two improvement, because it can further expand the practical application of text to image generation.

---

> > > > > ### Author Response · Authors · 2024-11-30
> > > > > **Response to Reviewer MZYg**
> > > > >
> > > > > We deeply appreciate your additional feedback and questions on improving the paper further. We respond to each of your additional questions one-by-one in what follows.
> > > > >
> > > > > ---
> > > > >
> > > > > **[AQ1] Additional Metrics for Text-to-Image Evaluation.”**
> > > > >
> > > > > Following your suggestion, we have additionally compared the CLIP score (using CLIP ViT-B/32 [1]) and the Aesthetics score (using the fine-tuned ConvNeXt-Base CLIP [2]) of MMDiT and MMDiT+REPA as in the table below. Indeed, we observe MMDiT+REPA not only improves FID but also CLIP score and Aesthetics significantly, verifying the effectiveness of REPA on text-image and perceptual alignments, respectively. This is also qualitatively shown in Figure 37 in our manuscript: e.g., for a prompt “A zebra standing at the edge of a pond,” MMDiT struggles to generate a pond, and the zebra is on the ground, but MMDiT+REPA generates ponds and correctly locates the generated zebra at the edge of the pond. We will add this discussion in the final draft.
> > > > >
> > > > > [1] https://huggingface.co/openai/clip-vit-base-patch32
> > > > > [2] https://huggingface.co/laion/CLIP-convnext_base_w_320-laion_aesthetic-s13B-b82K
> > > > >
> > > > > \begin{array}{lcc}
> > > > > \hline
> > > > > \text{Method} & \text{FID}\downarrow & \text{CLIPSIM}\uparrow  & \text{Aesthetics}\uparrow \newline
> > > > > \hline
> > > > > \text{MMDiT (ODE; NFE=50, 150K iter)} & \phantom{0}6.05 & 0.2883  & 0.3757 \newline
> > > > > \textbf{MMDiT+REPA (ODE; NFE=50, 150K iter)} & \phantom{0}\textbf{4.73}& \textbf{0.2940} & \textbf{0.3915} \newline
> > > > > \hline
> > > > > \text{MMDiT (SDE; NFE=250, 150K iter)} & \phantom{0}5.30 & 0.2894 & 0.3793 \newline
> > > > > \textbf{MMDiT+REPA (SDE; NFE=250, 150K iter)} & \phantom{0}\bf{4.14} & \textbf{0.2950}  & \textbf{0.3944} \newline
> > > > > \hline
> > > > > \end{array}
> > > > >
> > > > >
> > > > >
> > > > >
> > > > > ---
> > > > >
> > > > > **[AQ2] Application to Pretrained Models.**
> > > > >
> > > > > Thank you for the suggestion. We agree that exploring the applicability of REPA for fine-tuning (text-to-image) pretrained models can enhance the application values; for now, we leave this as an interesting future work. In particular, there can be some nontrivial design choices to be explored when adapting REPA for fine-tuning. For instance, REPA requires an MLP to project transformer hidden states, which are jointly optimized with the diffusion transformer, but existing pretrained models do not include such MLPs. Thus, one needs to attach a randomly initialized MLP to fine-tune pretrained models with REPA, which can affect the training dynamics. We expect that a careful consideration of such aspects would make REPA also beneficial in the fine-tuning setup. We will include this discussion and will continue to work on providing related experimental results and insights in the final draft.

---

> > > > > > ### Comment · Reviewer_MZYg · 2024-11-30
> > > > > > **Final decision and thanks to author's efforts**
> > > > > >
> > > > > > Thanks for the author's reply and efforts during the rebuttal.  After reviewing all the other reviewers' opinion and the discussion between the author and the reviewers, I decided to raise my score to 10, believing that this work will bring new insights and inspiration to the subsequent research and the community.

---

### Official Review · Reviewer_i9uk · 2024-10-29

**Soundness:** 3
**Presentation:** 4
**Contribution:** 3
**Rating:** 8
**Confidence:** 4

**Summary:**

This paper proposes a simple method to improve the diffusion-based generative model’s training efficiency. It discovers the semantic difference in self-supervised vision encoders and diffusion generative models, and the motivation is to use pre-trained vision encoder representation to regularize the diffusion transformer’s feature representation. The paper is well-written, the motivation, metrics, and methods are easy to understand, and the experiments conducted on various pre-trained vision encoders under different settings demonstrate the effectiveness of the proposed method in both generation quality measured in FID and representation ability measured in linear-probing accuracy.

**Strengths:**

--the proposed method is simple and very easy to implement, also the idea is well-motivated by the semantic measurement metrics

--the experiments are strong and complete, and examine various vision encoders under various setups (model size, depth) and with different DiT model sizes, all show promising improvement in terms of FID

**Weaknesses:**

-- the paper only shows results under resolution 256x256, which is the scale most of the vision encoders are pretrained on, it raises the concern if the pre-trained low-res representation can transfer to the generative models when train on higher resolution 512, 1024, 2048 etc

-- missing dataset analysis on the pre-trained vision encoder, i.e. is the performance of the encoder pre-trained on ImageNet the same as on in-the-wild dataset?

**Questions:**

--The pretrained vision encoder usually also finetunes on the ImageNet dataset, how do you separate the dataset leakage effect when measuring generative models trained on the ImageNet?

-- The same techniques can be applied to the video generative model, where training cost/efficiency is more important, do you think the proposed method can be transferred to the video domain?

---

> ### Author Response · Authors · 2024-11-22
> **Response to Reviewer i9uk (1/2)**
>
> We deeply appreciate your insightful comments and efforts in reviewing our manuscript. We respond to each of your comments one-by-one in what follows. In the revised draft, we mark our major revisions as “blue”.
>
> ---
>
> **[W1] Higher resolution experiments (e.g., 512x512).**
>
> Thanks for the suggestion! We agree that high-resolution results will make the paper more convincing. To address this concern, we additionally train SiT-XL/2 on ImageNet-512x512 with REPA and report the quantitative evaluation in the table below. Notably, as shown in this table, the model (with REPA) already outperforms the vanilla SiT-XL/2 in terms of four metrics (FID, sFID, IS, and Rec) using >3x fewer training iterations. We added this result in Appendix J in the revised manuscript with qualitative results. We will also add the final results in the final draft with longer training (please understand that it takes more than 2 weeks, so it will not be possible to update them until the end of the discussion period).
>
> \begin{array}{lcccccc}
> \hline
> \phantom{0}\phantom{0} \text{Model} & \text{Epochs} & \text{FID $\downarrow$} & \text{sFID $\downarrow$} & \text{IS $\uparrow$} & \text{Pre. $\uparrow$} & \text{Rec. $\uparrow$} \newline
> \hline
> \text{\emph{Pixel diffusion}} \newline
> \phantom{0}\phantom{0} \text{VDM++} & - & 2.65 & - & 278.1 & - & - \newline
> \phantom{0}\phantom{0} \text{ADM-G, ADM-U} & 400 & 2.85 & 5.86 & 221.7 & 0.84 & 0.53 \newline
> \phantom{0}\phantom{0} \text{Simple diffusion (U-Net)} & 800 & 4.28 & - & 171.0 & - & - \newline
> \phantom{0}\phantom{0} \text{Simple diffusion (U-ViT, L)} & 800 & 4.53 & - & 205.3 & - & - \newline
> \hline
> \text{\emph{Latent diffusion, Transformer}} \newline
> \phantom{0}\phantom{0} \text{MaskDiT} & 800 & 2.50 & 5.10 & 256.3 & 0.83 & 0.56 \newline
> \phantom{0}\phantom{0} \text{DiT-XL/2} & 600 & 3.04 & 5.02 & 240.8 & 0.84 & 0.54 \newline
> \phantom{0}\phantom{0} \text{SiT-XL/2} & 600 & 2.62 & 4.18 & 252.2 & 0.84 & 0.57 \newline
> \phantom{0}\phantom{0} \text{+REPA (ours)} & \phantom{0}80 & {2.44} & 4.21 & 247.3 & 0.84 & 0.56 \newline
> \phantom{0}\phantom{0} \text{+REPA (ours)} & 100 & 2.32 & \bf{4.16} &  255.7 & 0.84 & 0.56 \newline
> \phantom{0}\phantom{0} \text{+REPA (ours)} & 200 & \bf{2.08} & 4.19 & \bf{274.6} & 0.83 & \bf{0.58} \newline
> \hline
> \end{array}
>
> ---
>
> **[W2] Missing dataset analysis on the pretrained visual encoders.**
>
> Thanks for pointing this out. In the table below, we additionally provide the dataset analysis for visual encoders (we also added I-JEPA [1] and SigLIP [2] in Table 2 in the revision for a more extensive analysis). As shown in this table, better visual representations learned from massive amounts of image data provide more improvement, regardless of whether the dataset does not include ImageNet. We also added this discussion in Appendix C in the revised manuscript.
>
> \begin{array}{llccc}
> \hline
> \text{Method} & \text{Dataset} & \text{w/ ImageNet-1K} & \text{Text sup.} &
> \text{FID$\downarrow$} \newline
> \hline
> \text{MAE} & \text{ImageNet-1K} & \text{O} & \text{X} & 12.5 \newline
> \text{DINO} & \text{ImageNet-1K} & \text{O} & \text{X} & 11.9 \newline
> \text{MoCov3} & \text{ImageNet-1K} & \text{O} & \text{X} & 11.9 \newline
> \text{I-JEPA} & \text{ImageNet-1K} & \text{O} & \text{X} & 11.6 \newline
> \hline
> \text{CLIP} & \text{WIT-400M} & \text{X} & \text{O} & 11.0 \newline
> \text{SigLIP} & \text{WebLi-4B} & \text{X} & \text{O} & 10.2 \newline
> \text{DINOv2} & \text{LVD-142M} & \text{O} & \text{X} & 10.0 \newline
> \hline
> \end{array}

---

> ### Author Response · Authors · 2024-11-22
> **Response to Reviewer i9uk (2/2)**
>
> **[Q1] The pretrained vision encoder usually also finetunes on the ImageNet dataset; how do you separate the dataset leakage effect when measuring generative models trained on the ImageNet?**
>
> First, we did not fine-tune encoders (e.g., SigLIP and CLIP) with the ImageNet dataset, particularly if they were trained with another dataset, thereby separating the dataset leakage effect if we use these encoders. As shown in [W2], REPA also achieves significant improvements with these encoders, which validates that the improvement does not simply come from data leakages.
>
> To further address your concern, we conduct an additional text-to-image (T2I) generation experiment on MS-COCO by using DINOv2 as the target representation. Since the training dataset of DINOv2 does not include MS-COCO, it can validate whether the gain comes from data leakage or from aligning diffusion representation with a good visual representation. As shown in the table below, REPA with DINOv2 also shows improvement in this setup, which further highlights our claim.
>
> \begin{array}{lcc}
> \hline
> \text{Method} & \text{Type} & \text{FID} \newline
> \hline
> \text{AttnGAN} & \text{GAN} & 35.49 \newline
> \text{DM-GAN}  & \text{GAN} & 32.64 \newline
> \text{VQ-Diffusion} & \text{Discrete Diffusion} & 19.75 \newline
> \text{DF-GAN}  & \text{GAN} & 19.32 \newline
> \text{XMC-GAN} & \text{GAN} & \phantom{0}9.33  \newline
> \text{Frido}   & \text{Diffusion} & \phantom{0}8.97 \newline
> \text{LAFITE}  & \text{GAN} & \phantom{0}8.12 \newline
> \hline
> \text{U-Net}  & \text{Diffusion} & \phantom{0}7.32 \newline
> \text{U-ViT-S/2} & \text{Diffusion} & \phantom{0}5.95 \newline
> \text{U-ViT-S/2 (Deep)} & \text{Diffusion} & \phantom{0}5.48 \newline
> \hline
> \text{MMDiT (ODE; NFE=50, 150K iter)} & \text{Diffusion} & \phantom{0}6.05 \newline
> \textbf{MMDiT+REPA (ODE; NFE=50, 150K iter)} & \text{Diffusion} & \phantom{0}\bf{4.73} \newline
> \hline
> \text{MMDiT (SDE; NFE=250, 150K iter)} & \text{Diffusion} & \phantom{0}5.30 \newline
> \textbf{MMDiT+REPA (SDE; NFE=250, 150K iter)} & \text{Diffusion} & \phantom{0}\bf{4.14} \newline
> \hline
> \end{array}
>
> ---
>
> **[Q2] Apply REPA to video generation model?**
>
> Similar to the success in the image domain, we strongly believe that REPA can be applied to video diffusion models by exploiting strong video self-supervised representations, such as V-JEPA [1]. Exploring this should be an interesting future direction.
>
> [1] Revisiting Feature Prediction for Learning Visual Representations from Video, arXiv 2024.

---

> ### Author Response · Authors · 2024-11-25
> **Gentle Reminder**
>
> Dear Reviewer i9uk,
>
> Thank you again for your time and efforts in reviewing our paper.
>
> As the discussion period draws close, we kindly remind you that two days remain for further comments or questions. We would appreciate the opportunity to address any additional concerns you may have before the discussion phase ends.
>
> Thank you very much!
>
> Many thanks,
> Authors

---

> > ### Comment · Reviewer_i9uk · 2024-11-25
> >
> > After reading the author's rebuttal, it mostly addresses my concerns regarding the high-resolution training and dataset leakage of ImageNet. I raise my score to 8.

---

> > > ### Author Response · Authors · 2024-11-30
> > > **Response to Reviewer i9uk**
> > >
> > > Dear Reviewer i9uk,
> > >
> > > We are happy to hear that our rebuttal addressed your concerns well! Also, we appreciate your support for our work. If you have any further questions or suggestions, please do not hesitate to let us know.
> > >
> > > Best regards,
> > > Authors

---

### Official Review · Reviewer_vfsY · 2024-11-03

**Soundness:** 3
**Presentation:** 3
**Contribution:** 3
**Rating:** 8
**Confidence:** 3

**Summary:**

This paper presents a technique called REPA, which speeds up the process of training diffusion transformers. The high-level idea of REPA is to align the diffusion model representation with the visual representation of pretrained models. Experimental results show that after using REPA, the training process of diffusion transformers can be much faster, and also achieve better performance when training for the same number of steps.

**Strengths:**

++ The idea flow of this paper is very clear. First, the paper presents the semantic gap between diffusion transformer features and discriminative features. Then, the paper shows that the quality of the feature representations from diffusion transformers are correlated with the generation quality. Finally, the proposed method REPA is an intuitive solution based the previous observations and arguments. Therefore, the proposed technique has clear and reasonable motivations.

++ The benefit of applying REPA in training diffusion transformers seem to be significant to me. Experimental results like the quantitative ones from Tables 2-4 show that REPA helps the models reach the same performance with fewer iterations, and further achieve better generation results when trained for the same number of steps. Therefore, this technique can be useful in training diffusion models.

++ The proposed solution is simple but effective, just to align the features from diffusion models with pretrained visual foundation models. Therefore, it is a neat yet efficient solution for speeding up diffusion models.

**Weaknesses:**

-- The quantitative evaluations of REPA on DiT/SiT are only conducted on $256\times 256$ images, which is kind of low-resolution with respect to the capability of state-of-the-art diffusion models. I am wondering how REPA performs in higher-resolution cases, like $512\times 512$ which is also be supported by DiT/SiT. Is it still useful in training higher-resolution diffusion transformers? I think this evaluation is crucial, otherwise the application scenario of this technique will be relatively restricted to lower-resolution.

**Questions:**

-- I understand that this paper mainly focuses on speeding up diffusion transformers, as mentioned in the title. However, I am curious about why the application scenario is only limited in diffusion transformers instead of more general diffusion models. Is it because pretrained visual encoders like DINOv2 are mainly in vision transformer architecture, so enforcing the alignment of the features make more sense? Or the authors think diffusion transformers are the future trend for developing diffusion models, so studying the training dynamics of diffusion transformers is more meaningful?

---

> ### Author Response · Authors · 2024-11-22
> **Response to Reviewer vfsY**
>
> We deeply appreciate your insightful comments and efforts in reviewing our manuscript. We respond to each of your comments one-by-one in what follows. In the revised draft, we mark our major revisions as “blue”.
>
>
> ---
>
> **[W1] Higher resolution experiments (e.g., 512x512).**
>
> Thanks for the suggestion! We agree that high-resolution results will make the paper more convincing. To address this concern, we additionally train SiT-XL/2 on ImageNet-512x512 with REPA and report the quantitative evaluation in the table below. Notably, as shown in this table, the model (with REPA) already outperforms the vanilla SiT-XL/2 in terms of four metrics (FID, sFID, IS, and Rec) using >3x fewer training iterations. We added this result in Appendix J in the revised manuscript with qualitative results. We will also add the final results in the final draft with longer training (please understand that it takes more than 2 weeks, so it will not be possible to update them until the end of the discussion period).
>
> \begin{array}{lcccccc}
> \hline
> \phantom{0}\phantom{0} \text{Model} & \text{Epochs} & \text{FID $\downarrow$} & \text{sFID $\downarrow$} & \text{IS $\uparrow$} & \text{Pre. $\uparrow$} & \text{Rec. $\uparrow$} \newline
> \hline
> \text{\emph{Pixel diffusion}} \newline
> \phantom{0}\phantom{0} \text{VDM++} & - & 2.65 & - & 278.1 & - & - \newline
> \phantom{0}\phantom{0} \text{ADM-G, ADM-U} & 400 & 2.85 & 5.86 & 221.7 & 0.84 & 0.53 \newline
> \phantom{0}\phantom{0} \text{Simple diffusion (U-Net)} & 800 & 4.28 & - & 171.0 & - & - \newline
> \phantom{0}\phantom{0} \text{Simple diffusion (U-ViT, L)} & 800 & 4.53 & - & 205.3 & - & - \newline
> \hline
> \text{\emph{Latent diffusion, Transformer}} \newline
> \phantom{0}\phantom{0} \text{MaskDiT} & 800 & 2.50 & 5.10 & 256.3 & 0.83 & 0.56 \newline
> \phantom{0}\phantom{0} \text{DiT-XL/2} & 600 & 3.04 & 5.02 & 240.8 & 0.84 & 0.54 \newline
> \phantom{0}\phantom{0} \text{SiT-XL/2} & 600 & 2.62 & 4.18 & 252.2 & 0.84 & 0.57 \newline
> \phantom{0}\phantom{0} \text{+REPA (ours)} & \phantom{0}80 & {2.44} & 4.21 & 247.3 & 0.84 & 0.56 \newline
> \phantom{0}\phantom{0} \text{+REPA (ours)} & 100 & 2.32 & \bf{4.16} &  255.7 & 0.84 & 0.56 \newline
> \phantom{0}\phantom{0} \text{+REPA (ours)} & 200 & \bf{2.08} & 4.19 & \bf{274.6} & 0.83 & \bf{0.58} \newline
> \hline
> \end{array}
>
> ---
>
> **[Q1] Why are the experiments focused on diffusion transformers?**
>
> We mainly focused on the diffusion transformer (DiT) architecture for several reasons. First, as you mentioned, recent state-of-the-art visual encoders (e.g., DINOv2) use vision transformer architecture, so we thought enforcing the alignment of the features with DiTs makes more sense. Next, many papers have demonstrated that DiT architecture has great scalability, thereby being used in recent state-of-the-art diffusion models like Flux, Stable Diffusion 3.5, and Sora. Finally, DiT does not contain complicated components (e.g., up/downsampling blocks, skip connections) compared with U-Nets; thus, it is easy for us to perform extensive analysis of REPA under a simplified design space. In these respects, our experiments mainly focused on diffusion transformers. Applying REPA with other diffusion model architectures like U-Net would be an interesting direction in the future.

---

> > ### Comment · Reviewer_vfsY · 2024-11-22
> >
> > Thanks the authors for the rebuttal! I think the provided additional experimental results on $512\times512$ solved my concern, and the explanation about why only experimenting with diffusion Transformers also makes sense. Therefore, I would like to raise my score to 8.
> >
> > By the way, a kind reminder that your included tables in the rebuttal only work occasionally on the website. In other cases, it is just Latex code without actually showing the table. Not sure if it is some technical issue, or probably it is better to use Markdown formatting.

---

> ### Author Response · Authors · 2024-11-23
> **Response to Reviewer vfsY**
>
> Dear Reviewer vfsY,
>
> We are happy to hear that our rebuttal addressed your concerns well. Also, we appreciate your support for our work. If you have any further questions or suggestions, please do not hesitate to let us know. Moreover, thanks for letting us know about the issues about the ables that we added in the response. We will take a look and update to Markdown formatting if the problem persists.
>
> Best regards,
> Authors

---

> > ### Comment · Reviewer_U4Y4 · 2024-11-25
> >
> > Hi, thanks for your rebuttal. I would like to raise my score.

---

### Official Review · Reviewer_K7ig · 2024-11-03

**Soundness:** 4
**Presentation:** 4
**Contribution:** 3
**Rating:** 10
**Confidence:** 4

**Summary:**

The paper introduces representation alignment for diffusion models, a new technique that aligns diffusion features with those of a pretrained foundation model during training. The method is motivated by the observation that the features of pretrained diffusion models inherently align with the features of pretrained, diffusion-based transformer models. Inspired by this, the authors conducted a thorough analysis of the correlation between features at different layers of a foundation model and a diffusion model, finding that certain layers exhibited higher feature correlation. To enhance this alignment, the authors introduced a representation alignment loss, which results in a performance boost. An extensive study was conducted with various foundation models and across different diffusion model layers to identify the most suitable model and layer for enforcing similarity. The authors achieved 17.5x faster convergence on DiT when trained with RePA.

**Strengths:**

1. The paper is well-motivated. The introduction section clearly states the logic and reasoning behind why and how the features of foundation models can be leveraged to improve the convergence of a diffusion model.
2. The authors analyze different visual foundation models for representation learning and demonstrate that RePA is invariant to the choice of foundation model, with almost all visual foundation models enhancing generation performance and convergence.
3. The authors also study the correlation between generation performance and discriminative classification performance using validation accuracy on ImageNet, showing that both exhibit a linear correlation. A model trained to provide better representation capabilities automatically achieves better generation quality as well.
4. These insights are extremely valuable and novel, representing a significant step forward in the field.

**Weaknesses:**

1. The main claims of the paper is valid in the cases where classifier free guidance is not applied. Could the authors provide an analysis of the benefit of RePA in the presence of classifier free guidance. Specifically it would be great to see the results in Figure 1 with classifier free guidance included
2. How well does RePA work when there is already an alignment with text like cross attention in text-to-image models. Will RePA cause a conflict in representation alignment between CLIP/T5 and Dinov2 representations and reduce performance? If time permits I would request the authors to add an analysis in this aspect. This could be a simple experiment like finetuning the existing model an small text to image datasets like CUB-200[3]. If not a detailed explanation of why/why not this may work would suffice.
3. Diffusion features are generally noisy in the encoder layers and tends to become clearer in the decoder[1,2]. However in REPA we find that the best suited layer is indeed in the encoder which suggests cleaner representations in the encoder itself. Could the authors comment on why although the correlations seems better in the decoder features, the encoder layer 8 is best suited for representation alignment? An answer to Q1 in the questions section would suffice for this query.
4. There are no limitations section or future works section in the manuscript. I would advise the authors to please include these as well in an updated version. The request for these sections is for enabling the research community to further tackle problems that the authors might not have time to tackle.


   [1] Tumanyan, N., Geyer, M., Bagon, S. and Dekel, T., 2023. Plug-and-play diffusion features for text-driven image-to-image translation. In Proceedings of the IEEE/CVF Conference on Computer Vision and Pattern Recognition (pp. 1921-1930).

   [2] Cao, M., Wang, X., Qi, Z., Shan, Y., Qie, X. and Zheng, Y., 2023. Masactrl: Tuning-free mutual self-attention control for consistent image synthesis and editing. In Proceedings of the IEEE/CVF International Conference on Computer Vision (pp. 22560-22570).

    [3] Wah, C., Branson, S., Welinder, P., Perona, P. and Belongie, S., 2011. The caltech-ucsd birds-200-2011 dataset.

**Questions:**

1. In Figure 3, was the analysis performed by training different networks with RePA applied at different layers or was the training process for a single layer and evaluation was performed at different layer? This aspect is not clear from the paper. Could the authors please include this clarification in the figure caption or the related section.
2. In cross attention layers, the main learning process happens through alignment of the image space and text space . Diffusion models leveraging cross attention functions better if the  cross attention is applied at different layers. Considering a contrast of REPA for diffusion alignment. Would doing representation alignment based training leveraging different levels of features from the foundation model and the diffusion model lead to an improve in performance? Could the authors comment on this.
3. Diffusion features are generally noisy towards the start of the sampling process and becomes clearer towards the end. In Figure 2a, the authors showed that the semantic gap for cleaner features tends to be lower. However does this also suggest that a time varying repa might lead to a even better boost in performance
4. In Figure 5, the authors have shown the benefit of different visual foundation models for represenation alignment. However, there is also a correlation that the linear probing performance also follows the same trend i.e
Dinov2> CLIP>MoCov3>DINO>MAE
Hence I'm curious if repa using a pre-trained SOTA VIT based classifier may further boost the performance. An experiment for a limited number of training iterations or a detailed explanation would suffice for this issue

---

> ### Author Response · Authors · 2024-11-22
> **Response to Reviewer K7ig (1/2)**
>
> We deeply appreciate your insightful comments and efforts in reviewing our manuscript. We respond to each of your comments one-by-one in what follows. In the revised draft, we mark our major revisions as “blue”.
>
> ---
>
> **[W1] Benefit of REPA with the classifier-free guidance: something like Figure 1?**
>
> First, we have already shown FID improvement with CFG: In Table 2 of our manuscript, REPA outperforms the vanilla SiT-XL/2 with CFG using 7x fewer iterations. We have also provided experimental results showing that the result can be further improved with longer training (1.96 at 1M iteration to 1.80 at 4M iteration). This has been highlighted in L482-L483 in our main manuscript.
>
> Nevertheless, following your suggestion, we additionally provide a similar plot to Figure 1 with classifier-free guidance in Appendix G in the revised manuscript. The plot also shows a similar trend to Figure 1 in our manuscript, validating the effectiveness of REPA with CFG.
>
> ---
>
> **[W2] How does REPA work in text-to-image models?**
>
> To address your concern, we have additionally conducted a text-to-image (T2I) generation experiment on MS-COCO [1], following the setup in previous literature [2]. We tested REPA on MMDiT [3], a variant of DiT designed for T2I generation, given that the standard DiT/SiT architectures are not suited for this task. As shown in the table below, REPA also shows clear improvements in T2I generation, highlighting the importance of alignment of visual representations even under the presence of text representations. In this respect, we believe that scaling up REPA training to large-scale T2I models will be a promising direction for future research. We added these results with qualitative comparison in Appendix K in the revised manuscript.
>
> \begin{array}{lcc}
> \hline
> \text{Method} & \text{Type} & \text{FID} \newline
> \hline
> \text{AttnGAN} & \text{GAN} & 35.49 \newline
> \text{DM-GAN}  & \text{GAN} & 32.64 \newline
> \text{VQ-Diffusion} & \text{Discrete Diffusion} & 19.75 \newline
> \text{DF-GAN}  & \text{GAN} & 19.32 \newline
> \text{XMC-GAN} & \text{GAN} & \phantom{0}9.33  \newline
> \text{Frido}   & \text{Diffusion} & \phantom{0}8.97 \newline
> \text{LAFITE}  & \text{GAN} & \phantom{0}8.12 \newline
> \hline
> \text{U-Net}  & \text{Diffusion} & \phantom{0}7.32 \newline
> \text{U-ViT-S/2} & \text{Diffusion} & \phantom{0}5.95 \newline
> \text{U-ViT-S/2 (Deep)} & \text{Diffusion} & \phantom{0}5.48 \newline
> \hline
> \text{MMDiT (ODE; NFE=50, 150K iter)} & \text{Diffusion} & \phantom{0}6.05 \newline
> \textbf{MMDiT+REPA (ODE; NFE=50, 150K iter)} & \text{Diffusion} & \phantom{0}\bf{4.73} \newline
> \hline
> \text{MMDiT (SDE; NFE=250, 150K iter)} & \text{Diffusion} & \phantom{0}5.30 \newline
> \textbf{MMDiT+REPA (SDE; NFE=250, 150K iter)} & \text{Diffusion} & \phantom{0}\bf{4.14} \newline
> \hline
> \end{array}
>
> [1] Microsoft COCO: Common Objects in Context, ECCV 2014
> [2] All are Worth Words: A ViT Backbone for Diffusion Models, CVPR 2023
> [3] Scaling Rectified Flow Transformers for High-Resolution Image Synthesis, ICML 2024
>
> ---
>
> **[W3, Q1] In Figure 3, was the analysis performed by training different networks with RePA applied at different layers, or was the training process for a single layer and evaluation performed at different layers?**
>
> Thank you for pointing this out. The analysis was performed by using a single network trained with REPA at layer 8 and evaluated at different layers. This is why the model with REPA has the minimum representational gap at layer 8. This allows more layers after alignment to focus on capturing high-frequency details (based on a good representation) and achieves more improvement in generation performance, as explained in the “Alignment depth” paragraph in Section 4.2. As you suggested, we added clarification of Figure 3 in the caption in the revised manuscript.
>
> ---
>
> **[W4] No limitation or future works section.**
>
> Thanks for the suggestion. In this paper, we mainly focused on latent diffusion in the image domain. Exploring REPA with pixel-level diffusion or on other data domains like videos would be interesting future directions. Moreover, training large-scale text-to-image diffusion models with REPA will also be an interesting direction. Exploring theoretical insights into why REPA works well will also be an exciting future direction. We also added such discussions in Appendix M in the revised manuscript.

---

> ### Author Response · Authors · 2024-11-22
> **Response to Reviewer K7ig (2/2)**
>
> **[Q2] Representation alignment with different levels of features from the foundation model?**
>
> We think it is a really interesting future direction to explore. Given that recent diffusion model literature has shown that coarse-to-fine generation can make diffusion model training more efficient, we think that exploring the alignment of representations from foundation models and diffusion models in a hierarchical manner will be an exciting direction. We think there are many other possible directions, and we leave it for future work.
>
> ---
>
> **[Q3] Time-varying REPA: might lead to an even better boost in performance?**
>
> Thanks for the insightful suggestion. Indeed, as diffusion models deal with noisy inputs over different noise scales, time-varying REPA could lead to further performance improvements. One possible direction can be designing a weight function based on the noise schedule used in the diffusion process. We have not explored this in this work as our main focus is more on performing extensive analysis on other perspectives, such as target representations used for alignment, alignment depth, scalability of the method, etc. We leave this as an exciting direction for future work.
>
> ---
>
> **[Q4] Performance of REPA with the state-of-the-art ViT classifier?**
>
> Following your suggestion, we conducted an additional experiment using the VIT-L classifier fine-tuned from SWAG [4], which has 88.1% accuracy on ImageNet and is better than the visual encoder that we used in our experiment. As shown in the table below, it shows less improvement than other powerful self-supervised ViTs like DINOv2. In this respect, we hypothesize that aligning with a generic good visual representation learned from massive amounts of data is more effective in diffusion transformer training compared to discriminative representation on the target dataset.
>
> \begin{array}{lccccc}
> \hline
> \text{Method} & \text{FID $\downarrow$} & \text{sFID $\downarrow$} & \text{IS $\uparrow$} & \text{Pre. $\uparrow$} & \text{Rec. $\uparrow$} \newline
> \hline
> \text{DINOv2-L} & \phantom{0}9.9 & 5.34 & 111.9 & 0.68 & 0.65 \newline
> \text{ViT-L Classifier} & 11.4 & 5.24 & 100.3 & 0.68 & 0.64 \newline
> \hline
> \end{array}
>
> [4] Revisiting Weakly Supervised Pre-Training of Visual Perception Models, CVPR 2022

---

> ### Comment · Reviewer_K7ig · 2024-11-22
>
> I thank the authors for the detailed response. I have some doubts regarding some of the responses.
>
> [W2] Was the MS-COCO finetuning performed  by finetuning an existing model or was the training performed from scratch. In case it was trained with an existing models. From my understanding one of the main benefits RePA provides is faster convergence. Would this mean that finetuning with RePA may also lead to better performance?
>
> [W2] Was the representation alignment performed using DinoV2 embeddings or CLIP embeddings. Would there be an added advantage in performing RePA with CLIP embeddings for T2I models
>
> [W3] I thank the authors for this clarification. Seeing this response and after seeing the PCA visualization in Figure 38. Would performing RePA on later transformer layer embeddings (layers 16-24) than layer 8 be more suited. If Yes, I would appreciate if the authors may include this aspect in the future works section in the manuscript.
>
> [Q4] Dear authors, I thank for the experiment with the ViT classifier fine-tuned on SWAG[4]. I think the authors may have misread my question in the questions section. My concern was whether utilizing a classifier trained just for ViT classification with labels may perform better when compared to a network trained in a self-supervised fashion. From my understanding, In the experiment provided by the authors, another self supervised network was utilized.

---

> > ### Author Response · Authors · 2024-11-23
> > **Response to Reviewer K7ig**
> >
> > We are happy to hear that our response could help to address your concerns well. Due to your valuable and constructive suggestions, we also believe that our paper is much improved. We respond to each of your additional questions one-by-one in what follows. In the revised draft, we mark our additional revisions as “red”.
> >
> > **[W2-1] “Was the MS-COCO finetuning performed by finetuning an existing model or was the training performed from scratch. In case it was trained with an existing models. From my understanding one of the main benefits RePA provides is faster convergence. Would this mean that finetuning with RePA may also lead to better performance?”**
> >
> > We did not perform any finetuning here; rather, we trained a new MMDiT model from scratch on MS-COCO. From Table 3 and 4 in our manuscript, we show that REPA not only shows faster convergence but also better performance at the end of the training. We think exploring the effectiveness of REPA in fine-tuning setups will be a promising future direction. We have clarified these points in the revised manuscript.
> >
> > ---
> >
> > **[W2-2] “Was the representation alignment performed using DinoV2 embeddings or CLIP embeddings? Would there be an added advantage in performing RePA with CLIP embeddings for T2I models”**
> >
> > For the T2I experiment, we apply REPA DINOv2 embeddings, while the backbone MMDiT uses CLIP text embeddings as the condition. This is to more directly address the concern you asked: “Will RePA cause a conflict in representation alignment between CLIP/T5 and Dinov2 representations and reduce performance?” The results show that REPA does not suffer from conflict with CLIP text embeddings and can improve performance.
> >
> > Finally, as you mentioned, there might be better representations (like CLIP image embeddings) that can be more beneficial in T2I experiments; we leave it for future work as our main focus is not on applying REPA better for T2I setup.
> >
> > ---
> >
> > **[W3] “I thank the authors for this clarification. Seeing this response and after seeing the PCA visualization in Figure 38. Would performing RePA on later transformer layer embeddings (layers 16-20) than layer 8 be more suited. If Yes, I would appreciate if the authors may include this aspect in the future works section in the manuscript.”**
> >
> > We have empirically shown that applying REPA to layer 8 is more beneficial than later transformer layer embeddings, and we have already highlighted this in Table 2 and Section 4.2. Performing further analysis of this result will be an interesting direction. We added this aspect in future works section in the revised version of the manuscript.
> >
> > ---
> >
> > **[Q4] “Dear authors, I thank for the experiment with the ViT classifier fine-tuned on SWAG[4]. I think the authors may have misread my question in the questions section. My concern was whether utilizing a classifier trained just for ViT classification with labels may perform better when compared to a network trained in a self-supervised fashion. From my understanding, another self-supervised network was utilized in the experiment provided by the authors.”**
> >
> > Following your suggestion, we conducted an additional experiment by training SiT-L/2+REPA with the representation of a pretrained ViT-L trained just for classification (i.e., not pretrained in a self-supervised fashion). At 400K iterations, it achieves FID=11.6, which is worse than FID=9.9 of SiT-L/2+REPA with DINOv2-L representations. This highlights that using a strong visual representation, learned in a self-supervised manner from massive amounts of data, is more effective for training diffusion transformers than relying on a discriminative representation learned from direct classifier training.

---

> > > ### Comment · Reviewer_K7ig · 2024-11-25
> > >
> > > Dear authors,
> > >
> > > I thank you for clarifying these queries and including the future works section.
> > >
> > > 1. The experiments with T2I models portray that RePA is beneficial for T2I training as well in COCO dataset.
> > > 2. I understand that more experiments with future works may be required to find the best possible alignment scheme.
> > > 3. I thank the authors for including the relevance of different layers in the future works section.
> > > 4. Follow-up:-  Could the authors clarify which layers of the classifier were utilized in this experiment?
> > >
> > > [W2] Could you report the ImageNet accuracies of the classifier and Dino-v2 as well if time permits?

---

> > > > ### Author Response · Authors · 2024-11-30
> > > > **Response to Reviewer K7ig**
> > > >
> > > > We deeply appreciate your frequent communication with us. We respond to each of your additional questions one-by-one in what follows.
> > > >
> > > > ---
> > > >
> > > > **[AQ1] “Follow-up:- Could the authors clarify which layers of the classifier were utilized in this experiment?”**
> > > > Similar to our previous experimental setup, we used the last layer embeddings (i.e., before a classifier head) of the pretrained ViTs for experiments.
> > > >
> > > > ---
> > > >
> > > > **[AQ2] “Could you report the ImageNet accuracies of the classifier and Dino-v2 as well if time permits?”**
> > > >
> > > > We report the ImageNet accuracies of the ViTs in the table below. We also report FIDs of SiT-L/2 models trained with REPA using those of VIT classifiers. Moreover, following your suggestion, we additionally conducted experiments and reported results using DeIT III-L, a ViT classifier that has better accuracy than DINOv2-L.
> > > >
> > > >
> > > > \begin{array}{lcc}
> > > > \hline
> > > > \text{Model} & \text{Acc.}\uparrow & \text{FID}\downarrow \newline
> > > > \hline
> > > > \text{ViT-L Classifier (Vanilla)} & 79.7 & 11.6 \newline
> > > > \text{DeiT III-L} & 84.2 & 10.0 & \newline
> > > > \hline
> > > > \text{DINOv2-L} & 83.5 & \phantom{0}9.9 \newline
> > > > \hline
> > > > \end{array}
> > > >
> > > > As shown in this figure, using a ViT classifier shows a worse FID than using DINOv2, even with the classifier that has a better accuracy than DINOv2. This highlights the effectiveness of using a strong visual representation learned in a self-supervised manner from massive amounts of data for diffusion transformer training.

---

> > > > > ### Comment · Reviewer_K7ig · 2024-11-30
> > > > >
> > > > > I thank the authors for conducting the additional experiments.
> > > > >
> > > > > The new experiments demonstrate a strong correlation between ImageNet classification performance and FID scores. However, I politely disagree with the authors' statement that "using a ViT classifier shows a worse FID than using DINOv2, even with a classifier that has better accuracy than DINOv2." From the presented table, the numbers appear almost equal. I suggest the authors include the performance in the presence of classifier-free guidance as well.
> > > > >
> > > > > The observed pattern across different classifiers indicates that better classification performance is correlated to better FID scores.
> > > > >
> > > > > Thus, alignment with DINOv2 embeddings may not be ideal for text-to-image modeling. A representation closer to the text embeddings being used might be more suitable, particularly when leveraging RePA for text-to-image modeling. This opens up avenues for future work, such as HART [1], where text embeddings are derived from an LLM. In such cases, aligning the vision encoder's image embeddings (of the corresponding MLLM) with the latent features may lead to better performance compared to DINOv2.
> > > > >
> > > > > While I understand this discussion may be beyond the scope of the current paper, I encourage the authors to consider adding this aspect to the future works section.
> > > > >
> > > > > I deeply appreciate the authors' efforts in presenting the requested experiments and fostering active discussion. I believe the ideas in this paper are highly valuable to the research community, and I have therefore increased my score to 10.
> > > > >
> > > > > [1] Tang, Haotian, et al. "Hart: Efficient Visual Generation with Hybrid Autoregressive Transformer." arXiv preprint arXiv:2410.10812 (2024).

---

> > > > > > ### Author Response · Authors · 2024-11-30
> > > > > > **Response to Reviewer K7ig**
> > > > > >
> > > > > > Dear Reviewer K7ig,
> > > > > >
> > > > > > We sincerely appreciate your incisive and thoughtful comments on our manuscript. Thanks to your valuable and constructive suggestions, we believe our paper has improved significantly.
> > > > > >
> > > > > > As you suggested, we will include additional results with CFG and related discussions in the future works section in the final draft. Additionally, we will continue to explore the strong correlation between ImageNet classification performance and FID scores and will incorporate it in the final draft.
> > > > > >
> > > > > > Thank you once again for your invaluable feedback.
> > > > > >
> > > > > > Best regards,
> > > > > > Authors

---

### Official Review · Reviewer_U4Y4 · 2024-11-04

**Soundness:** 3
**Presentation:** 3
**Contribution:** 3
**Rating:** 8
**Confidence:** 4

**Summary:**

This paper presents REPresentation Alignment (REPA), a regularization technique designed to enhance the efficiency and quality of training in diffusion models. By aligning the noisy input states within denoising networks with clean image representations taken from pretrained visual encoders, REPA helps diffusion models learn better internal representations. The approach significantly improves training efficiency—such as a 17.5× speed-up for SiT models—and boosts generation quality, achieving a FID score of 1.80 on ImageNet with guidance. The authors emphasize the importance of aligning diffusion model representations with powerful self-supervised representations to make training easier and more effective.

**Strengths:**

- I like the idea of combining diffusion model training and representation learning. The authors highlight the importance of aligning diffusion model representations with powerful self-supervised representations to make training easier and more effective.

- The writing in the paper is clean and easy to follow. The authors start with their observations from the empirical study on pretrained diffusion models, which provides the audience with enough background to understand the problem.

- The experimental section of this paper is solid. They conduct extensive ablation studies on the proposed method, including aspects such as encoder type, encoder size, alignment depth, and more.

**Weaknesses:**

- My main concern with this work is that the authors focus on class-conditioned diffusion models rather than text-conditioned diffusion models. The success of diffusion heavily depends on text prompts, as better prompts lead to improved performance and faster training of the diffusion model. I am worried that representation learning may not be necessary for text-conditioned diffusion models, as the text itself already acts as a regularizer to guide the training. Additionally, the improvement in FID with classifier-free guidance (CFG) in this paper is not significant, which partially validates my concern that additional representation learning may not be needed when there is strong guidance.

- The empirical study presented in Figure 2 is based on linear probing, which is unfair to the diffusion model as it is trained with reconstruction objective. In this context, I suspect that the performance gap for the diffusion model would be smaller on tasks like semantic segmentation.

**Questions:**

I believe this is a solid piece of work and an important first step toward combining representation learning and diffusion model training. My main concern is whether this approach is generalizable to text-conditioned diffusion models. Additionally, I would suggest that the authors add [a] and [b] to the literature review:

[a] Learning Disentangled Representation by Exploiting Pretrained Generative Models: A Contrastive Learning View. ICLR 2022.

[b] DisDiff: Unsupervised Disentanglement of Diffusion Probabilistic Models. NeurIPS 2023.

---

> ### Author Response · Authors · 2024-11-22
> **Response to Reviewer U4Y4 (1/2)**
>
> We deeply appreciate your insightful comments and efforts in reviewing our manuscript. We respond to each of your comments one-by-one in what follows. In the revised draft, we mark our major revisions as “blue”.
>
> ---
>
> **[W1-1] Focus is class-conditioned diffusion models rather than text-conditioned diffusion models.**
>
> To address your concern, we have additionally conducted a text-to-image (T2I) generation experiment on MS-COCO [1], following the setup in previous literature [2]. We tested REPA on MMDiT [3], a variant of DiT designed for T2I generation, given that the standard DiT/SiT architectures are not suited for this task. As shown in the table below, REPA also shows clear improvements in T2I generation, highlighting the importance of alignment of visual representations even under the presence of text representations. In this respect, we believe that scaling up REPA training to large-scale T2I models will be a promising direction for future research. We added these results with qualitative comparison in Appendix K in the revised manuscript.
>
> \begin{array}{lcc}
> \hline
> \text{Method} & \text{Type} & \text{FID} \newline
> \hline
> \text{AttnGAN} & \text{GAN} & 35.49 \newline
> \text{DM-GAN}  & \text{GAN} & 32.64 \newline
> \text{VQ-Diffusion} & \text{Discrete Diffusion} & 19.75 \newline
> \text{DF-GAN}  & \text{GAN} & 19.32 \newline
> \text{XMC-GAN} & \text{GAN} & \phantom{0}9.33  \newline
> \text{Frido}   & \text{Diffusion} & \phantom{0}8.97 \newline
> \text{LAFITE}  & \text{GAN} & \phantom{0}8.12 \newline
> \hline
> \text{U-Net}  & \text{Diffusion} & \phantom{0}7.32 \newline
> \text{U-ViT-S/2} & \text{Diffusion} & \phantom{0}5.95 \newline
> \text{U-ViT-S/2 (Deep)} & \text{Diffusion} & \phantom{0}5.48 \newline
> \hline
> \text{MMDiT (ODE; NFE=50, 150K iter)} & \text{Diffusion} & \phantom{0}6.05 \newline
> \textbf{MMDiT+REPA (ODE; NFE=50, 150K iter)} & \text{Diffusion} & \phantom{0}\bf{4.73} \newline
> \hline
> \text{MMDiT (SDE; NFE=250, 150K iter)} & \text{Diffusion} & \phantom{0}5.30 \newline
> \textbf{MMDiT+REPA (SDE; NFE=250, 150K iter)} & \text{Diffusion} & \phantom{0}\bf{4.14} \newline
> \hline
> \end{array}
>
> [1] Microsoft COCO: Common Objects in Context, ECCV 2014
> [2] All are Worth Words: A ViT Backbone for Diffusion Models, CVPR 2023
> [3] Scaling Rectified Flow Transformers for High-Resolution Image Synthesis, ICML 2024
>
> ---
>
> **[W1-2] it seems FID improvements with classifier-free guidance are not significant.**
>
> We politely disagree with your concern. We strongly believe that we have already shown that FID improvement is also quite significant with CFG; for instance, in Table 2 of our manuscript, REPA outperforms the vanilla SiT-XL/2 with CFG using 7x fewer iterations. We have also provided experimental results showing that the result can be further improved with longer training (1.96 at 1M iteration to 1.80 at 4M iteration). This has also been highlighted in L482-L483 in our main manuscript. Moreover, the above text-to-image generation results in [W1-1] are achieved with CFG, where REPA shows considerably better FID scores (5.30$\to$4.14). To further address your concern, we also conducted an additional experiment on ImageNet 512x512 and compared FID values with CFG: SiT-XL/2 trained with REPA outperforms the vanilla SiT-XL/2 FID by improving values from 2.62 to 2.08 using 3x fewer training iterations.
>
> Finally, we additionally provide a similar plot to Figure 1 with classifier-free guidance in Appendix G in the revised manuscript. The plot also shows similar trends to those in Figure 1 in our manuscript, validating the effectiveness of REPA even with CFG.
>
>
>
> ---
>
> **[W2] Linear probing comparison is unfair: Performance for the diffusion model would be smaller on tasks like semantic segmentation.**
>
> We note that linear probing has been widely used in previous diffusion model representation analysis [4, 5, 6, 7]. Moreover, we could not perform the semantic segmentation that you mentioned because most open-sourced pretrained diffusion transformers (particularly DiT and SiT) are latent diffusion models. This makes the spatial resolution of DiTs much smaller (8x) in contrast to pretrained visual encoders, and thus it is difficult to perform semantic segmentation in this latent space. Nevertheless, we agree that exploring representational gaps using other tasks like dense prediction (e.g. [8]) should be an interesting direction in the future by training an additional pixel-level diffusion model with DiT architectures.
>
> [4] Deconstructing Denoising Diffusion Models for Self-Supervised Learning, arXiv 2024
> [5] Denoising Diffusion Autoencoders are Unified Self-supervised Learners, ICCV 2023
> [6] Diffusion Models Beat GANs on Image Classification. arXiv 2023
> [7] Do Text-free Diffusion Models Learn Discriminative Visual Representations?, ECCV 2024
> [8] Diffusion Model as Representation Learner, ICCV 2023

---

> ### Author Response · Authors · 2024-11-22
> **Response to Reviewer U4Y4 (2/2)**
>
> **[Q1] Add literature review: [a] and [b]**
>
> Thanks for introducing relevant works. [a] proposes a method to learn disentangled representations using a variety of frozen generative models, such as pretrained GANs. [b] also tries to solve a similar problem but focuses on disentangling diffusion model representations.  Our goal is to improve diffusion model training using frozen visual representations. We also did several analyses using pretrained models. We also added this discussion and citations in the revised manuscript.

---

> > ### Author Response · Authors · 2024-11-30
> > **Response to Reviewer U4Y4**
> >
> > Dear Reviewer U4Y4,
> >
> > We are happy to hear that our rebuttal addressed your concerns well! Also, we appreciate your support for our work. If you have any further questions or suggestions, please do not hesitate to let us know.
> >
> > Best regards,
> > Authors

---

> ### Author Response · Authors · 2024-11-25
> **Gentle Reminder**
>
> Dear Reviewer U4Y4,
>
> Thank you again for your time and efforts in reviewing our paper.
>
> As the discussion period draws close, we kindly remind you that two days remain for further comments or questions. We would appreciate the opportunity to address any additional concerns you may have before the discussion phase ends.
>
> Thank you very much!
>
> Many thanks,
> Authors

---

### Official Review · Reviewer_XnZq · 2024-11-04

**Soundness:** 4
**Presentation:** 3
**Contribution:** 4
**Rating:** 10
**Confidence:** 4

**Summary:**

The paper presents a novel argument that a primary bottleneck in training large-scale diffusion models for generation lies in learning effective representations. The authors suggest that training can be significantly simplified when the model is supported by strong external visual representations. To address this, they introduce a straightforward regularization technique called REPresentation Alignment (REPA), which aligns the projections of noisy input hidden states in denoising networks with clean image representations obtained from pretrained external visual encoders, such as DINOv2. Comprehensive experiments validate this perspective, reinforcing the paper’s innovative viewpoint.

**Strengths:**

1. The paper’s key finding—that self-supervised representations, like those from DINOv2, can serve as a teacher to distill base features in a diffusion model—is both novel and insightful. By using these high-quality representations, the diffusion model can more quickly learn a robust semantic foundation, allowing deeper layers to focus on higher-frequency details. This approach is innovative and inspiring.
2. The experimental setup is thorough, covering various aspects such as types of self-supervised encoders, encoder scalability, the number and location of layers used for regularization, and different alignment objectives. These comprehensive experiments add credibility to the proposed method.

**Weaknesses:**

1. High-resolution results (e.g., 512x512) are needed to support and further validate findings.
2. FID may be unreliable here, as it relies on the Inception network, which tends to lose significant spatial information and focusing on class information. I recommend re-evaluating using DINO-FID to obtain a more accurate measure. I would consider raising the score to 10 if state-of-the-art fid is demonstrated at resolution 512x512 and DINO-FID at 256x256. For DINO-FID, it’s necessary to pass the ImageNet training set through DINOv2 to store features for dino-FID calculation, which might be challenging to implement. If resources are limited, DINO-FID can be computed at 256x256 resolution, while 512x512 results can use the Inception FID npz provided by ADM.
3. It would be beneficial to include visualizations of the features before and after alignment. Although differences can already be observed in the generated images, since the core concept is feature alignment, visualizing the impact directly at the feature level would strengthen the argument and provide deeper insights.
4. I recommend exploring the effects of different diffusion objectives. For example, the intermediate features of an epsilon-prediction diffusion model might differ from those of a v-prediction or x0-prediction model. An analysis of these variations could provide valuable insights into the interaction between diffusion objectives and feature alignment.
5. The use of class conditions may not fully represent the entire generative domain, particularly in areas like text-to-image and text-to-video generation. This raises concerns about the generalization of the claim that feature alignment is universally important for generative tasks. It would be beneficial to clarify or qualify this claim in light of these broader application areas. In scenarios that already include rich semantic prompts, such as text-to-video generation, would the proposed method still be effective? DINO’s semantic extraction during training may differ from the semantics derived by an LLM when captioning or re-captioning. It would be helpful to include experiments that use detailed textual descriptions as conditions to evaluate the method’s performance under these circumstances.

**Questions:**

1. I recommend releasing the training and evaluation codes to support future research efforts, as feature representation plays a crucial role in diffusion models.
2. You mention that directly fine-tuning a DINOv2 model within a diffusion framework fails due to the noisy input. This raises some questions, as fine-tuning is generally quite robust and should be capable of handling noise. Additionally, have you considered training a diffusion model from scratch while incorporating a partial DINOv2-based loss (similar to a LPIPS loss) alongside the denoising loss? This approach might aid in feature preservation during denoising.
3. How do you handle cases where the spatial dimensions (height × width) differ between DINOv2 and DIT? Would using cross-attention mechanisms be a feasible method for aligning features in such cases? This might offer a flexible solution to manage discrepancies in spatial resolution between the models.

---

> ### Author Response · Authors · 2024-11-22
> **Response to Reviewer XnZq (1/3)**
>
> We deeply appreciate your insightful comments and efforts in reviewing our manuscript. We respond to each of your comments one-by-one in what follows. In the revised draft, we mark our major revisions as “blue”.
>
> ---
>
> **[W1] Higher-resolution results (e.g., 512x512).**
>
> Thanks for the suggestion! We agree that high-resolution results will make the paper more convincing. To address this concern, we additionally train SiT-XL/2 on ImageNet-512x512 with REPA and report the quantitative evaluation in the table below. Notably, as shown in this table, the model (with REPA) already outperforms the vanilla SiT-XL/2 in terms of four metrics (FID, sFID, IS, and Rec) using >3x fewer training iterations. We added this result in Appendix J in the revised manuscript, along with qualitative results. We will also add the final results in the final draft with longer training (please understand that it takes more than 2 weeks, so it will not be possible to update them until the end of the discussion period).
>
> \begin{array}{lcccccc}
> \hline
> \phantom{0}\phantom{0} \text{Model} & \text{Epochs} & \text{FID $\downarrow$} & \text{sFID $\downarrow$} & \text{IS $\uparrow$} & \text{Pre. $\uparrow$} & \text{Rec. $\uparrow$} \newline
> \hline
> \text{\emph{Pixel diffusion}} \newline
> \phantom{0}\phantom{0} \text{VDM++} & - & 2.65 & - & 278.1 & - & - \newline
> \phantom{0}\phantom{0} \text{ADM-G, ADM-U} & 400 & 2.85 & 5.86 & 221.7 & 0.84 & 0.53 \newline
> \phantom{0}\phantom{0} \text{Simple diffusion (U-Net)} & 800 & 4.28 & - & 171.0 & - & - \newline
> \phantom{0}\phantom{0} \text{Simple diffusion (U-ViT, L)} & 800 & 4.53 & - & 205.3 & - & - \newline
> \hline
> \text{\emph{Latent diffusion, Transformer}} \newline
> \phantom{0}\phantom{0} \text{MaskDiT} & 800 & 2.50 & 5.10 & 256.3 & 0.83 & 0.56 \newline
> \phantom{0}\phantom{0} \text{DiT-XL/2} & 600 & 3.04 & 5.02 & 240.8 & 0.84 & 0.54 \newline
> \phantom{0}\phantom{0} \text{SiT-XL/2} & 600 & 2.62 & 4.18 & 252.2 & 0.84 & 0.57 \newline
> \phantom{0}\phantom{0} \text{+REPA (ours)} & \phantom{0}80 & {2.44} & 4.21 & 247.3 & 0.84 & 0.56 \newline
> \phantom{0}\phantom{0} \text{+REPA (ours)} & 100 & 2.32 & \bf{4.16} &  255.7 & 0.84 & 0.56 \newline
> \phantom{0}\phantom{0} \text{+REPA (ours)} & 200 & \bf{2.08} & 4.19 & \bf{274.6} & 0.83 & \bf{0.58} \newline
> \hline
> \end{array}
>
>
> ---
>
> **[W2] DINO-FD on ImageNet-256x256.**
>
> This is a good point. Following your suggestion, we additionally measure the DINOv2-FD [1] of SIT-XL/2+REPA, as reported in the table below. It shows that our method achieves state-of-the-art DINO-FD on 256x256 resolution with a value of 63.90. Note that this improvement does not simply come from using SiT-XL/2 as a backbone; we observe that the vanilla SiT-XL/2 achieves an even worse DINOv2-FD of 90.93 than DiT-XL/2. We will also add the DINOv2-FD values on ImageNet 512x512 in the final draft after we complete training the 512x512 model with a large number of training iterations.
>
> \begin{array}{lc}
> \hline
> \text{Method} & \text{DINOv2-FD}\downarrow \newline
> \hline
> \text{BigGAN} & 401.22 \newline
> \text{RQ-Transformer} & 304.05 \newline
> \text{GigaGAN} & 228.37 \newline
> \text{MaskGIT} & 214.45 \newline
> \text{LDM} & 112.40 \newline
> \text{ADMG-ADMU} & 111.24 \newline
> \text{DiT-XL/2} & \phantom{0}79.36 \newline
> \hline
> \text{SiT-XL/2} & \phantom{0}90.93 \newline
> \text{+REPA (ours)} & \phantom{0}\bf{63.90} \newline
> \hline
> \end{array}
>
> Note that we follow the evaluation setup in [1] for a fair comparison with baseline results. Specifically, we do not tune the CFG coefficient for DINOv2-FD and use the same coefficient used for FID computation. To validate the improvement solely from REPA, We also report the DINOv2-FD of the vanilla SiT-XL/2 using the official implementation of SiT.
>
> [1] Exposing Flaws of Generative Model Evaluation Metrics and Their Unfair Treatment of Diffusion Models, NeurIPS 2023
>
> ---
>
> **[W3] Feature visualization before vs. after REPA.**
>
> Following your suggestion, we added a feature map visualization of pretrained SiT models before and after alignment in Appendix L in the revised manuscript. As shown in the figure, vanilla SiT-XL/2 exhibits a noisy feature map in the early layers at large timestep $t$. In contrast, with alignment, the model shows cleaner feature maps with clear coarse-to-fine patterns across model depths.

---

> ### Author Response · Authors · 2024-11-22
> **Response to Reviewer XnZq (2/3)**
>
> **[W4] Effect of different diffusion objectives.**
>
> We remark that our analyses on SiT (in Figure 2) and those on DiT (in Figure 9) already correspond to two different diffusion objectives, i.e., the v-prediction and $\epsilon$-prediction models, respectively. The results in Table 2 also validate the effectiveness of REPA for these models; we have clarified this point in the revised manuscript.
>
> To further address your concern, we additionally report the performance of DiT-L/2 + REPA using the x0-prediction objective in the table below. Similar to our experimental results with v-prediction or $\epsilon$-prediction objectives in Table 2, REPA is also effective in this case.
>
> \begin{array}{lc}
> \hline
> \text{Method} & \text{FID}\downarrow \newline
> \hline
> \text{DiT-L/2 ($\epsilon$-pred.)} & 23.3 \newline
> \text{DiT-L/2+REPA ($\epsilon$-pred.)} & 15.6 \newline
> \text{DiT-L/2+REPA ($\mathbf{x}_0$-pred.)} & 16.5 \newline
> \hline
> \end{array}
>
> ---
>
> **[W5] REPA for text-to-image or text-to-video generation**
>
> To address your concern, we additionally conducted a text-to-image (T2I) generation experiment on MS-COCO [2], following the setup in previous literature [3]. We tested REPA on MMDiT [4], a variant of DiT designed for T2I generation, given that the standard DiT/SiT architectures are not suited for this task. As shown in the table below, REPA also shows clear improvements in T2I generation, highlighting the importance of alignment of visual representations even under the presence of text representations. In this respect, we believe that scaling up REPA training to large-scale T2I models will be a promising direction for future research. We added these results with qualitative comparison in Appendix K in the revised manuscript.
>
> \begin{array}{lcc}
> \hline
> \text{Method} & \text{Type} & \text{FID} \newline
> \hline
> \text{AttnGAN} & \text{GAN} & 35.49 \newline
> \text{DM-GAN}  & \text{GAN} & 32.64 \newline
> \text{VQ-Diffusion} & \text{Discrete Diffusion} & 19.75 \newline
> \text{DF-GAN}  & \text{GAN} & 19.32 \newline
> \text{XMC-GAN} & \text{GAN} & \phantom{0}9.33  \newline
> \text{Frido}   & \text{Diffusion} & \phantom{0}8.97 \newline
> \text{LAFITE}  & \text{GAN} & \phantom{0}8.12 \newline
> \hline
> \text{U-Net}  & \text{Diffusion} & \phantom{0}7.32 \newline
> \text{U-ViT-S/2} & \text{Diffusion} & \phantom{0}5.95 \newline
> \text{U-ViT-S/2 (Deep)} & \text{Diffusion} & \phantom{0}5.48 \newline
> \hline
> \text{MMDiT (ODE; NFE=50, 150K iter)} & \text{Diffusion} & \phantom{0}6.05 \newline
> \textbf{MMDiT+REPA (ODE; NFE=50, 150K iter)} & \text{Diffusion} & \phantom{0}\bf{4.73} \newline
> \hline
> \text{MMDiT (SDE; NFE=250, 150K iter)} & \text{Diffusion} & \phantom{0}5.30 \newline
> \textbf{MMDiT+REPA (SDE; NFE=250, 150K iter)} & \text{Diffusion} & \phantom{0}\bf{4.14} \newline
> \hline
> \end{array}
>
> [2] Microsoft COCO: Common Objects in Context, ECCV 2014
> [3] All are Worth Words: A ViT Backbone for Diffusion Models, CVPR 2023
> [4] Scaling Rectified Flow Transformers for High-Resolution Image Synthesis, ICML 2024
>
> ---
>
> **[Q1] Code release.**
>
> Thank you! We plan to make all the training and eval code, model checkpoints, and supporting scripts from the paper open source, along with the necessary tools to replicate the results presented in the paper.
>
> ---
>
> **[Q2-1] Why direct DINOv2 fine-tuning fails?**
>
> We hypothesize that this is mainly due to the input mismatch problem: DINOv2 has never been exposed to noisy inputs, and they are trained with image pixels rather than compressed latent images computed from the Stable Diffusion VAE. Thus, DINOv2 initialization might not be that beneficial for training latent diffusion transformers.
>
> ---
>
> **[Q2-2] Diffusion model from scratch with partial DINOv2-based loss.**
>
> In our work, we did not consider such a training scheme for several reasons. First, since our method is inspired by analysis of the “internal” representation of diffusion transformers, we focus on a more explicit form of regularization that aligns between representations of diffusion transformers and powerful pretrained visual encoders. Moreover, applying an LPIPS-like loss using DINOv2 is both memory- and computation-intensive compared to REPA, particularly for latent diffusion models. This is because inputs of pretrained visual encoders (e.g., DINOv2) are image pixels, but outputs of diffusion transformers are compressed latent images. Thus, one should decode the output from diffusion transformers to image pixels through pretrained VAE at every training iteration before feeding them to pretrained visual encoders like DINOv2.

---

> ### Author Response · Authors · 2024-11-22
> **Response to Reviewer XnZq (3/3)**
>
> **[Q3] How to handle if the spatial dimensions differ between DINOv2 and DiT?**
>
> When spatial dimensions differ, we align them by interpolating the positional embeddings of pretrained visual encoders and resizing input images accordingly (e.g., as done in [5]). For instance, the original MoCov3 has a spatial dimension of 14x14 after patch embedding. However, the spatial dimensions of DiT/SiT models that we used in our experiments are 16x16; thus, we interpolated positional embedding of MoCov3 and resized input image resolution (from 224x224 to 256x256) to make the spatial dimension to be 16x16. This strategy also worked well with higher-resolution experiments (512x512 resolution experiments in [W1]) with DINOv2.
>
> [5] ResFormer: Scaling ViTs with Multi-Resolution Training, CVPR 2023.

---

> ### Comment · Reviewer_XnZq · 2024-11-24
> **The rebuttal address most of my concern, and i will raise my score to 10**
>
> 1. experiments of high-resolution 512x512 are provided, remaining good improvement
> 2. experiments of text-to-image are provided, showing good improvement even with text condition
> 3. the results of DINO-Fid are provided, demonstrating robust performance improvement
>
> Therefore, the rebuttal address most of my concerns, and i will raise my score to 10

---

> ### Author Response · Authors · 2024-11-24
> **Response to Reviewer XnZq**
>
> Dear Reviewer XnZq,
>
> We are happy to hear that our rebuttal addressed your concerns well! Also, we appreciate your support for our work. If you have any further questions or suggestions, please do not hesitate to let us know.
>
> Best regards,
> Authors

---

### Author Response · Authors · 2024-11-22
**Common Response**

Dear reviewers and AC,

We sincerely appreciate your valuable time and effort spent reviewing our manuscript.

We propose REPA, a regularization for improving diffusion transformers by aligning their representations with powerful pretrained target representations. As reviewers highlighted, our method is well-motivated (K7ig, vfsY, i9uK, MZyg), simple yet effective (vfsY, i9uk), providing a new meaningful insight (XnZq, K7ig), demonstrated by strong empirical results and comprehensive experiments and analysis (all).
We appreciate your constructive feedback on our manuscript. In response to the comments, we have carefully revised and enhanced the manuscript as follows:

- ImageNet 512x512 experiment (Appendix J)
- Text-to-image experiment (Appendix K)
- Feature map visualization (Appendix L)
- Dataset analysis used for training target visual encoders (Appendix C.3)
- State-of-the-art FID on ImageNet 256x256  with guidance interval (Table 2)
- Limitation and Future Work (Appendix M)
- Fixing typos (L290)

In the revised manuscript, these updates are temporarily highlighted in blue for your convenience to check.

We sincerely believe that these updates may help us better deliver the benefits of the proposed REPA to the ICLR community.

Thank you very much,
Authors.

---

### Meta-Review · Area_Chair_963h · 2024-12-17

**Metareview:**

This paper introduces REPresentation Alignment (REPA), a simple yet effective regularization technique that significantly enhances the efficiency and performance of diffusion transformers by aligning their representations with pretrained self-supervised visual encoders. Reviewers praised the paper's strong empirical results, including substantial speedups in training (e.g., 17.5× faster for SiT) and improved generation quality, as well as the thorough experimental analyses across various settings. The clear motivation, innovative insights into representation learning, and demonstration of REPA's effectiveness in both class- and text-conditioned generative tasks make this work a strong contribution to the field.

**Additional Comments On Reviewer Discussion:**

During the rebuttal, reviewers raised concerns about higher-resolution experiments, text-to-image (T2I) generation, metric trends, and dataset leakage, alongside suggestions for exploring alignment depth and pretrained classifier comparisons. The authors addressed these by conducting additional experiments at 512x512 resolution, evaluating REPA on T2I tasks (MS-COCO), providing new metrics like CLIP score and Aesthetics, and clarifying their design choices and findings. These comprehensive responses resolved reviewer concerns, leading to increased scores and strong consensus on the paper's contributions and significance

---

### Decision · Program_Chairs · 2025-01-22

Accept (Oral)